# MITIGATING FORGETTING IN LOW-RANK ADAPTATION

## ABSTRACT

Parameter-efficient fine-tuning methods, such as Low-Rank Adaptation (LoRA), enable fast specialization of large pre-trained models to different downstream applications. However, this process often leads to catastrophic forgetting of the model's prior domain knowledge. We address this issue with **LaLoRA**, a weight-space regularization technique that applies a **La**place **a**pproximation to **Lo**w-**R**ank **A**daptation. Our approach estimates the model's confidence in each parameter and constrains updates in high-curvature directions, preserving prior knowledge while enabling efficient target-domain learning. By applying the Laplace approximation only to the LoRA weights, the method remains lightweight. We evaluate LaLoRA by fine-tuning a Llama model for mathematical reasoning and demonstrate an improved learning–forgetting trade-off, which can be directly controlled via the method's regularization strength. We further explore different loss landscape curvature approximations for estimating parameter confidence, analyze the effect of the data used for the Laplace approximation, and study robustness across hyperparameters.

## 1 INTRODUCTION

Fine-tuning large pre-trained models, such as LLMs, is essential for specializing them for downstream applications. But updating all parameters of a model is often prohibitively expensive at today's scales. To address this, Low-Rank Adaptation (LoRA) (Hu et al., 2021) has emerged as a popular tool for parameter-efficient fine-tuning (PEFT). LoRA freezes the original pre-trained weights and restricts training to additive low-rank adapter layers (Fig. 1, *left*). For a linear layer with $W_0 \in \mathbb{R}^{D_{out} \times D_{in}}$, LoRA cuts the trainable parameters from $D_{out} \cdot D_{in}$ to $2r \cdot (D_{out} + D_{in})$ for a small rank $r$.

However, fine-tuning suffers from a well-known side effect called *catastrophic forgetting*. During fine-tuning, the model forgets capabilities acquired during pre-training, such as commonsense reasoning or general knowledge. Although LoRA tends to forget less than full fine-tuning (Biderman et al., 2024), drops in source-domain accuracy are still a critical issue, exposing the classic stability-plasticity trade-off: how can we preserve the model's pre-training knowledge (i.e., high source-domain accuracy) while still being flexible enough to learn new tasks (i.e., high target domain accuracy)?

To improve this learning–forgetting trade-off, we propose **LaLoRA** (Fig. 1), a regularization method for fine-tuning that estimates parameter uncertainty via the Laplace approximation only for the LoRA adapters. Inspired by continual learning methods like EWC (Kirkpatrick et al., 2017) or OSLA (Ritter et al., 2018), we restrict parameter updates in directions critical for source-domain performance. Specifically, we estimate the importance of each LoRA parameter for pre-training loss by computing a Laplace approximation using (surrogate) source domain data batches. This provides a local Gaussian approximation around the pre-trained solution, capturing the uncertainty of each LoRA weight. Intuitively, parameters with low uncertainty (high curvature) are critical to the source task. Changes to them substantially increase the source-domain loss, and they should thus not be changed significantly. In contrast, parameters with high uncertainty (low curvature) can be freely adjusted to learn new tasks. During fine-tuning, LaLoRA uses these uncertainty estimates to regularize the updates, penalizing changes to important parameters. By applying this approximation only to the lightweight LoRA adapters, our method remains computationally efficient and serves as a drop-in for standard LoRA pipelines.

**Contributions.** We combine Laplace approximations to efficiently estimate parameter uncertainty of LoRA weights and mitigate forgetting during fine-tuning.

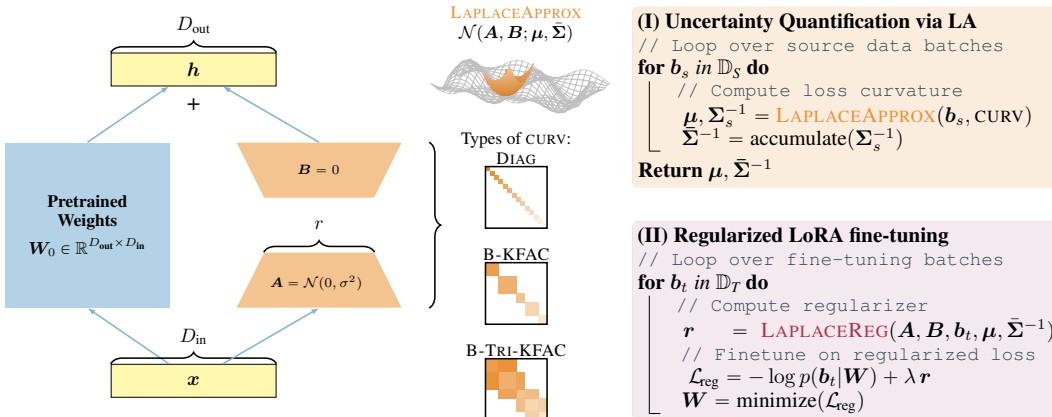

Figure 1: **Our Laplace regularizer LaLoRA maintains learning while limiting source domain forgetting.** The left panel illustrates the standard LoRA setup, where the pre-trained weights $W_0$ remain frozen and only the LoRA adapters $A$ and $B$ are trained. The center panel visualizes the Laplace approximation to the LoRA weights, supporting different curvature approximations to identify parameters critical for source performance. The right panel presents the **LaLoRA** algorithm in two parts: (I) uncertainty quantification of the trainable weights via the LAPLACEAPPROX and (II) LAPLACE-REGULARIZED fine-tuning on the target domain that mitigates source forgetting.

1. *We propose LaLoRA, a lightweight, curvature-aware regularizer for LoRA fine-tuning.* It brings together the efficiency of LoRA adapters and the Laplace approximation's ability to capture parameter uncertainty in order to protect source-critical directions (Section 3). The resulting regularizer offers a direct, controllable trade-off between prioritizing target-domain learning and source-domain preservation via the regularization strength $\lambda$. This simple, yet principled approach achieves a superior learning–forgetting Pareto frontier on an LLM fine-tuning task, compared to curvature-agnostic baselines, that rely on activation patterns or singular vectors (Section 5.1).

2. *We evaluate our method in different settings and provide practical guidance.* Our study (Section 5.2) probes LaLoRA's sensitivity to key fine-tuning hyperparameters (e.g., LoRA rank, training duration). Crucially, we show that LaLoRA is effective even when using only minimal surrogate data to estimate the source-domain curvature, highlighting its practical applicability in settings where original pre-training data is unavailable.

3. *We investigate three structured curvature approximations.* LaLoRA works flexibly with different curvature approximations to estimate parameter importance. We focus on three variants: (i) a computationally efficient diagonal approximation, (ii) a block-diagonal K-FAC version that also captures limited cross-parameter interactions, and (iii) a more expressive block tri-diagonal K-FAC approximation (Sections 3 and 5.2).

## 2 BACKGROUND

**Notation.** We consider a supervised learning setting, where the goal is to fine-tune a pre-trained model $f_W$, parameterized by its weights $W \in \mathbb{R}^D$, on a *target* dataset $\mathbb{D}_T = \{b_i = (x_i, y_i) \mid i = 1, \ldots, N\}$ containing batches of training inputs $x_i$ and outputs $y_i$. The pre-trained model's knowledge originates from a *source* domain, given (or approximated) by the source dataset $\mathbb{D}_S$, which might stem from different sub-datasets $\mathbb{D}_S = \{\mathbb{D}_{S_1}, \ldots, \mathbb{D}_{S_n}\}$ where $\mathbb{D}_{S_j} = \{b_i = (x_i, y_i) \mid i = 1, \ldots, N_s\}$. Our goal is to achieve high performance on the target dataset (*learning*) while avoiding a drop in performance on the source dataset (*forgetting*). The ideal trade-off between these objectives depends on the application, and is thus a choice for the practitioner. Throughout this paper, we quantify this balance by measuring learning as the gain in target-domain accuracy and forgetting as the drop in source-domain accuracy.

**LoRA.** Low-Rank Adaptation (Hu et al., 2021) s based on the premise that weight updates during fine-tuning have a low intrinsic rank. Instead of updating the full pre-trained weight matrix $W_0 \in \mathbb{R}^{D_{out} \times D_{in}}$,

LoRA freezes $\boldsymbol{W}_0$ and introduces additive updates $\Delta\boldsymbol{W}$ that have the low-rank decomposition $\Delta\boldsymbol{W} = \boldsymbol{B}\boldsymbol{A}$ where $\boldsymbol{B} \in \mathbb{R}^{D_{\text{out}} \times r}$ and $\boldsymbol{A} \in \mathbb{R}^{r \times D_{\text{in}}}$ for a small rank $r \ll \min(D_{\text{out}}, D_{\text{in}})$ (see Fig. 1, *left*). Fine-tuning is only done on the LoRA adapter weights $\boldsymbol{A}$ and $\boldsymbol{B}$. The forward pass is then computed as

$$\boldsymbol{h} = \boldsymbol{W}_0\boldsymbol{x} + \Delta\boldsymbol{W}\boldsymbol{x} = \boldsymbol{W}_0\boldsymbol{x} + \boldsymbol{B}\boldsymbol{A}\boldsymbol{x}. \tag{1}$$

The matrix $\boldsymbol{A}$ is usually initialized with random Gaussian noise and $\boldsymbol{B}$ with zeros such that the forward pass is initially identical to the pre-trained model before fine-tuning the LoRA adapter weights. This approach dramatically reduces the number of trainable parameters for fine-tuning, to $2r(D_{\text{in}} + D_{\text{out}})$ per module, down from the $D_{\text{in}}D_{\text{out}}$ parameters changed by full fine-tuning. However, standard LoRA provides no mechanism to prevent the updates $\Delta\boldsymbol{W}$ from interfering with the pre-trained knowledge stored in $\boldsymbol{W}_0$, often leading to catastrophic forgetting. Many studies have compared LoRA to full fine-tuning, some with a focus on source-domain forgetting (e.g. Dettmers et al., 2023; Ivison et al., 2023; Ghosh et al., 2024; Zhao et al., 2024b; Zhuo et al., 2024; Biderman et al., 2024; Shuttleworth et al., 2025).

**Laplace approximation.** From a Bayesian perspective, the posterior distribution over the model's parameters, $p(\boldsymbol{W} \,|\, \mathbb{D}_S)$, describes the belief about parameter values after observing source data and thus reflects (un-)certainty about each parameter's value. It, therefore, identifies which parameters are critical for source-domain performance (low uncertainty) and which still offer flexibility to learn new tasks (high uncertainty). The Laplace approximation (LA) (e.g. MacKay, 1992; Daxberger et al., 2022) provides a local Gaussian approximation to this typically intractable posterior. It stems from a second-order Taylor expansion of the loss around the parameter's maximum a posteriori (MAP) estimate, i.e. the trained $\boldsymbol{\mu}$, as $\mathcal{L}(\boldsymbol{W}, \mathbb{D}_S) \approx \mathcal{L}(\boldsymbol{\mu}, \mathbb{D}_S) + \frac{1}{2}\,(\boldsymbol{W} - \boldsymbol{\mu})^\top \bar{\boldsymbol{\Sigma}}^{-1}(\boldsymbol{\mu})\,(\boldsymbol{W} - \boldsymbol{\mu})$, where $\bar{\boldsymbol{\Sigma}}^{-1} = \nabla_{\boldsymbol{W}}^2 \log p(\mathbb{D}_S, \boldsymbol{W}) \in \mathbb{R}^{D \times D}$ is the Hessian of the loss with respect to the parameters. This results in a Gaussian distribution $p(\boldsymbol{W} \,|\, \mathbb{D}_S) \approx \mathcal{N}(\boldsymbol{W}; \boldsymbol{\mu}, \bar{\boldsymbol{\Sigma}}(\boldsymbol{\mu}))$ called Laplace approximation.

## 3 METHODOLOGY

Our proposed LaLoRA method consists of two stages: First, we quantify parameter uncertainty via the Laplace approximation STAGE I, before using this information to perform regularized LoRA fine-tuning STAGE II.

STAGE I: UNCERTAINTY QUANTIFICATION VIA LA (Figure 1, *top right*). Our goal is to create a regularizer that penalizes changes to weights critical for source-domain performance, which we do via the Laplace approximation. Fitting a Laplace approximation over the full model $\boldsymbol{W}$ is often computationally infeasible. We therefore restrict the approximation to only the trainable LoRA weights, i.e. $\Delta\boldsymbol{W} = \boldsymbol{B}\boldsymbol{A}$ for a chosen module. This dramatically reduces computational cost to a level practical on a single GPU. This targeted approach focuses the uncertainty estimate exclusively on the parameters that will be updated, allowing us to precisely constrain changes in high-curvature directions. Before fine-tuning, the LoRA adapters $\boldsymbol{A}$ and $\boldsymbol{B}$ are initialized such that $\boldsymbol{B}\boldsymbol{A}\boldsymbol{x} = 0$, meaning they do not alter the model's output vs. its pre-trained state $\boldsymbol{W}_0$ since $\boldsymbol{W}\boldsymbol{x} = \boldsymbol{W}_0\boldsymbol{x} + \boldsymbol{B}\boldsymbol{A}\boldsymbol{x} = \boldsymbol{W}_0\boldsymbol{x}$. Maintaining closeness to this initial "zero-effect" LoRA state for the important LoRA weights is key to mitigating forgetting.

To compute a Laplace approximation, we need access to the source data, or since pre-training data is typically unavailable, at least a surrogate representation of it, i.e. $\boldsymbol{b}_i \subset \mathbb{D}_S$. The Laplace approximation on the LoRA adapters is defined, then, by:

$$p(\Delta\boldsymbol{W} \,|\, \mathbb{D}_S) \approx \mathcal{N}(\Delta\boldsymbol{W}; \boldsymbol{\mu}, \bar{\boldsymbol{\Sigma}}) = \mathcal{N}\left(\boldsymbol{A}, \boldsymbol{B}; \begin{bmatrix} \boldsymbol{\mu}_A \\ \boldsymbol{\mu}_B \end{bmatrix}, \bar{\boldsymbol{\Sigma}}\right). \tag{2}$$

The mean $\boldsymbol{\mu}$ is equal to the LoRA weights at initialization (the pre-trained state) i.e. $\boldsymbol{\mu}_A$ is Gaussian noise and $\boldsymbol{\mu}_B = 0$, with $\bar{\boldsymbol{\Sigma}}^{-1}$ denoting the precision matrix. If multiple proxy sub-datasets are used, their individual precision matrices (each estimated as a mean precision from $N_s$ mini-batches) are summed, namely, $\bar{\boldsymbol{\Sigma}}^{-1} = \sum_{j=1}^{n} \boldsymbol{\Sigma}_{\mathbb{D}_{S_j}}^{-1}$. For brevity, we denote $\bar{\boldsymbol{\Sigma}}_j^{-1} := \boldsymbol{\Sigma}_{\mathbb{D}_{S_j}}^{-1}$ in what follows.

The challenge of STAGE I is to efficiently compute the precision matrices $\boldsymbol{\Sigma}^{-1}$ for the source domain sub-datasets. Our goal is to obtain a per-parameter precision without incurring full-matrix costs.

Firstly, we can simplify the precision $\mathbf{\Sigma}^{-1}$ of the LoRA adapters to a diagonal of a Fisher Information Matrix, as:

$$\mathbf{\Sigma}^{-1} = \mathrm{diag}(\boldsymbol{F}) \quad \text{where} \quad \boldsymbol{F} = \mathbb{E}\left[\left(\frac{\partial \log p(\boldsymbol{y} \,|\, \boldsymbol{x}, \Delta \boldsymbol{W})}{\partial \Delta \boldsymbol{W}}\right)\left(\frac{\partial \log p(\boldsymbol{y} \,|\, \boldsymbol{x}, \Delta \boldsymbol{W})}{\partial \Delta \boldsymbol{W}}\right)^{\top}\right] \quad (3)$$

In practice, we use the empirical diagonal Fisher computed on $N_s$ mini-batches $(\boldsymbol{x}_i, \boldsymbol{y}_i) \subset \mathbb{D}_{\mathrm{S}_j}$ for each sub-dataset.

$$(\mathbf{\Sigma}^{ij})^{-1} = \begin{bmatrix} \boldsymbol{D}_A^{ij} & 0 \\ 0 & \boldsymbol{D}_B^{ij} \end{bmatrix}, \quad (4)$$

$$\boldsymbol{D}_A^{ij} = \left(\frac{\partial \log p(\boldsymbol{y}_i \,|\, \boldsymbol{x}_i, \boldsymbol{A})}{\partial \boldsymbol{A}}\right)^2 \in \mathbb{R}^{rD_{\mathrm{in}} \times 1}, \qquad \boldsymbol{D}_B^{ij} = \left(\frac{\partial \log p(\boldsymbol{y}_i \,|\, \boldsymbol{x}_i, \boldsymbol{B})}{\partial \boldsymbol{B}}\right)^2 \in \mathbb{R}^{rD_{\mathrm{out}} \times 1}. \quad (5)$$

This approach, which we denote DIAG, results in the following regularizer.

$$\boldsymbol{r} = \mathrm{vec}(\boldsymbol{A} - \boldsymbol{\mu}_A)^{\top} \bar{\boldsymbol{D}}_A \mathrm{vec}(\boldsymbol{A} - \boldsymbol{\mu}_A) + \mathrm{vec}(\boldsymbol{B} - \boldsymbol{\mu}_B)^{\top} \bar{\boldsymbol{D}}_B \mathrm{vec}(\boldsymbol{B} - \boldsymbol{\mu}_B). \quad (6)$$

It is computationally efficient because it requires calculating only the gradient, it matches the second derivative of the loss near $\boldsymbol{\mu}$, and the precision is guaranteed to be positive semi-definite for standard loss functions. But the diagonal approximation naturally discards information about the interactions among weights within a layer, and across layers. Additionally, note that the diagonal Laplace on LoRA is also a diagonal Laplace on $\Delta \boldsymbol{W}$ as a whole (for more details see Appendix C.1). The resulting mean $\boldsymbol{\mu}$ and precision $\bar{\mathbf{\Sigma}}^{-1}$ together define the regularizer used in the next stage.

**Alternate curvature approximations.** LaLoRA works flexibly with different curvature approximations. To capture more complex interactions, we investigate two richer curvature structures based on Kronecker-Factored Approximations (K-FAC) (Martens and Grosse, 2020, see also Appendix A):

- B-K-FAC which looks at *intra*-layer interactions of each adapter and,
- B-TRI-K-FAC which adds *inter*-layer interactions between $\boldsymbol{A}$ and $\boldsymbol{B}$ for a given $\Delta \boldsymbol{W}$.

**B-K-FAC.** A *block-diagonal K-FAC* approximation is computed using the Kronecker factors, namely $\mathbf{\Lambda}_{i,i}^{L}$ and $\mathbf{\Lambda}_{j,j}^{R}$. The precision is given as:

$$\mathbf{\Sigma}^{-1} = \begin{bmatrix} \mathbf{\Sigma}_{A,A}^{-1} & 0 \\ 0 & \mathbf{\Sigma}_{B,B}^{-1} \end{bmatrix} = \begin{bmatrix} \mathbf{\Lambda}_{0,0}^{L} \otimes \mathbf{\Lambda}_{1,1}^{R} & 0 \\ 0 & \mathbf{\Lambda}_{1,0}^{L} \otimes \mathbf{\Lambda}_{2,2}^{R} \end{bmatrix}. \quad (7)$$

The regularizer resulting from such an approximation is defined as:

$$\boldsymbol{r} = \mathrm{vec}(\boldsymbol{A} - \boldsymbol{\mu}_A)^{\top} \mathrm{vec}(\mathbf{\Lambda}_{1,1}^{R}(\boldsymbol{A} - \boldsymbol{\mu}_A)\mathbf{\Lambda}_{0,0}^{L\top}) + \mathrm{vec}(\boldsymbol{B} - \boldsymbol{\mu}_B)^{\top} \mathrm{vec}(\mathbf{\Lambda}_{2,2}^{R}(\boldsymbol{B} - \boldsymbol{\mu}_B)\mathbf{\Lambda}_{1,1}^{L\top}). \quad (8)$$

Note that a diagonal Kronecker on the adapters is also a Kronecker (scaled by a single number) for the whole $\Delta \boldsymbol{W}$ (Appendix C.1).

**B-TRI-K-FAC.** Additionally, unlike the block-diagonal K-FAC used previously (Ritter et al., 2018; Yang et al., 2024), we treat the LoRA adapters $\boldsymbol{A}$ and $\boldsymbol{B}$ jointly, capturing both inter- and intra-layer interactions, i.e., $\mathbf{\Sigma}_{A,A}^{-1}, \mathbf{\Sigma}_{A,B}^{-1}, \mathbf{\Sigma}_{B,A}^{-1}, \mathbf{\Sigma}_{B,B}^{-1}$. This yields a *block tridiagonal K-FAC*:

$$\mathbf{\Sigma}^{-1} = \begin{bmatrix} \mathbf{\Sigma}_{A,A}^{-1} & \mathbf{\Sigma}_{A,B}^{-1} \\ \mathbf{\Sigma}_{B,A}^{-1} & \mathbf{\Sigma}_{B,B}^{-1} \end{bmatrix} = \begin{bmatrix} \mathbf{\Lambda}_{0,0}^{L} \otimes \mathbf{\Lambda}_{1,1}^{R} & \mathbf{\Lambda}_{1,0}^{L} \otimes \mathbf{\Lambda}_{1,2} \\ \mathbf{\Lambda}_{1,0}^{L} \otimes \mathbf{\Lambda}_{2,1}^{R} & \mathbf{\Lambda}_{1,0}^{L} \otimes \mathbf{\Lambda}_{2,2}^{R} \end{bmatrix}. \quad (9)$$

The regularizer for this approximation is efficiently computed with matrix vector products as follows:

$$\boldsymbol{r} = \mathrm{vec}(\boldsymbol{A} - \boldsymbol{\mu}_A)^{\top} \mathrm{vec}(\mathbf{\Lambda}_{1,1}^{R}(\boldsymbol{A} - \boldsymbol{\mu}_A)\mathbf{\Lambda}_{0,0}^{L\top}) + 2\mathrm{vec}(\boldsymbol{A} - \boldsymbol{\mu}_A)^{\top} \mathrm{vec}(\mathbf{\Lambda}_{1,2}^{R}(\boldsymbol{B} - \boldsymbol{\mu}_B)\mathbf{\Lambda}_{1,0}^{L\top})$$

$$+ \mathrm{vec}(\boldsymbol{B} - \boldsymbol{\mu}_B)^{\top} \mathrm{vec}(\mathbf{\Lambda}_{2,2}(\boldsymbol{B} - \boldsymbol{\mu}_B)\mathbf{\Lambda}_{1,0}^{L\top}). \quad (10)$$

**STAGE II: REGULARIZED LORA FINE-TUNING** (Figure 1, *bottom right*). We now treat the uncertainty estimate from the Laplace approximation as a Gaussian prior or weight-space regularizer $\boldsymbol{r}$ for fine-tuning. This transforms the standard training objective into a *Laplace-regularized* loss:

$$\mathcal{L}_{\mathrm{reg}}(\boldsymbol{W}, \mathbb{D}_{\mathrm{T}}) = \mathcal{L}(\boldsymbol{W}, \mathbb{D}_{\mathrm{T}}) + \lambda \boldsymbol{r}(\Delta \boldsymbol{W}, \boldsymbol{\mu}, \bar{\mathbf{\Sigma}}^{-1}) = -\log p(\mathbb{D}_{\mathrm{T}} \,|\, \boldsymbol{W}) + \lambda \boldsymbol{r}(\Delta \boldsymbol{W}, \boldsymbol{\mu}, \bar{\mathbf{\Sigma}}^{-1}) \quad (11)$$

$$= -\log p(\boldsymbol{y}_t \,|\, \boldsymbol{x}_t, \boldsymbol{W}) + \tfrac{\lambda}{2}\,(\Delta \boldsymbol{W} - \boldsymbol{\mu})^{\top}\,\bar{\mathbf{\Sigma}}^{-1}(\boldsymbol{\mu})\,(\Delta \boldsymbol{W} - \boldsymbol{\mu}), \quad (12)$$

with $\log p(\mathbb{D}_{\mathrm{T}} \,|\, \boldsymbol{W})$ the log-likelihood and $\boldsymbol{r}$ the regularizer scaled by $\lambda$. Through such an approach, we favor that the critical parameters (with high precision) stay near the pre-training solution $\boldsymbol{\mu}$ (informed by $\mathbb{D}_{\mathrm{S}}$). We allow variation in low-curvature directions but preserve those whose change would sharply raise pre-training loss. Minimizing the regularized loss (i) learns the target domain by maximizing the log-likelihood and (ii) mitigates forgetting of the source domain via minimizing the regularizer. This method is a simple plug-in regularizer without changes to the base LoRA method.

Importantly, this two-stage process is modular. If pre-computed uncertainty information (e.g. a Hessian) is available, e.g. as part of a model release (see Section 6), practitioners could skip STAGE I.

## 4 RELATED WORK

Our work combines elements from parameter-efficient fine-tuning, continual learning, and Bayesian inference, and we position our contributions in relation to these three areas.

**Mitigating forgetting in LoRA.** Catastrophic forgetting during fine-tuning of large pre-trained models is the main motivation of our study. Biderman et al. (2024) showed that LoRA retains more source-domain knowledge than full fine-tuning but improves less on the target task. Shuttleworth et al. (2025) found that while LoRA often forgets less than full fine-tuning, this is not universal. Their SVD analysis shows that hyperparameter choices can induce high-magnitude singular directions misaligned with pre-trained weights, and these directions drive the increased forgetting. We adapt the source/target domain setup for measuring the learning–forgetting trade-off from Biderman et al. (2024). Several strategies have been proposed to address forgetting in LoRA, typically by applying heuristics that identify and preserve important weights. Below, we describe the approaches most relevant to our work; see Appendix B for a broader review. MIGU (Du et al., 2024) reduces forgetting by updating only parameters with large $L^1$-normalized magnitudes in the linear layer's output, based on the observation that tasks show distinct activation patterns. The method uses forward-pass information and introduces a hyperparameter $t$ that controls the fraction of parameters to update. MiLoRA (Wang et al., 2025) constrains updates by initializing LoRA weights $\boldsymbol{A}$, $\boldsymbol{B}$ with the minor-$r$ singular components from the SVD of the pre-trained weight matrix, assuming those directions are less critical for pre-trained knowledge. Interestingly, PiSSA (Meng et al., 2025) instead initializes with the *top-r* singular vectors. Its main objective, however, is to accelerate convergence and match full fine-tuning rather than mitigating forgetting. Our method differs by leveraging the backward pass: we use loss curvature, a more direct measure of parameter sensitivity, to guide updates.

**Weight-space regularizers for continual learning.** The core idea of penalizing updates to important parameters has a rich history in continual learning. For example, EWC (Kirkpatrick et al., 2017) proposed a quadratic penalty on changes to important parameters, as measured by the diagonal of the Fisher information matrix. This concept was extended by subsequent works like OSLA (Ritter et al., 2018), which used more structured, Kronecker-factored curvature approximations. With LaLoRA, we adapt this classical, principled approach to the setting of parameter-efficient fine-tuning by applying a structured weight-spaced regularizer only to the compact LoRA space.

**Laplace approximations for LoRA.** The combination of the Laplace approximation and LoRA has been studied before, though for different purposes. Most notably, Yang et al. (2024) applied a post-hoc LA to an already LoRA fine-tuned model to improve uncertainty quantification and calibration. Our work differs in both goal and methodology. We use LA to estimate *source*-domain parameter importance and integrate it as a regularizer *during* fine-tuning to mitigate forgetting. We further extend the curvature approximation from block-diagonal to block tri-diagonal form.

## 5 EXPERIMENTS

We evaluate LaLoRA's ability to improve the trade-off between learning a new, specialized skill and retaining broad, pre-trained knowledge. Specifically, we fine-tune a pre-trained model on mathematical reasoning with the target of achieving similar performance to vanilla LoRA on the target domain, while exhibiting less forgetting on the source domain.

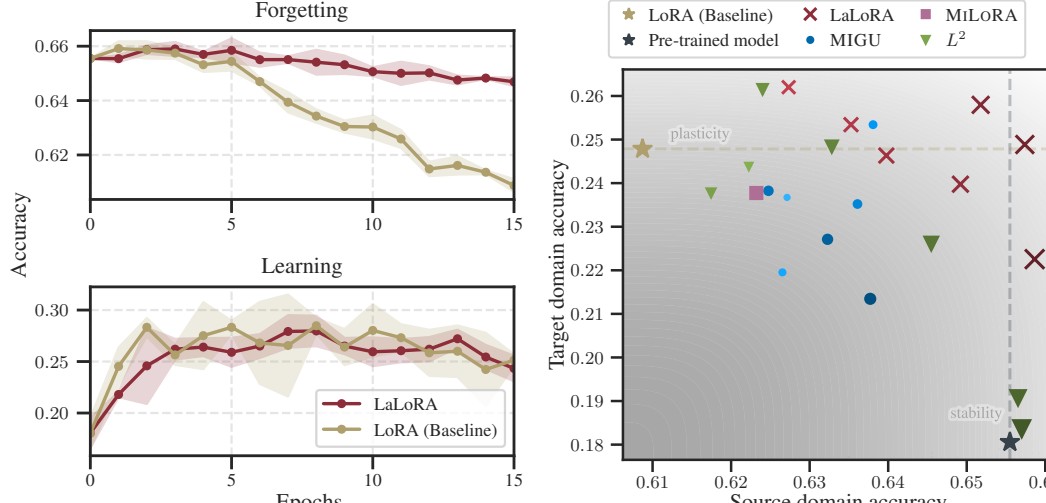

(a) **Diagonal LALoRA regularization reduces forgetting while maintaining good learning ability.** Accuracy while fine-tuning: (*top*) average source-domain accuracy (forgetting) and (*bottom*) target-dataset accuracy (learning). We show mean and standard deviation across three random seeds. LALoRA is shown for $\lambda_{\text{stability}}$, which retaining knowledge.

(b) **LaLoRA regularization improves the learning–forgetting trade-off.** Final source ($x$-axis) vs. target ($y$-axis) accuracy for LaLoRA and competing methods, averaged over three seeds. The marker size & brightness indicates the value of each method's hyperparameter controlling the learning–forgetting trade-off (e.g. $\lambda$ for LaLoRA, $t$ for MIGU).

Figure 2: **Mitigating source domain forgetting**: (a) Forgetting and learning dynamics over the course of fine-tuning on math reasoning data and (b) the resulting trade-off.

**Experimental setup.** We use a pre-trained LLAMA-3.2-3B [1] model (Grattafiori et al., 2024) as the base model for all experiments. Unless otherwise noted, all LoRA fine-tuning, including LaLoRA, use a fixed rank of $r = 16$. To assess the model's ability to learn a new task (plasticity), we perform instruction fine-tuning on GSM-8K ($\mathbb{D}_{\text{T}}$, Cobbe et al., 2021). It consists of high-quality, grade-school-level math problems, which require multi-step reasoning abilities. We call the final accuracy achieved after fine-tuning on GSM-8K the *target domain accuracy* (our measure for *learning*).

To quantify forgetting (stability), we would ideally evaluate the performance on pre-training data. For most foundation models, including LLAMA-3.2, this data is unavailable, and following Biderman et al. (2024); Shuttleworth et al. (2025); Du et al. (2024), we use three commonsense reasoning datasets as a proxy for the original pre-training source domain $\mathbb{D}_{\text{S}}$: (1) WINOGRANDE ($\mathbb{D}_{\text{S}_1}$, Sakaguchi et al., 2019) collects pronoun resolution problems requiring contextual understanding. (2) ARC CHALLENGE ($\mathbb{D}_{\text{S}_2}$, Clark et al., 2018) is a set of multiple-choice science questions requiring reasoning and scientific knowledge. (3) HELLASWAG ($\mathbb{D}_{\text{S}_3}$, Zellers et al., 2019) tests logical sentence continuations requiring a nuanced understanding of everyday situations. We call the final average accuracy across these three datasets *source domain accuracy* (our measure for *forgetting*). Unless noted, we compute the LA using $N_s = 1$ batch per sub-dataset. For full setup please see Table 2 and Table 3.

For this setup, we first present our main results, demonstrating diagonal LaLoRA's improved learning–forgetting trade-off compared to existing baselines (Section 5.1). We then conduct an in-depth analysis of LaLoRA's internal mechanisms, design choices, and hyperparameter sensitivity (Section 5.2).

## 5.1 LaLoRA Improves the Learning–Forgetting Trade-Off

**LaLoRA preserves more source knowledge than LoRA.** We first establish the performance of vanilla, unregularized LoRA (equivalent to LaLoRA with $\lambda = 0$) to quantify the extent of forgetting. The results, shown in Fig. 26 (Baseline, —), highlight the core issue of forgetting. The model is able to improve its performance on the fine-tuning task $\mathbb{D}_{\text{T}}$, however, its performance on the source-domain $\mathbb{D}_{\text{S}}$ drops substantially. In contrast, LaLoRA with a simple *diagonal* regularizer (Fig. 26, —)

---

[1]Model used: https://huggingface.co/meta-llama/Llama-3.2-3B

Table 1: **Comparing LaLoRA regularization to other LoRA fine-tuning methods**. Final source/target domain accuracy ($\pm$ one standard deviation across three seeds). The hyperparameters $\lambda_i$ or $t_i$ were set to either prioritize source accuracy (stability) or target accuracy (plasticity). The right-most columns shows percentage-point differences compared to the Baseline (LoRA), i.e. fine-tuning without regularizer, (last row). Higher is better for both forgetting and learning.

| | Source domain | Target domain | Forgetting | Learning |
|---|---|---|---|---|
| **LA-LoRA**, $\lambda_{\text{stability}}$ | $64.9\% \pm 0.1\%$ | $24.0\% \pm 1.7\%$ | $+4.0$ | $-0.8$ |
| **LA-LoRA**, $\lambda_{\text{plasticity}}$ | $63.5\% \pm 0.4\%$ | $25.3\% \pm 2.3\%$ | $+2.6$ | $+0.5$ |
| MIGU, $t_{\text{stability}}$ | $63.8\% \pm 0.3\%$ | $25.3\% \pm 1.1\%$ | $+2.9$ | $+0.5$ |
| MIGU, $t_{\text{plasticity}}$ | $63.8\% \pm 0.7\%$ | $21.3\% \pm 4.6\%$ | $+2.9$ | $-3.5$ |
| MiLoRA | $62.3\% \pm 0.2\%$ | $24.2\% \pm 1.1\%$ | $+1.4$ | $-0.6$ |
| $L^2$, $\lambda_{\text{stability}}$ | $65.7\% \pm 0.0\%$ | $18.4\% \pm 3.6\%$ | $+4.8$ | $-6.4$ |
| $L^2$, $\lambda_{\text{plasticity}}$ | $63.3\% \pm 0.4\%$ | $24.8\% \pm 1.4\%$ | $+2.4$ | $+0.0$ |
| Baseline (LoRA) | $60.9\% \pm 0.9\%$ | $24.8\% \pm 0.7\%$ | $+0.0$ | $+0.0$ |

limits source-domain forgetting significantly (roughly 4 percentage points (pp) improvement vs. the baseline) while achieving a comparable target-domain performance. For this illustrative example, the regularization strength $\lambda$ of LaLoRA was chosen to prioritize stability (i.e., least amount of forgetting); see (Tables 4 and 5 in Appendix D). Thus, our principled, curvature-aware approach effectively preserves pre-trained knowledge, even with a lightweight diagonal approximation.

**Controlling the learning–forgetting trade-off with $\lambda$.** LaLoRA provides a flexible mechanism to navigate the learning–forgetting trade-off. By varying the regularization strength $\lambda$, we can trace a full Pareto frontier of performance, shown in Fig. 2b. The series of red markers (✗) represent the final source- and target-domain accuracies for different values of $\lambda \in \{1, 10, 10^2, 10^3, 10^4, 10^5, 10^6\}$. Higher regularization strengths (indicated by both marker size and brightness, i.e. $\lambda = 1$ is the smallest and brightest ✗) increasingly penalize deviations from the source model, leading to less forgetting at the cost of less or slower learning (see Fig. 6 for this trade-off vs. epoch). The regularization strength $\lambda$ enables us to adjust this trade-off between prioritizing retaining knowledge and learning new tasks. This allows practitioners to select the optimal balance based on their specific application. To simplify the presentation in the remainder of the paper, we focus on specific hyperparameter choices that either maximize source accuracy ($\lambda_{\text{stability}}$) or a weighted average prioritizing target accuracy ($\lambda_{\text{plasticity}}$), in order to concisely cover important scenarios (see Appendix D for details).

**Pushing the frontier compared to existing methods.** Finally, we compare LaLoRA against other methods designed to mitigate forgetting, including MIGU (for different threshold values $t \in \{0.5, 0.6, 0.7, 0.75, 0.8, 0.85, 0.9\}$), MILoRA (no hyperparameters), and, for completeness, a simple $L^2$-regularizer. This $L^2$ regularizer penalizes the LoRA weights by $\lambda \|\Delta W - \mu\|_2^2$, which is essentially LaLoRA with an uninformed identity replacing $\bar{\Sigma}^{-1}$. From Fig. 2b and Table 1, we observe that the Pareto frontier traced by LaLoRA generally outperforms other methods. For example, our best stability-focused model achieves a source accuracy of $65.9\%$ compared to $60.9\%$ for the baseline and $63.8\%$ for the best stability-focused model of MIGU. We can also observe that the additional curvature information in LaLoRA compared to the $L^2$ baseline significantly helps. Hyperparameters were chosen via validation runs (Tables 4 and 5 in Appendix D). In Table 1, we report two validation-selected settings (stability- and plasticity-focused) per method from Fig. 2b. We also show per-seed performance in Figs. 7 and 8. Overall, we observe that LaLoRA establishes a superior Pareto frontier, offering better accuracy across nearly all stability–plasticity trade-offs.

## 5.2 UNDERSTANDING LaLoRA: UPDATE PATTERNS, CURVATURE ESTIMATION, AND SENSITIVITY

We now analyze LaLoRA's internal mechanisms, data dependency, curvature approximation, and its sensitivity to fine-tuning settings to provide a better understanding of its behavior.

**How LaLoRA guides weight updates.** We first verify our intuition that the weights deemed crucial for source-domain performance (i.e. weights with high curvature/precision, denoted "critical" weight

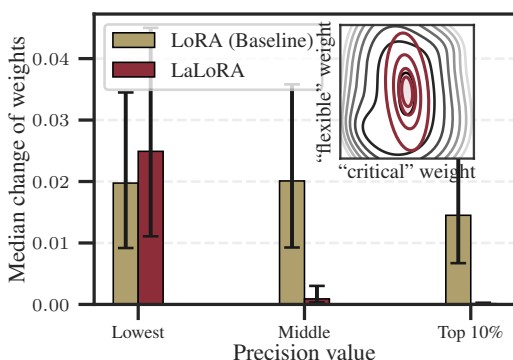

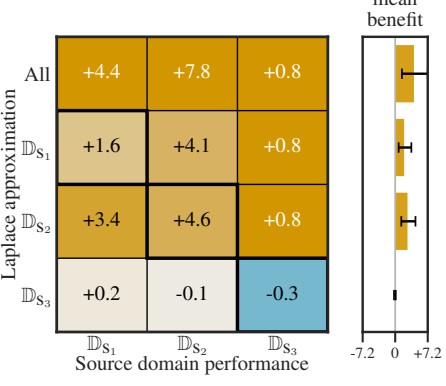

(a) **LaLoRA protects high-precision weights.** In contrast to the baseline's uniform approach, LaLoRA suppresses updates to high-precision weights ("Top 10%"), deemed critical for retaining source knowledge, while allowing low-precision parameters to adapt freely. *(Top right)* The inset illustrates this principle of identifying "critical" (high-curvature) vs. "flexible" (low-curvature) directions via the Laplace approximation (red).

(b) **Broader source data representation leads to less forgetting.** The heatmap illustrates the benefit (in percentage points) of using different surrogate datasets ($y$-axis) for the Laplace approximation on forgetting across the same three source datasets ($x$-axis). Higher values (orange) indicate better knowledge retention. The bar plot *(right)* shows the average benefit across all three datasets and three seeds.

Figure 3: **Analysis of LaLoRA's update pattern and data dependency.** (a) Unlike the baseline, LaLoRA limits updates to flexible (low-precision) weights. (b) Using a more comprehensive set of surrogate datasets for the Laplace approximation leads to less source-domain forgetting.

in the illustrative sketch Fig. 3a *(top right)*) are strongly regularized and thus change less during fine-tuning compared to more "flexible" weights (smaller curvature/precision). To study this, we categorize LoRA parameters into three groups based on their precision: flexible (lowest), middle, and important (top 10% highest precision). As hypothesized and shown in Fig. 3a, LaLoRA primarily modifies the "flexible" weights, allowing them to change during fine-tuning. Conversely, updates to the weights deemed critical for source-domain knowledge are highly suppressed (see Fig. 9 for its update distribution). This is in stark contrast to the unregularized baseline, which updates weights more uniformly across all groups (see Table 6). This validates our approach and demonstrates that low-curvature directions offer sufficient flexibility to learn the new task.

**The role of data in curvature estimation for the Laplace approximation.** Since pre-training data for foundation models is rarely public, we approximate source-domain curvature using three proxy commonsense reasoning datasets. As an ablation, summarized in Fig. 3b, we perform our Laplace approximation on only one of these three datasets $\mathbb{D}_{S_i}$ but measure forgetting on all three separately. We observe that even a single proxy dataset like WINOGRANDE ($\mathbb{D}_{S_1}$) or ARC ($\mathbb{D}_{S_2}$) is sufficient to significantly mitigate forgetting across all three tasks. This demonstrates that even a rough source domain surrogate can be beneficial. As expected, a broader representation (using all three datasets) yields the best results. Fig. 3b used $\lambda_{\text{stability}}$, see Fig. 10 for $\lambda$ values with similar results. We also observe that relatively little source-domain data is needed to improve the learning–forgetting trade-off. We find that just two batches per dataset deliver the best trade-off. Fig. 4 tests this trade-off based on the number of batches per source sub-dataset ($N_s \in \{1, 2, 3\}$) used for LA across three regularization strengths $\lambda$. All settings yield similar forgetting benefit. Using two batches offers the best overall trade-off, but one batch is a close approximation. Given the marginal gains and added cost, we adopt $N_s = 1$ as a practical default. While this ad-hoc estimation works, it also points to a new best practice for model releases. The need for such proxies could be removed if model providers could release a pre-computed LA, alongside the model weights, following a formal privacy review. This would offer a compact, powerful summary of parameter importance without explicitly exposing proprietary data. It would enable downstream model users to perform cheap and accurate curvature-aware fine-tuning like LaLoRA and unlock other applications like improving model calibration (e.g. Yang et al., 2024).

**Sensitivity to fine-tuning hyperparameters, curvature approximations, and computational costs.** We confirm that LaLoRA's benefits are not confined to a narrow experimental setting. As shown

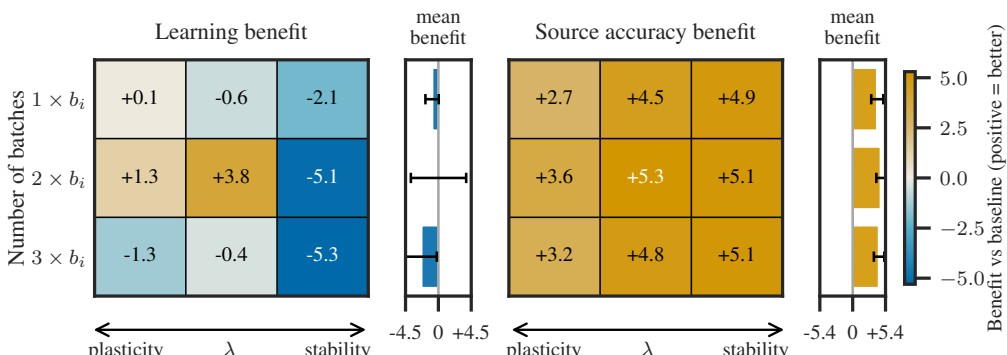

Figure 4: **Two batches per source dataset optimizes the learning–forgetting trade-off.** We analyze the impact of using one, two, or three batches ($y$-axis) per surrogate dataset for the Laplace approximation. Performance is measured as benefit over baseline (orange is better) for both target-domain learning (*left*) and source-domain accuracy (*right*) across three regularization strengths $\lambda$ ($x$-axis). The mean benefit subplots summarize the results across $\lambda$ and three seeds.

in Figs. 12 to 14, 18 and 20, the improved learning–forgetting trade-off is observed across a wide range of LoRA ranks, training durations, as well as for the individual source-domain datasets. We also investigate whether a more expressive curvature approximation, such as B-K-FAC (Section 3) provides additional benefits to the default LaLoRA variant using a diagonal approximation (Figs. 21 and 22 for using one & two data batches, respectively). Due to the low-rank structure of the LoRA update, different curvature approximations *on the LoRA weights* represent different covariance structures on the parameter space itself. In particular, a block-diagonal K-FAC approximation on the LoRA weights, despite containing a full Kronecker factor for the low-rank space, in fact collapses into a much simpler covariance structure on the parameter space that treats all low-rank terms the same (see Appendix C.1). This may be at least a partial explanation for why we do not see improved performance in our experimental ablations when moving from a diagonal to a block-diagonal K-FAC LA (Figs. 7 and 8 for per-seed results). We provide the memory and time requirements for computing the diagonal LA and the regularized training in Tables 9 and 10 (Appendix F).

## 6 CONCLUSION

**Summary.** This paper introduced LaLoRA, a practical, lightweight, and principled method to mitigate forgetting during LoRA fine-tuning. Our results demonstrate that a computationally efficient diagonal Laplace-regularizer on the LoRA weights effectively retains pre-trained knowledge while maintaining strong fine-tuning performance. This improves the learning–forgetting trade-off compared to existing baselines. We find this benefit is robust across fine-tuning scenarios and, crucially, can be achieved using only a small amount of proxy data to approximate the source-domain loss curvature.

**Limitations & future directions.** LaLoRA naturally inherits limitations common to weight-space regularization methods. It requires storing the curvature approximation, as well as, LoRA weights at initialization and introduces a regularization strength hyperparameter, $\lambda$. Setting $\lambda$ to achieve the optimal learning–forgetting trade-off requires tuning, which can be challenging in real-world continual learning or fine-tuning scenarios. Future work could focus on developing methods for automatically setting this hyperparameter. Additionally, recent work by Tatzel et al. (2025) demonstrated an efficient way to improve the Laplace approximation's accuracy by debiasing its stochastic estimates.

**The bigger picture.** This work reinforces the notion that parameter uncertainty, captured via curvature, is a fundamental property of trained models. We therefore argue for a new best practice: Open-weight model releases should provide more than the model weights (the point estimate provided by the MAP solution). As a critical complement, they should also include a pre-computed Laplace approximation. This lightweight summary of parameter importance would provide essential context for the model's weights, without requiring further access to pre-training data. This would unlock not only curvature-aware fine-tuning methods like LaLoRA but also broader applications in uncertainty quantification, paving the way for more robust, calibrated, and safely adaptable machine learning.

**Reproducibility Statement.** We fine-tune Llama 3.2 using LoRA, with the exact setup detailed in Tables 2 and 3. All hyperparameters are reported in Tables 4 and 5. The methodology is described in Section 3 and Appendix C. The datasets used are publicly available and listed in Section 5. The full codebase and configuration files required to reproduce our experiments will be released publicly.

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

# Appendix

## A BACKGROUND

**Full-fine-tuning:** Fine-tuning, adapting a pre-trained model to a specific task, has been extensively studied. Full fine-tuning is inefficient because *every* parameter in the model must be updated. For instance, Xu et al. (2023) estimate that fully fine-tuning Falcon-180B would require about 5.1 TB of GPU memory. Hu et al. (2021) observed that finetuning GPT-3 175B requires about 1.2 TB of VRAM, and is therefore impractical for most users. A practical alternative is to update only a subset of layers. Parameter-Efficient Fine-Tuning (PEFT, Houlsby et al., 2019) offers several such techniques—additive, partial, re-parameterized, hybrid, and unified fine-tuning, that greatly reduce the number of trainable parameters.

**Transformers:** (Vaswani et al., 2023) proposed a compilation of encoder-decoder modules with attention mechanism. For an input $\boldsymbol{X} \in \mathbb{R}^{n_s \times D_{\text{in}}}$ where $n_s$ is the length of the sequence and $D_{\text{in}}$ is the hidden dimension, we transform the input via query, key and value vectors given as $\boldsymbol{K}, \boldsymbol{Q}, \boldsymbol{V}$:

$$\boldsymbol{K} = \boldsymbol{X}\boldsymbol{W}_k + \boldsymbol{b}_k, \boldsymbol{Q} = \boldsymbol{X}\boldsymbol{W}_q + \boldsymbol{b}_q, \boldsymbol{V} = \boldsymbol{X}\boldsymbol{W}_v + \boldsymbol{b}_v \tag{13}$$

**Kronecker-factored Approximate Curvature (K-FAC):** Martens and Grosse (2020) propose an approximation to the loss curvature. They represent the Hessian as a Fisher Information Matrix,

$$\boldsymbol{F} = \mathbb{E}\left[ \left( \frac{\partial \log p(\boldsymbol{y}|\boldsymbol{x}, \boldsymbol{W})}{\partial \boldsymbol{W}} \right) \left( \frac{\partial \log p(\boldsymbol{y}|\boldsymbol{x}, \boldsymbol{W})}{\partial \boldsymbol{W}} \right)^\top \right], \tag{14}$$

where they factorize each layer's block into the Kronecker product of two much smaller matrices $\boldsymbol{\Lambda}_{l-1,m-1}^L \otimes \boldsymbol{\Lambda}_{l,m}^R$. We define the input as $\boldsymbol{a}_0 = \boldsymbol{x}$ which is passed through $1, \ldots, L$ layers, this leads to an output $\boldsymbol{h}_L$. The pre-activations are defined as $\boldsymbol{s}_l = \boldsymbol{W}_l \boldsymbol{a}_{l-1} \in \mathbb{R}^{D_l}$ and the activations as $\boldsymbol{a}_l = f_l(\boldsymbol{s}_l) \in \mathbb{R}^{D_{l-1}}$. The Hessian blocks of the layers $l, m$ can be written as

$$\boldsymbol{\Sigma}_{l,m}^{-1} = \mathbb{E}\left[ \left( \frac{\partial \log p(\boldsymbol{y}|\boldsymbol{x}, \boldsymbol{W}_l)}{\partial \boldsymbol{W}_l} \right) \left( \frac{\partial \log p(\boldsymbol{y}|\boldsymbol{x}, \boldsymbol{W}_m)}{\partial \boldsymbol{W}_m} \right)^\top \right], \tag{15}$$

with $\boldsymbol{W}_l \in \mathbb{R}^{(D_{\text{in}} \times D_{\text{out}})}$. Utilizing the K-FAC approximation, this simplifies to:

$$\boldsymbol{\Sigma}_{l,m}^{-1} = \boldsymbol{\Lambda}_{l-1,m-1}^L \otimes \boldsymbol{\Lambda}_{l,m}^R, \tag{16}$$

with $\boldsymbol{\Lambda}_{l-1,m-1}^L = \mathbb{E}[\boldsymbol{a}_{l-1}\boldsymbol{a}_{m-1}^\top] \in \mathbb{R}^{D_{l-1} \times D_{m-1}}$ and

$$\boldsymbol{\Lambda}_{l,m}^R = \mathbb{E}[\boldsymbol{g}_l \boldsymbol{g}_m^\top] = \frac{\partial \log p(\boldsymbol{y} \mid \boldsymbol{x}, \boldsymbol{W}) \partial \log p(\boldsymbol{y} \mid \boldsymbol{x}, \boldsymbol{W})}{\partial \boldsymbol{s}_l \partial \boldsymbol{s}_l} \tag{17}$$
$$\in \mathbb{R}^{D_l \times D_m}.$$

## B RELATED WORK

**Continual fine-tuning:** The desirable feature of LLMs is to equip the models with *new* knowledge or skills i.e. continual learning. (Wu et al., 2024) proposes three categories, continual pre-training, instruction tuning, and alignment. They correspond respectively to expanding the models' understanding of language, improving responses to the specific commands, and enforcing that the model is abiding by ethical norms. There has been a lot of work done on continual learning via rehearsal, architecture based methods (Scialom et al., 2022; Wang et al., 2024b; Gururangan et al., 2021; Qin et al., 2022) and more importantly for this work, parameter-based (Zheng et al., 2024; Zhu et al., 2024) or gradient-based (Wang et al., 2023b). Wang et al. (2023a) propose O-LoRA, which mitigates catastrophic forgetting by learning tasks in orthogonal vector subspaces under instruction tuning. In Li et al. (2024), the authors perform low-rank adaptation within flat areas of the model's parameter landscape, using random weight perturbation to locate such regions. Wistuba et al. (2023) use LoRA to train a dedicated expert model for each new incoming dataset. They use a $k$ cluster based method to infer which LoRA module to use for which task. By contrast, LoRAMoE Dou et al. (2024) introduces

a mixture of experts architecture in which multiple low-rank adapters work together, dynamically weighted by a router network.

**Model low-rank decomposition:** Biderman et al. (2024) show, via an SVD analysis, that full fine-tuning leaves the singular-value spectrum largely unchanged. Subsequent work explores low-rank decompositions of the full model to retain as much information, minimizing performance degradation. Fisher-Weighted SVD (FWSVD) Hsu et al. (2022) assigns importance scores via Fischer Information Matrix, they observe that singular values' magnitude doesn't directly correspond to performance drop therefore the smallest values may still be needed. Yuan et al. (2024) propose Activation-aware Singular Value Decomposition (ASVD) that manages activation outliers via scaling the weights accordingly. Wang et al. (2024a) propose direct mapping between singular values and model compression loss by ensuring that each channel is independent of each other.

**Other approaches:** DoRA (Liu et al., 2024) decomposes $W_0$ into magnitude and direction components. The method uses only the directional updates during fine-tuning and scales the magnitude. In contrast to LoRA, GaLore (Zhao et al., 2024a) leverages low-rank structure of *gradients*. The authors project the gradient matrix into a low rank updates, this results in substantial memory cost reduction with regard to full fine-tuning, mainly of optimizer states. They notice a slight improvement on GLUE tasks compared to LoRA.

## C    EXTENDED METHODOLOGY

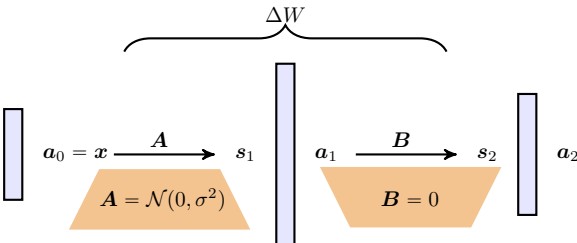

Figure 5: **Illustration of LoRA and its input and outputs. $A$** and **$B$** denote the adapter weigths at initialization, $a$ activations, $s$ pre-activations and $x$ the input to $\Delta W$.

Below, we delve into more details. We present a LoRA module for one layer. The precision for one layer (for a model with $L_m$ layers) can be approximated as:

$$\boldsymbol{\Sigma}_{\Delta W_l}^{-1} = \begin{bmatrix} \boldsymbol{\Lambda}_{0,0}^L \otimes \boldsymbol{\Lambda}_{1,1}^R & \boldsymbol{\Lambda}_{0,1}^L \otimes \boldsymbol{\Lambda}_{1,2}^R \\ \boldsymbol{\Lambda}_{1,0}^L \otimes \boldsymbol{\Lambda}_{2,1}^R & \boldsymbol{\Lambda}_{1,1}^L \otimes \boldsymbol{\Lambda}_{2,2}^R \end{bmatrix}. \tag{18}$$

with shapes

$$\boldsymbol{\Lambda}_{0,0}^L \otimes \boldsymbol{\Lambda}_{1,1}^R \in \mathbb{R}^{(D_{\text{in}} \times D_{\text{in}}) \times (r \times r) = (D_{\text{in}} r \times D_{\text{in}} r)} \tag{19}$$

$$\boldsymbol{\Lambda}_{0,1}^L \otimes \boldsymbol{\Lambda}_{1,2}^R \in \mathbb{R}^{(D_{\text{in}} \times r) \times (r \times D_{\text{out}}) = (D_{\text{in}} r \times r D_{\text{out}})} \tag{20}$$

$$\boldsymbol{\Lambda}_{1,0}^L \otimes \boldsymbol{\Lambda}_{2,1}^R \in \mathbb{R}^{(r \times D_{\text{in}}) \times (D_{\text{out}} \times r) = (D_{\text{out}} r \times r D_{\text{in}})} \tag{21}$$

$$\boldsymbol{\Lambda}_{1,1}^L \otimes \boldsymbol{\Lambda}_{2,2}^R \in \mathbb{R}^{(r \times r) \times (D_{\text{out}} \times D_{\text{out}}) = (r D_{\text{out}} \times r D_{\text{out}})} \tag{22}$$

The Laplace approximation for these low-rank weights can be represented as:

$$p(\Delta \boldsymbol{W}_l | \mathbb{D}_{\text{S}}) \sim \mathcal{N}\left(\boldsymbol{A}, \boldsymbol{B}; \boldsymbol{\mu}, \boldsymbol{\Sigma}_{\Delta \boldsymbol{W}_l}\right) \tag{23}$$

$$p(\Delta \boldsymbol{W}_l | \mathbb{D}_{\text{S}}) \sim \exp\left(-\frac{1}{2} \begin{bmatrix} \text{vec}(\boldsymbol{A} - \boldsymbol{\mu}_{\boldsymbol{A}})^\top & \text{vec}(\boldsymbol{B} - \boldsymbol{\mu}_B)^\top \end{bmatrix} \boldsymbol{\Sigma}_{\Delta W_l}^{-1} \begin{bmatrix} \text{vec}(\boldsymbol{A} - \boldsymbol{\mu}_{\boldsymbol{A}}) \\ \text{vec}(\boldsymbol{B} - \boldsymbol{\mu}_B) \end{bmatrix}\right) \tag{24}$$

$$p(\Delta \boldsymbol{W}_l | \mathbb{D}_{\text{S}}) \sim \exp\left(-\frac{1}{2} \begin{bmatrix} \text{vec}(\boldsymbol{A} - \boldsymbol{\mu}_{\boldsymbol{A}})^\top & \text{vec}(\boldsymbol{B} - \boldsymbol{\mu}_B)^\top \end{bmatrix}\right. \tag{25}$$

$$\left. \begin{bmatrix} \boldsymbol{\Lambda}_{0,0}^L \otimes \boldsymbol{\Lambda}_{1,1}^R & \boldsymbol{\Lambda}_{0,1}^L \otimes \boldsymbol{\Lambda}_{1,2}^R \\ \boldsymbol{\Lambda}_{1,0}^L \otimes \boldsymbol{\Lambda}_{2,1}^R & \boldsymbol{\Lambda}_{1,1}^L \otimes \boldsymbol{\Lambda}_{2,2}^R \end{bmatrix} \begin{bmatrix} \text{vec}(\boldsymbol{A} - \boldsymbol{\mu}_{\boldsymbol{A}}) \\ \text{vec}(\boldsymbol{B} - \boldsymbol{\mu}_B) \end{bmatrix}\right)$$

We take into account $(\boldsymbol{Q} \otimes \boldsymbol{U})\mathrm{vec}\boldsymbol{V} = \mathrm{vec}(\boldsymbol{U}\boldsymbol{V}\boldsymbol{Q}^\top)$, which leads to $(\boldsymbol{\Lambda}_{l-1,m-1}^L \otimes \boldsymbol{\Lambda}_{l,m}^R)\mathrm{vec}(\boldsymbol{W}_{l\to m} - \boldsymbol{W}^*) = \mathrm{vec}(\boldsymbol{\Lambda}_{l,m}^R(\boldsymbol{W}_{l\to m} - \boldsymbol{W}^*)\boldsymbol{\Lambda}_{l-1,m-1}^L)^\top$. We utilize it for the regularizer below

$$\begin{bmatrix} \mathrm{vec}(\boldsymbol{A} - \boldsymbol{\mu_A})^\top \mathrm{vec}(\boldsymbol{B} - \boldsymbol{\mu_B})^\top \end{bmatrix} \begin{bmatrix} \boldsymbol{\Lambda}_{0,0}^L \otimes \boldsymbol{\Lambda}_{1,1}^R & \boldsymbol{\Lambda}_{0,1}^L \otimes \boldsymbol{\Lambda}_{1,2}^R \\ \boldsymbol{\Lambda}_{1,0}^L \otimes \boldsymbol{\Lambda}_{2,1}^R & \boldsymbol{\Lambda}_{1,1}^L \otimes \boldsymbol{\Lambda}_{2,2}^R \end{bmatrix} \begin{bmatrix} \mathrm{vec}(\boldsymbol{A} - \boldsymbol{\mu_A}) \\ \mathrm{vec}(\boldsymbol{B} - \boldsymbol{\mu_B}) \end{bmatrix},$$

where $\mathrm{vec}(\boldsymbol{A} - \boldsymbol{\mu_A}) \in \mathbb{R}^{(D_{\mathrm{in}}r \times 1)}$ and $\mathrm{vec}(\boldsymbol{B} - \boldsymbol{\mu_B}) \in \mathbb{R}^{(rD_{\mathrm{out}} \times 1)}$. This leads to the following result:

$$\begin{bmatrix} \mathrm{vec}(\boldsymbol{A} - \boldsymbol{\mu_A})^\top & \mathrm{vec}(\boldsymbol{B} - \boldsymbol{\mu_B})^\top \end{bmatrix} \begin{bmatrix} (\boldsymbol{\Lambda}_{0,0}^L \otimes \boldsymbol{\Lambda}_{1,1}^R)\mathrm{vec}(\boldsymbol{A} - \boldsymbol{\mu_A}) + (\boldsymbol{\Lambda}_{0,1}^L \otimes \boldsymbol{\Lambda}_{1,2}^R)\mathrm{vec}(\boldsymbol{B} - \boldsymbol{\mu_B}) \\ (\boldsymbol{\Lambda}_{1,0}^L \otimes \boldsymbol{\Lambda}_{2,1}^R)\mathrm{vec}(\boldsymbol{A} - \boldsymbol{\mu_A}) + (\boldsymbol{\Lambda}_{1,1}^L \otimes \boldsymbol{\Lambda}_{2,2}^R)\mathrm{vec}(\boldsymbol{B} - \boldsymbol{\mu_B}) \end{bmatrix} =$$

$$= \mathrm{vec}(\boldsymbol{A} - \boldsymbol{\mu_A})^\top \mathrm{vec}\big(\boldsymbol{\Lambda}_{1,1}^R(\boldsymbol{A} - \boldsymbol{\mu_A})(\boldsymbol{\Lambda}_{0,0}^L)^\top\big)$$

$$(1 \times D_{\mathrm{in}}r) \cdot ((r \times r)\,(r \times D_{\mathrm{in}})\,(D_{\mathrm{in}} \times D_{\mathrm{in}}))$$

$$+ \mathrm{vec}(\boldsymbol{B} - \boldsymbol{\mu_B})^\top \mathrm{vec}\big(\boldsymbol{\Lambda}_{2,1}^R(\boldsymbol{A} - \boldsymbol{\mu_A})(\boldsymbol{\Lambda}_{1,0}^L)^\top\big)$$

$$(1 \times D_{\mathrm{out}}r) \cdot ((D_{\mathrm{out}} \times r)\,(r \times D_{\mathrm{in}})\,(D_{\mathrm{in}} \times r))$$

$$+ \mathrm{vec}(\boldsymbol{A} - \boldsymbol{\mu_A})^\top \mathrm{vec}\big(\boldsymbol{\Lambda}_{1,2}^R(\boldsymbol{B} - \boldsymbol{\mu_B})(\boldsymbol{\Lambda}_{0,1}^L)^\top\big)$$

$$(1 \times D_{\mathrm{in}}r) \cdot ((r \times D_{\mathrm{out}})\,(D_{\mathrm{out}} \times r)\,(r \times D_{\mathrm{in}}))$$

$$+ \mathrm{vec}(\boldsymbol{B} - \boldsymbol{\mu_B})^\top \mathrm{vec}\big(\boldsymbol{\Lambda}_{2,2}^R(\boldsymbol{B} - \boldsymbol{\mu_B})(\boldsymbol{\Lambda}_{1,1}^L)^\top\big)$$

$$(1 \times D_{\mathrm{out}}r) \cdot ((D_{\mathrm{out}} \times D_{\mathrm{out}})\,(D_{\mathrm{out}} \times r)\,(r \times r))$$

The cost of this operation is:

- compute: $\mathcal{O}(D_{\mathrm{in}}^2 r + D_{\mathrm{out}}^2 r + r^2(D_{\mathrm{in}} + D_{\mathrm{out}}))$

- memory: $\mathcal{O}(D_{\mathrm{in}}^2 + D_{\mathrm{out}}^2 + r^2)$

We can simplify the cross layer terms since $A_{ji} = (\boldsymbol{A}_{ij})^\top$.

$$\mathrm{vec}(\boldsymbol{A} - \boldsymbol{\mu_A})^\top(\boldsymbol{\Lambda}_{0,1}^L \otimes \boldsymbol{\Lambda}_{1,2}^R)\mathrm{vec}(\boldsymbol{B} - \boldsymbol{\mu_B}) = \tag{26}$$

$$= \mathrm{vec}(\boldsymbol{A} - \boldsymbol{\mu_A})^\top((\boldsymbol{\Lambda}_{1,0}^L)^\top \otimes (\boldsymbol{\Lambda}_{2,1}^R)^\top)\mathrm{vec}(\boldsymbol{B} - \boldsymbol{\mu_B}) =$$

$$= \mathrm{vec}(\boldsymbol{A} - \boldsymbol{\mu_A})^\top(\boldsymbol{\Lambda}_{1,0}^L \otimes \boldsymbol{\Lambda}_{2,1}^R)^\top\mathrm{vec}(\boldsymbol{B} - \boldsymbol{\mu_B}) =$$

$$= \big(\mathrm{vec}(\boldsymbol{A} - \boldsymbol{\mu_A})(\boldsymbol{\Lambda}_{1,0}^L \otimes \boldsymbol{\Lambda}_{2,1}^R)\mathrm{vec}(\boldsymbol{B} - \boldsymbol{\mu_B})^\top\big)^\top =$$

$$= \big(\mathrm{vec}(\boldsymbol{B} - \boldsymbol{\mu_B})^\top(\boldsymbol{\Lambda}_{1,0}^L \otimes \boldsymbol{\Lambda}_{2,1}^R)\mathrm{vec}(\boldsymbol{A} - \boldsymbol{\mu_A})\big)^\top =$$

$$= \mathrm{vec}(\boldsymbol{B} - \boldsymbol{\mu_B})^\top(\boldsymbol{\Lambda}_{1,0}^L \otimes \boldsymbol{\Lambda}_{2,1}^R)\mathrm{vec}(\boldsymbol{A} - \boldsymbol{\mu_A})$$

The number of stored blocks is $3L_m$.

## C.1 IMPACT OF THE ADAPTERS' CURVATURE APPROXIMATION ON THE CORRECTION TERM

DIAG: Note that if we consider the LoRA update $\Delta \boldsymbol{W} = \boldsymbol{B}\boldsymbol{A}$, then the covariance between $\Delta \boldsymbol{W}_{ij}$ and $\Delta \boldsymbol{W}_{k\ell}$ is

$$\mathrm{cov}\left(\sum_\alpha \boldsymbol{B}_{i\alpha}\boldsymbol{A}_{\alpha j}, \sum_\beta \boldsymbol{B}_{k\beta}\boldsymbol{A}_{\beta\ell}\right) = \sum_{\alpha,\beta} \mathrm{cov}(\boldsymbol{B}_{i\alpha}\boldsymbol{B}_{k\beta}) \cdot \mathrm{cov}(\boldsymbol{A}_{\alpha j}\boldsymbol{A}_{\beta\ell})$$

$$= \sum_{\alpha,\beta} \delta_{ik}\delta_{\alpha\beta}\delta_{j\ell}[\boldsymbol{D_B}]_{i\alpha}[\boldsymbol{D_A}]_{\alpha j}$$

$$= \sum_\alpha \delta_{ik}\delta_{jl}[\boldsymbol{D_B}]_{i\alpha}[\boldsymbol{D_A}]_{\alpha j}$$

$$= [\boldsymbol{D_B}\boldsymbol{D_A}^T]_{ij}\delta_{ik}\delta_{j\ell}$$

Table 2: **Target accuracy at common optimizer-step budget** $B^\star$. $B^\star$ is the chosen budget across configurations (here 10000). The baseline is the configuration with the highest target at $B^\star$. Batch size: bs, gradient accumulation: ga, effective batch size: ebs, learning rate: lr. We report the validation mean target accuracy across 3 seeds.

| Configuration | $B^\star=10000$ |
|---|---|
| bs: 12, ga: 4, ebs: 48, lr: 0.001 | 0.211 |
| bs: 12, ga: 2, ebs: 24, lr: 0.0005 | **0.247** |
| bs: 12, ga: 1, ebs: 12, lr: 0.00025 | 0.221 |

This is a diagonal Laplace approximation on the adapters that consists of an outer product of the diagonal terms.

B-KFAC: This yields, from Equation (8):

$$\text{cov}\left(\sum_\alpha \boldsymbol{B}_{i\alpha}\boldsymbol{A}_{\alpha j}, \sum_\beta \boldsymbol{B}_{k\beta}\boldsymbol{A}_{\beta\ell}\right) = \sum_{\alpha,\beta} \text{cov}\left(\boldsymbol{B}_{i\alpha}\boldsymbol{B}_{k\beta}\right) \cdot \text{cov}\left(\boldsymbol{A}_{\alpha_j}\boldsymbol{A}_{\beta\ell}\right) = \sum_{\alpha,\beta} \Lambda_{ik}^{LB}\Lambda_{\alpha\beta}^{RB}\Lambda_{j\ell}^{LA}\Lambda_{\alpha\beta}^{RA}$$

$$= (\Lambda^{LB}\otimes\Lambda^{LA})ij,k\ell \cdot \left(\sum_{\alpha,\beta}(\Lambda^{RB}\odot\Lambda^{RB})_{\alpha\beta}\right)$$

This is a KFAC on the adapters.

B-Tri-KFAC: Based on Equation (10), this leads to:

$$\text{cov}\left(\sum_\alpha \boldsymbol{B}_{i\alpha}\boldsymbol{A}_{\alpha j}, \sum_\beta \boldsymbol{B}_{k\beta}\boldsymbol{A}_{\beta\ell}\right) = \sum_{\alpha,\beta} \text{cov}(\boldsymbol{B}_{i\alpha}\boldsymbol{B}_{k\beta})\cdot\text{cov}(\boldsymbol{A}_{\alpha_j}\boldsymbol{A}_{\beta\ell}) + \text{cov}(\boldsymbol{B}_{i\alpha}\boldsymbol{A}_{\beta\ell})$$

$$+ \text{cov}(\boldsymbol{A}_{\alpha j}\boldsymbol{B}_{k\beta})$$

$$= \sum_{\alpha,\beta} \Lambda_{ik}^{LBB}\Lambda_{\alpha\beta}^{RBB}\Lambda_{j\ell}^{LAA}\Lambda_{\alpha\beta}^{RAA} + (\Lambda_{i\ell}^{LBA} + \Lambda_{jk}^{LBA})\Lambda_{\alpha\beta}^{RBA}$$

$$= (\Lambda^{LB}\otimes\Lambda^{LA})ij,k\ell \cdot \left(\sum_{\alpha,\beta}(\Lambda^{RB}\odot\Lambda^{RB})_{\alpha\beta}\right) + (\Lambda_{i\ell}^{LBA}$$

$$+ \Lambda_{jk}^{LBA})\sum_{\alpha\beta}\Lambda_{\alpha\beta}^{RBA}$$

## D EXPERIMENTAL SETUP

We follow LoRA setups found by Biderman et al. (2024) and others (Raschka, 2023; Dettmers et al., 2023). We target $\boldsymbol{W}_q, \boldsymbol{W}_k$ in attention modules. We use $\alpha = 2r$ and rank 16 and high learning rates (maximal and stable from $1e^{-5} - 5e^{-4}$). Biderman et al. (2024) states the memory and optimizer states needed for 3B model. Following their analysis, we opt for one A100. We find the baseline setting looking at different batch size, lr and gradient accumulation settings, described below.

**Fixed step budgets.** Epochs are not comparable because updates/epoch vary with batch × accumulation. We therefore evaluate all configurations at common optimizer-step budgets $B$. All configurations are tested for a chosen number of optimizer steps $B^\star = 10000$. The results are reported in Table 2. The baseline is the configuration with the highest target domain accuracy at $B^\star$ i.e. batch size 12, gradient accumulation 2 and learning rate 0.0005.

Other training arguments are shown in the Table 3.

The optimal strength of regularization $\lambda$, showcased in Table 1, as well as, experiments in Section 5.2, was chosen via runs with validation dataset split. The probed values of $\lambda$ were $\{1, 10, 10^2, 10^3, 10^4, 10^5, 10^6\}$. The results for each value are shown in Table 4. We focus on two settings that cover important scenarios. We either maximize source accuracy ($\lambda_{\text{stability}}$) or compute a weighted average prioritizing target accuracy ($\lambda_{\text{plasticity}}$). The weighted average is computed

Table 3: **Experimental setup**. The table presents the values for the arguments used in the experiments.

| Argument | |
| --- | --- |
| dataset name | GSM-8K |
| per device train batch size | 12 |
| per device evaluate batch size | 64 |
| per device LA batch size | 4 |
| curvature/method | DIAG, B-KFAC |
| learning rate | 5e-4 with a schedule |
| number of epochs | 15 |
| sequence length | 512 |
| causal generation length | 128 |
| seeds | 42 |
| evaluation frequency | every 5 epochs or every epoch |
| LoRA rank | 16 |
| LoRA $\alpha$ | 32 |
| LoRA dropout | 0.1 |

Table 4: **Validation metrics**. Accuracy values and learning prioritized score for all methods across one seed.

| | $\lambda = 0$ | $\lambda = 1$ | $\lambda = 10$ | $\lambda = 10^2$ | $\lambda = 10^3$ | $\lambda = 10^4$ | $\lambda = 10^5$ | $\lambda = 10^6$ |
| --- | --- | --- | --- | --- | --- | --- | --- | --- |
| DIAG | F: 0.618
L: 0.250
$S_B$: 0.44 | F: 0.616
L: 0.265
$S_B$: 0.59 | F: 0.635
L: 0.276
$S_B$: **0.83** | F: 0.646
L: 0.233
$S_B$: 0.47 | F: **0.658**
L: 0.232
$S_B$: 0.54 | F: 0.647
L: 0.226
$S_B$: 0.39 | F: 0.653
L: 0.209
$S_B$: 0.26 | F: **0.658**
L: 0.233
$S_B$: 0.55 |
| | $\lambda = 0$ | $\lambda = 0.05$ | $\lambda = 0.10$ | $\lambda = 0.15$ | $\lambda = 0.20$ | $\lambda = 0.25$ | $\lambda = 0.30$ | $\lambda = 0.35$ |
| B-KFAC | | F: 0.629
L: 0.238
$S_B$: 0.45 | F: 0.636
L: 0.232
$S_B$: 0.36 | F: 0.639
L: 0.246
$S_B$: **0.85** | F: 0.643
L: 0.235
$S_B$: 0.57 | F: **0.649**
L: 0.224
$S_B$: 0.30 | F: 0.644
L: 0.236
$S_B$: 0.63 | F: 0.644
L: 0.233
$S_B$: 0.52 |
| | $t = 0$ | $t = 0.5$ | $t = 0.6$ | $t = 0.7$ | $t = 0.75$ | $t = 0.8$ | $t = 0.85$ | $t = 0.9$ |
| MIGU | | F: 0.629
L: 0.220
$S_B$: 0.06 | F: 0.642
L: 0.226
$S_B$: 0.31 | F: **0.650**
L: 0.255
$S_B$: 0.76 | F: 0.634
L: 0.236
$S_B$: 0.32 | F: 0.643
L: 0.252
$S_B$: 0.62 | F: 0.629
L: 0.215
$S_B$: 0.00 | F: 0.649
L: 0.276
$S_B$: **0.98** |
| | $\lambda = 0$ | $\lambda = 10^{-7}$ | $\lambda = 10^{-6}$ | $\lambda = 10^{-5}$ | $\lambda = 5e10^{-4}$ | $\lambda = 10^{-3}$ | $\lambda = 10^{-2}$ | $\lambda = 10^{-1}$ |
| $L^2$ | | F: 0.627
L: 0.279
$S_B$: 0.65 | F: 0.626
L: 0.270
$S_B$: 0.58 | F: 0.629
L: 0.250
$S_B$: 0.47 | F: 0.649
L: 0.286
$S_B$: **0.89** | F: 0.655
L: 0.217
$S_B$: 0.43 | F: 0.659
L: 0.202
$S_B$: 0.36 | F: **0.663**
L: 0.189
$S_B$: 0.30 |

as follows–learning accuracy $L$ and final accuracy $F$ are combined into a normalized, learning-prioritized score:

$$L' = \frac{L - \min L}{\max L - \min L}, \quad F' = \frac{F - \min F}{\max F - \min F}, \qquad S_B(\alpha) = \alpha\,L' + (1 - \alpha)\,F',$$

with $\alpha = 0.7$ to emphasize learning. Therefore, for $\lambda_{\text{plasticity}}$, we select $\lambda$ that maximizes $S_B$ on validation.

The same procedure is applied to all methods. The selected values (in bold) are shown in Table 4. The resulting hyperparameter values, $\lambda_{\text{stability}}, t_{\text{stability}}$ (forgetting) and $\lambda_{\text{plasticity}}, t_{\text{plasticity}}$ (learning), are reported in Table 5.

# E  MORE EXPERIMENTAL RESULTS

In this section we present the additional results in Figures 6 to 10, 12 to 14, 18, 20 and 22 and Table 6.

Table 5: **Best stability vs. plasticity settings**. Stability ranked by highest $S_B$; plasticity by highest $F$.

| Method | Stability (least forgetting) | Plasticity (best learning) |
|---|---|---|
| DIAG | $\lambda_{\text{stability}} = 10^3$, (tied with $10^6$) | $\lambda_{\text{plasticity}} = 10$ |
| B-KFAC | $\lambda_{\text{stability}} = 0.25$ | $\lambda_{\text{plasticity}} = 0.15$ |
| MIGU | $t_{\text{stability}} = 0.7$ | $t_{\text{plasticity}} = 0.9$ |
| L2 | $\lambda_{\text{stability}} = 10^{-1}$ | $\lambda_{\text{plasticity}} = 5e10^{-4}$ |

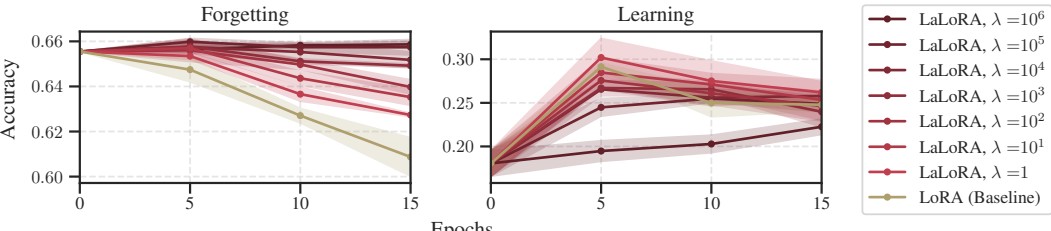

Figure 6: **A stronger diagonal regularizer mitigates source domain forgetting more.** The figure shows accuracy over the course of fine-tuning: (*left*) average source domain accuracy (forgetting), (*right*) target dataset accuracy. Each setting corresponds to a single random seed. As the regularization strength $\lambda$ increases, forgetting declines, but the model's ability to learn the new task is reduced.

## F  COMPUTATIONAL COSTS

We study the time and memory constraints of the proposed method. As stated by Biderman et al. (2024), for LoRA with Adam, storing the weights of the model in fp32 requires 4 bytes per parameter. Storing the gradient requires 4 bytes per trainable parameter, and storing the optimizer state for Adam requires 8 bytes per trainable parameter. In our case, the number of parameters is equal to $\Psi = 3B$, and for rank 32 the number of trainable parameters is around $0.3\%$. We need at least $4 \times \Psi(1 + 0.003) + 4 \times \Psi \times 0.003 + 8 \times \Psi \times 0.003 = 12.12$GB (in comparison to 48 GB for full fine-tuning). We can further reduce it to $2 \times \Psi + 16 \times \Psi \times 0.003 = 6.144$GB by using fp16 for the non-tuned weights. The above description considers only the model states, i.e., we don't look at residual states such as activations and temporary buffers for intermediate quantities. They depend on batch size and maximum sequence.

The training fitted on one A100 with 40 GB, with a per-device batch size of 12 and gradient accumulation steps 2, resulted and lasted around 3-4 hours for 15 epochs (with several evaluations lasting 8 minutes).

Tables 9 and 10 show the measured wall-clock runtime and GPU memory of computing the Laplace approximation and fine-tuning step, respectively. The overhead for computing the regularizer is negligible (additional 0.006s and 40 MB). Tables 11 and 12 show compute and memory requirements for LaLoRA and other baselines, and for different curvature approximations.

USE OF LLMS: A large language model was used to polish the manuscript, plotting settings and functions. The methodology, analyses, dataset choices, and parameter settings were determined by the authors. All LLM-generated text and code were reviewed and edited prior to inclusion.

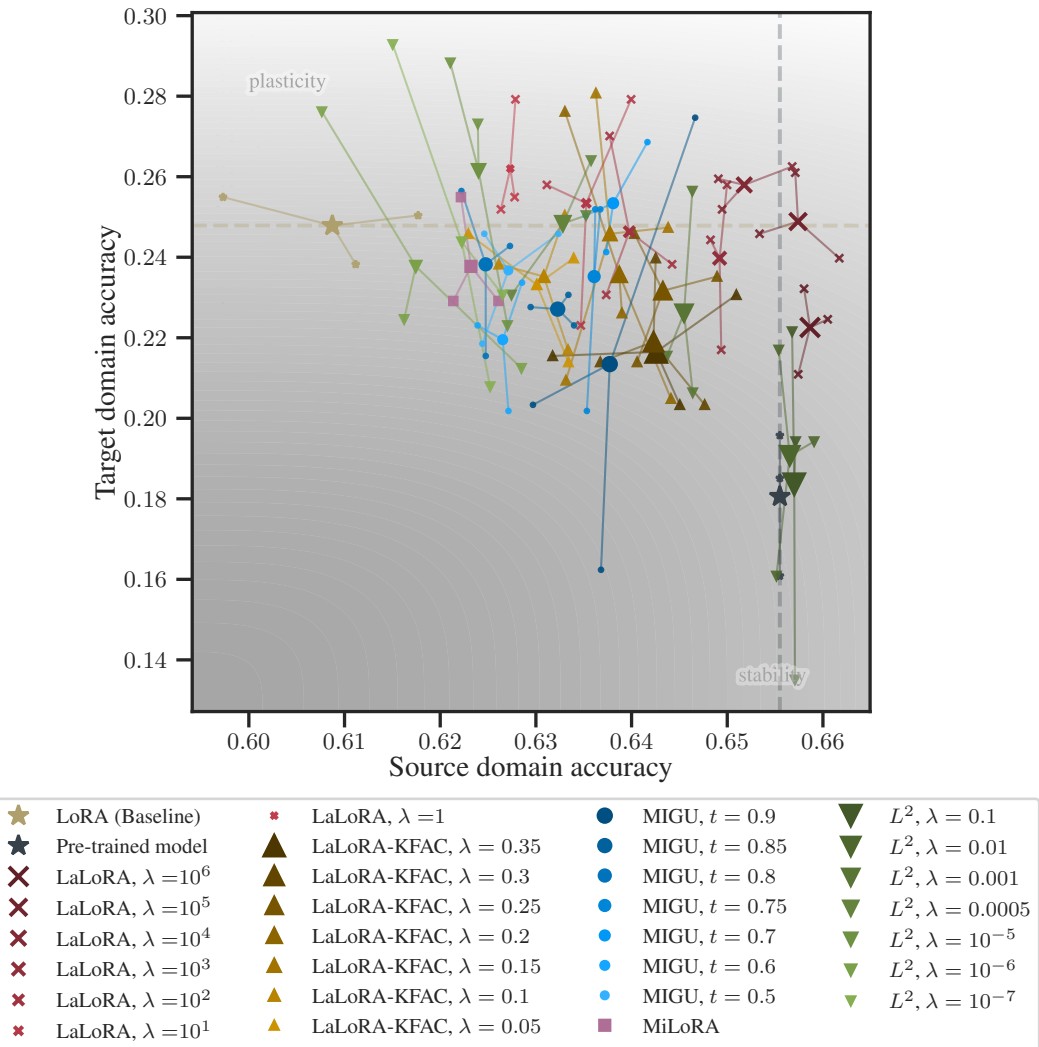

Figure 7: **Laplace regularization leads to improved learning-forgetting trade-off.** The figure shows on $x$-axis average source domain accuracy (forgetting) and on $y$-axis target dataset accuracy (learning). The smaller markers are individual runs and the marker they are all connected to is the mean. The final epoch accuracy is plotted for different values of hyperparameters and methods.

Table 6: **Change of weights corresponding to the precision values.** The table showcases the parameters change grouped by their corresponding precision. The columns list the number of parameters in each group, mean, minimum and maximum precision as well as the mean difference for LaLoRA and LoRA (Baseline).

| Group | $n$ | $\bar{\Sigma}^{-1}_{\text{mean}}$ | $\bar{\Sigma}^{-1}_{\text{min}}$ | $\bar{\Sigma}^{-1}_{\text{max}}$ | $|\Delta_{\text{LaLoRA}}|$ | $|\Delta_{\text{LoRA}}|$ |
|---|---|---|---|---|---|---|
| Flexible | 2,752,512 | 0 | 0 | 0 | 0.0249 | 0.0197 |
| Middle | 1,376,256 | $1.06 \times 10^{-5}$ | $7.61 \times 10^{-22}$ | $7.24 \times 10^{-5}$ | $9.00 \times 10^{-4}$ | 0.0201 |
| Important | 458,752 | $1.52 \times 10^{-3}$ | $7.24 \times 10^{-5}$ | 0.402 | $1.32 \times 10^{-4}$ | 0.0145 |

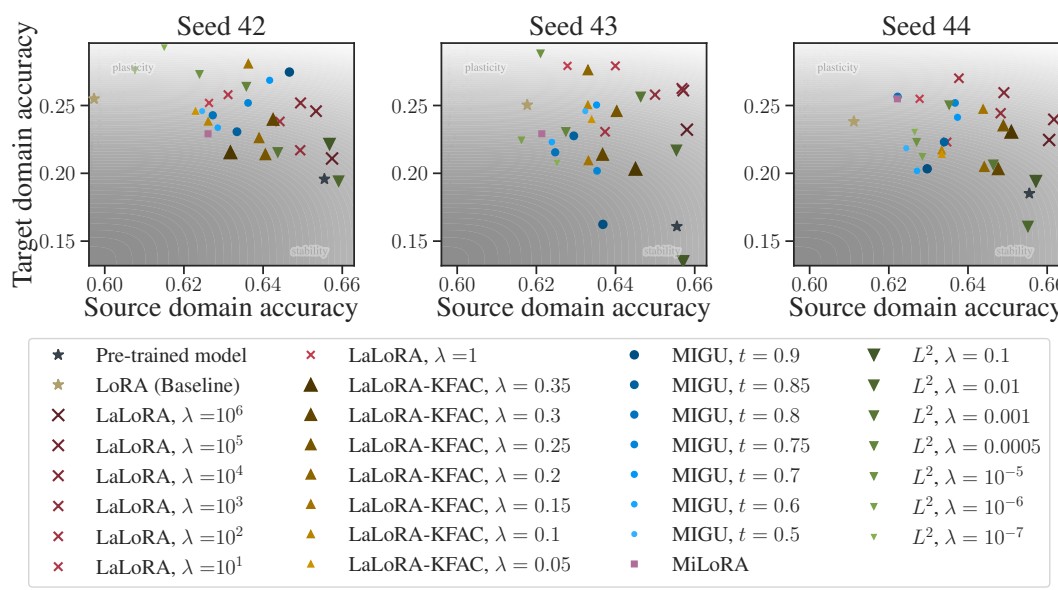

**Figure 8: Laplace regularization leads to improved learning-forgetting trade-off, across each three seeds.** Final source ($x$-axis) vs. target ($y$-axis) accuracy for LaLoRA and competing methods, for each seed. The marker size & brightness indicates the value of each method's hyperparameter controlling the learning–forgetting trade-off.

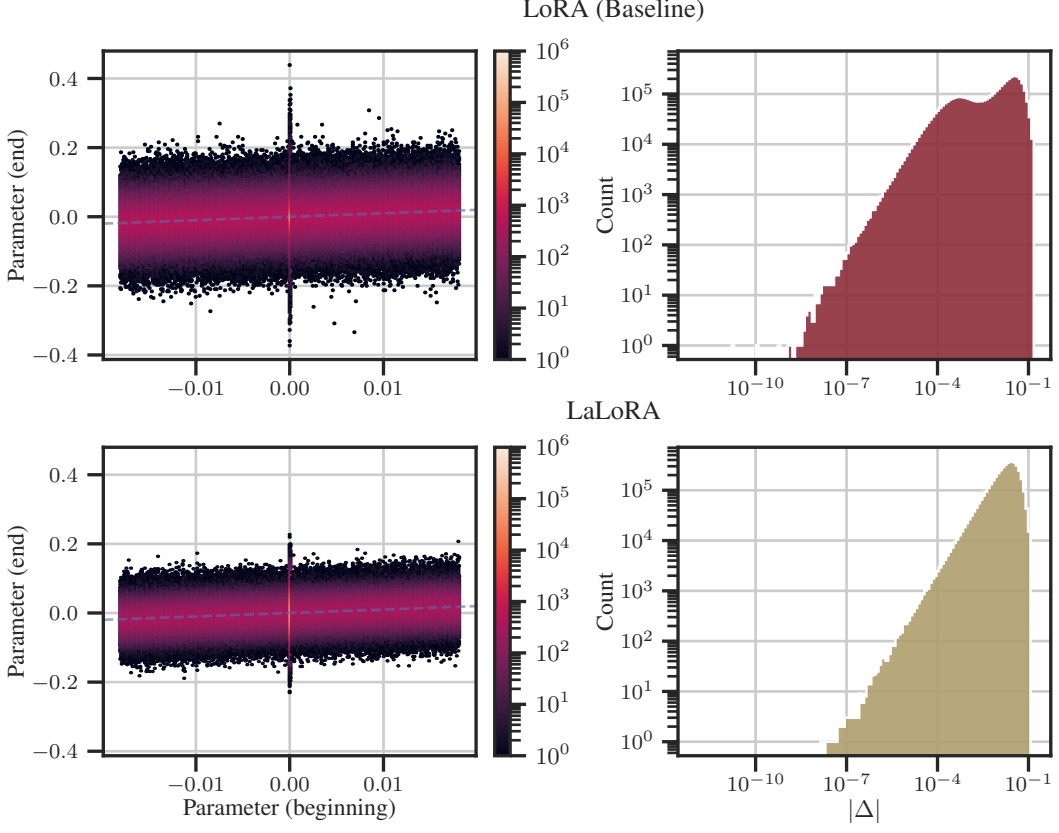

**Figure 9: Most weights barely change, a small fraction move a lot.** Weight changes for the fine-tuning. (*Left*) Joint density of parameters at the beginning ($x$) vs end ($y$) of training. (*Right*) Histogram of differences on log–log axes. (*Top*) The results for the baseline and (*bottom*) for LaLoRA

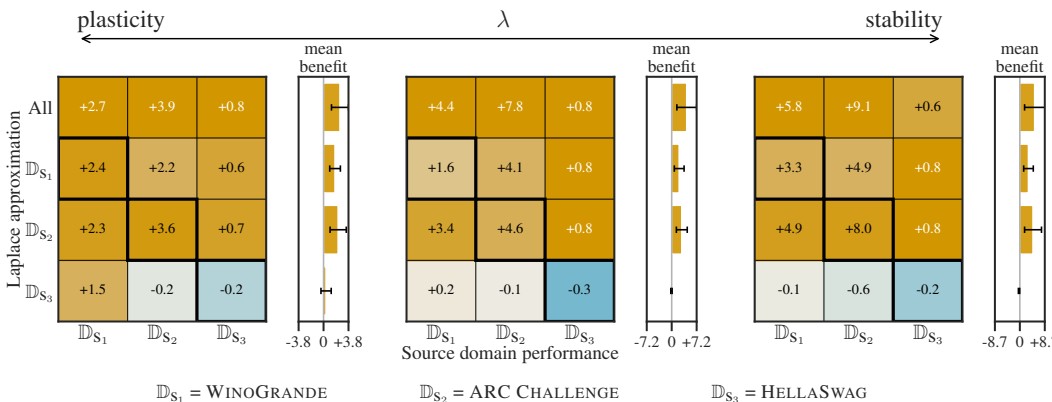

Figure 10: **Using more datasets to represent source domain, leads to less forgetting**. The figure shows the influence of the datasets used for LA ($y$-axis) and their forgetting rate ($x$-axis). The bold squares show the case when we use dataset $\mathbb{D}_{S_i}$ for the LA and evaluate on this dataset $\mathbb{D}_{S_i}$. The mean benefit (*right*) shows the averaged forgetting rate across three seeds. The heatmaps are presented for three strengths of regularization $\lambda = \{10, 10^3, 10^6\}$.

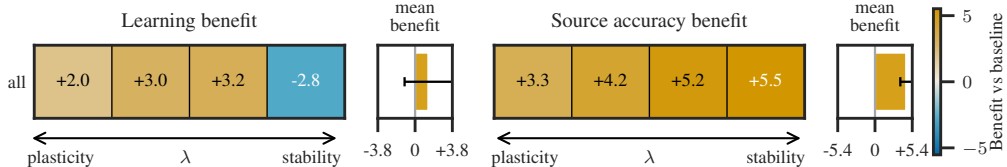

Figure 11: **Using the full sub-datasets for LA leads to better learning capabilities as per one batch, however the forgetting gains are comparable.** We analyze the impact of using one, two, or three batches ($y$-axis) per surrogate dataset for the Laplace approximation. Performance is measured as benefit over baseline (orange is better) for both target-domain learning (*left*) and source-domain accuracy (*right*) across three regularization strengths $\lambda$ ($x$-axis). The mean benefit subplots summarize the results across $\lambda$ and three seeds.

Table 7: **MATH: Validation metrics.** Final forgetting and learning accuracies, and the combined score $S_B$ (with $\alpha = 0.7$) for different values of $\lambda$.

| | $\lambda = 0$ | $\lambda = 1$ | $\lambda = 10$ | $\lambda = 10^2$ | $\lambda = 1.5 \times 10^2$ | $\lambda = 2.5 \times 10^2$ | $\lambda = 10^3$ | $\lambda = 5 \times 10^3$ |
|---|---|---|---|---|---|---|---|---|
| DIAG | F: 0.591 | F: 0.603 | F: 0.619 | F: 0.645 | F: 0.644 | F: 0.650 | F: 0.652 | F: **0.660** |
| | L: 0.088 | L: 0.094 | L: 0.083 | L: 0.091 | L: 0.089 | L: 0.086 | L: 0.080 | L: 0.047 |
| | $S_B$: 0.61 | $S_B$: 0.75 | $S_B$: 0.67 | $S_B$: **0.88** | $S_B$: 0.86 | $S_B$: 0.84 | $S_B$: 0.76 | $S_B$: 0.30 |

Table 8: **Llama-3.1-8B: Validation metrics.** Final forgetting and learning accuracies, and the combined score $S_B$ (with $\alpha = 0.7$) for different values of $\lambda$ (diag_rank_32 + Laplace setup).

| | $\lambda = 0$ | $\lambda = 1$ | $\lambda = 50$ | $\lambda = 10^2$ | $\lambda = 10^3$ | $\lambda = 5 \times 10^3$ | $\lambda = 10^4$ | $\lambda = 5 \times 10^4$ |
|---|---|---|---|---|---|---|---|---|
| DIAG | F: 0.619 | F: 0.650 | F: 0.690 | F: 0.698 | F: 0.710 | F: **0.728** | F: 0.726 | F: 0.726 |
| | L: 0.370 | L: 0.296 | L: 0.400 | L: 0.380 | L: 0.352 | L: 0.385 | L: 0.389 | L: 0.399 |
| | $S_B$: 0.50 | $S_B$: 0.09 | $S_B$: 0.90 | $S_B$: 0.79 | $S_B$: 0.63 | $S_B$: 0.90 | $S_B$: 0.92 | $S_B$: **0.98** |

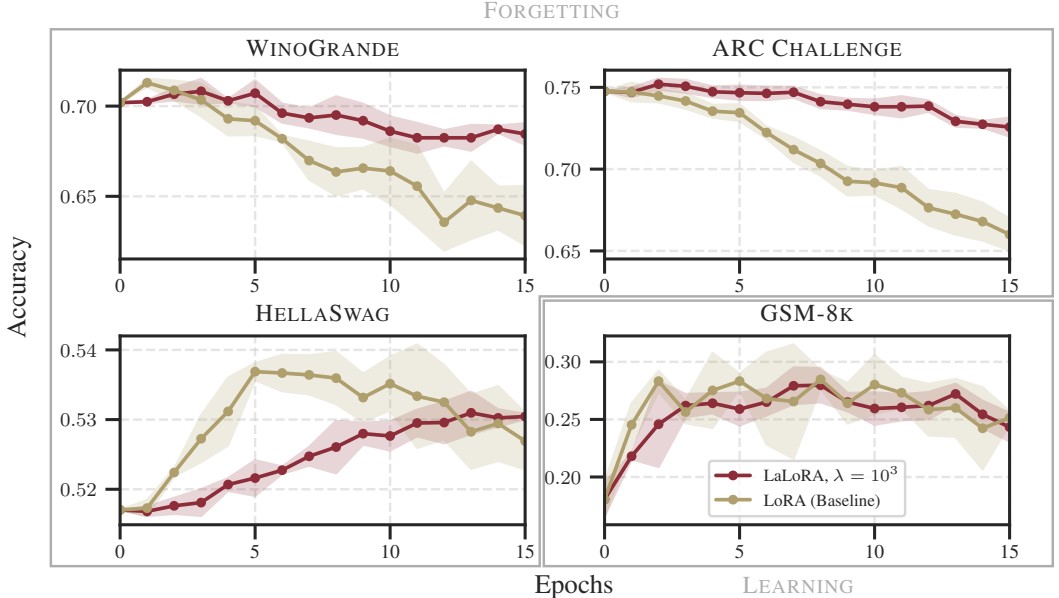

Figure 12: **Diagonal regularization reduces forgetting across all datasets.** The figure shows accuracy over the course of fine-tuning: average source domain accuracy (forgetting) and target dataset accuracy and corresponding standard deviation. Each setting corresponds to three random seeds. Results are displayed for the three source domain datasets: WINOGRANDE, ARC CHALLENGE, and HellaSwag, and, on the bottom-right, for the target math dataset GSM8K. We notice less forgetting for the regularized approach compared to the baseline, and similar learning capabilities.

Table 9: **Laplace summary (per dataset)- time and compute.** We present the mean duration of each step and compute usage.

| Step | Time (s) | GPU Allocated (GB) | GPU Reserved (GB) |
|------|----------|--------------------|--------------------|
| WINOGRANDE fit | 0.693 | 6.05 | 8.04 |
| ARC CHALLENGE fit | 0.837 | 6.09 | 12.57 |
| HELLASWAG fit | 0.835 | 6.12 | 12.64 |

Table 10: **Training summary - time and compute.** We present the mean duration of each step and compute usage.

| Step | Time (s) | GPU Allocated (GB) | GPU Reserved (GB) |
|------|----------|--------------------|--------------------|
| forward | 0.309 | 28.44 | 36.04 |
| regularizer | 0.006 | 28.48 | 36.04 |
| backward | 0.439 | 8.75 | 37.51 |
| optimizer step | 0.058 | 8.74 | 37.51 |

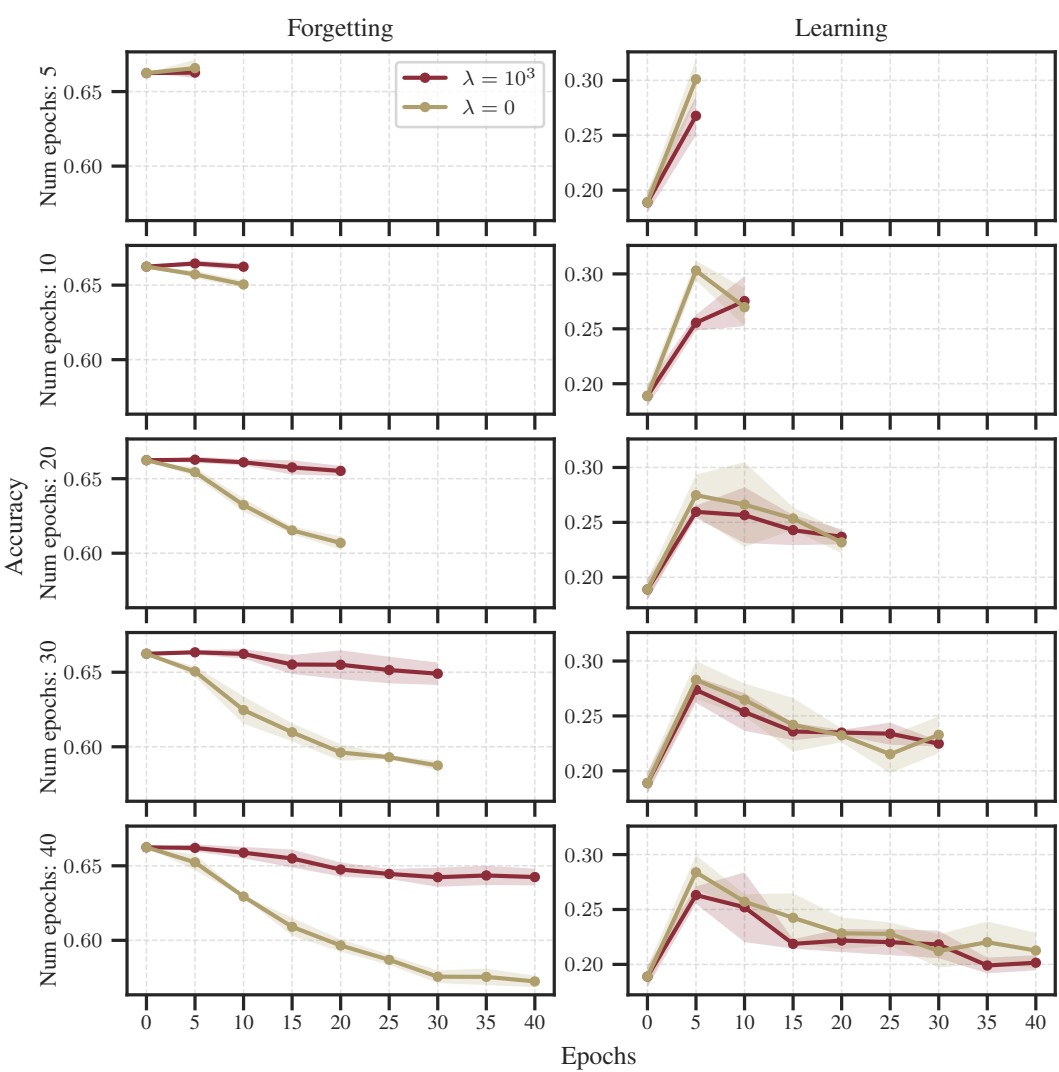

Figure 13: **Learning forgetting tradeoff is visible across different lengths of training**. Accuracy while fine-tuning: (*left*) average source-domain accuracy (forgetting) and (*right*) target-dataset accuracy (learning). We show mean and standard deviation across three random seeds. We present results for $\lambda = 10^3$ and baseline. The lr schedule is dependent on different maximal number of training steps, this leads to different results in different subplots, especially, we notice deviation for training of 5 epochs.

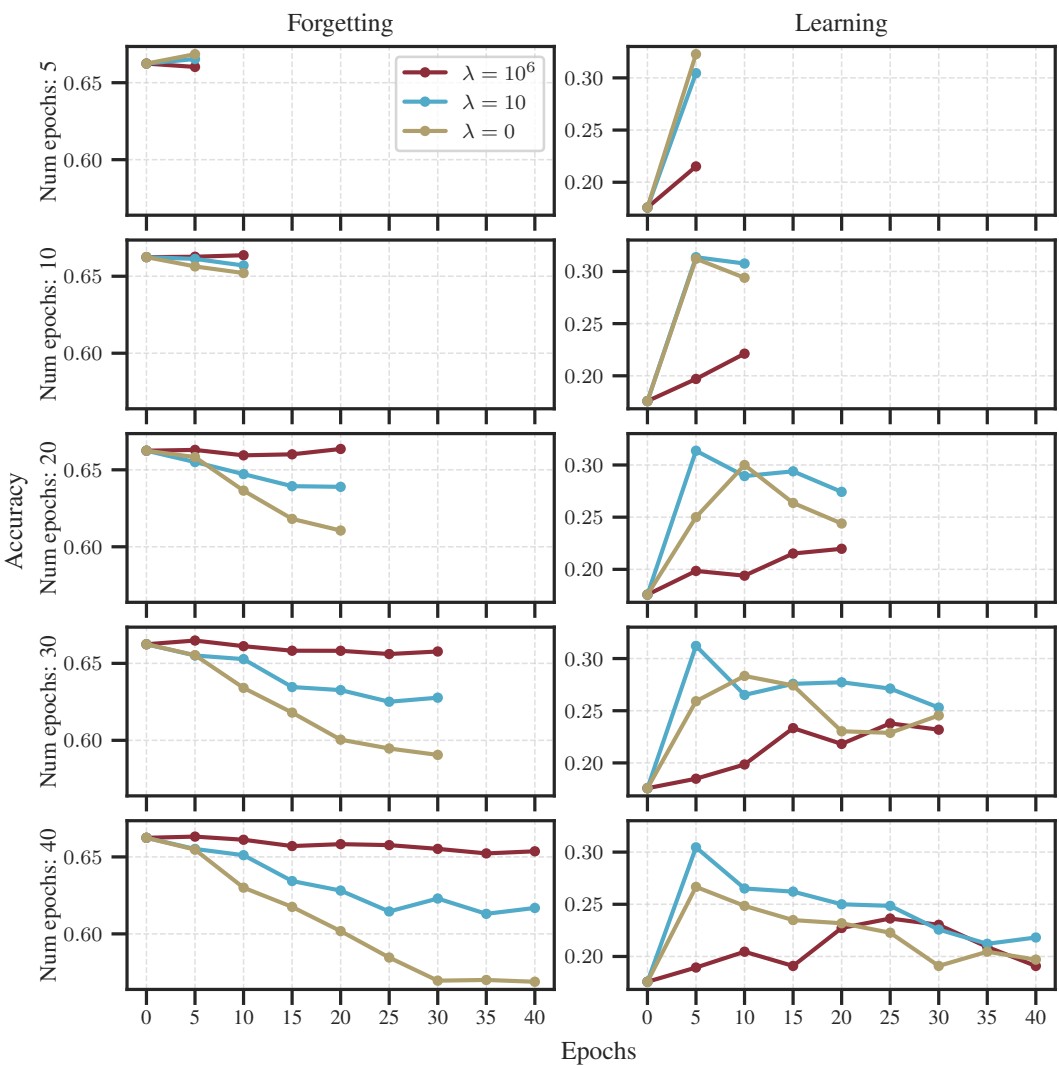

Figure 14: **Learning forgetting tradeoff is visible across different lengths of training**. Accuracy while fine-tuning: (*left*) average source-domain accuracy (forgetting) and (*right*) target-dataset accuracy (learning). We show the results for one seed for $\lambda = 10$, $\lambda = 10^6$ and baseline. The lr schedule is dependent on different maximal number of training steps, this leads to different results in different subplots, especially, we notice deviation for training of 5 epochs.

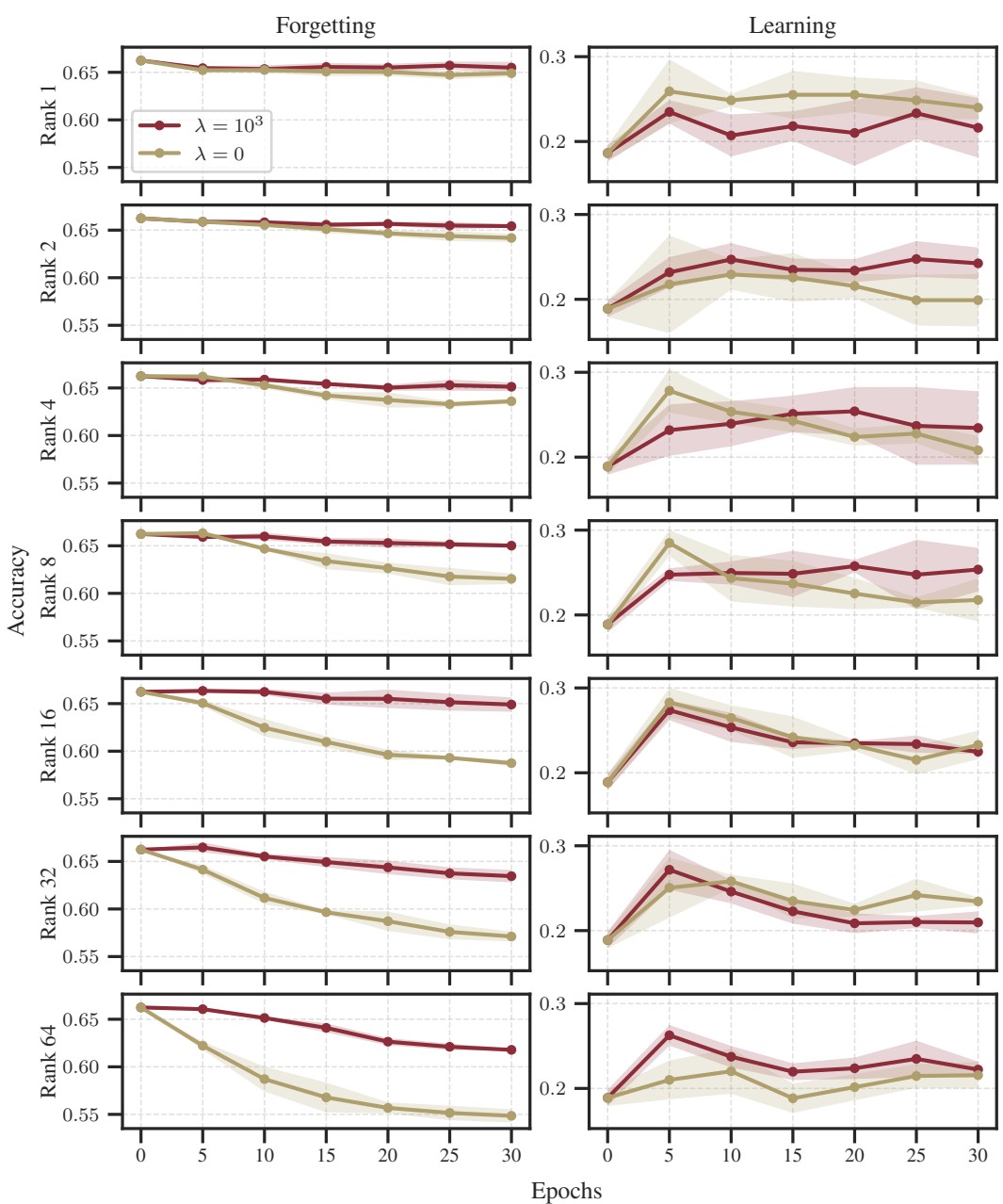

Figure 15: **Learning forgetting tradeoff is visible across different ranks of LoRA adapters**. Accuracy while fine-tuning: (*left*) average source-domain accuracy (forgetting) and (*right*) target-dataset accuracy (learning). We show mean and standard deviation across three random seeds. We present results for $\lambda = 10^3$ and baseline. With bigger rank we notice more forgetting, as expected.

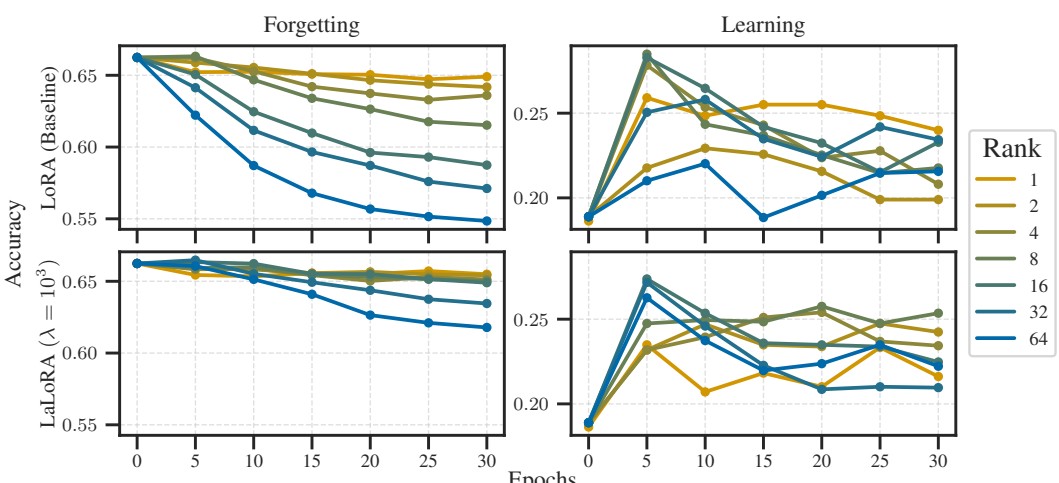

Figure 16: **Learning forgetting tradeoff is visible across different ranks of LoRA adapters**. Accuracy while fine-tuning: (*left*) average source-domain accuracy (forgetting) and (*right*) target-dataset accuracy (learning), (*top*) baseline (*bottom*) LaLoRA with $\lambda = 10^3$. We show mean across three random seeds.

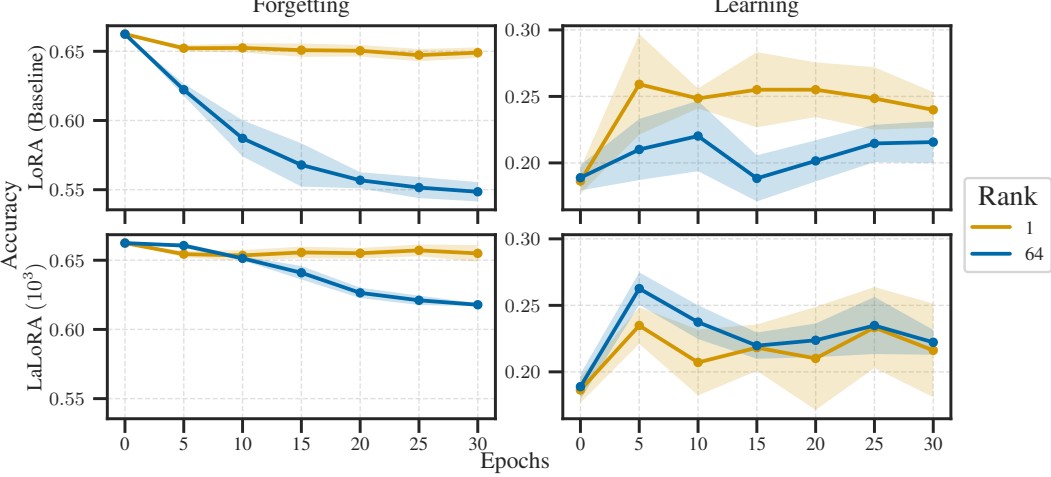

Figure 17: **Rank 1 vs rank 64: learning and forgetting trade-off**. Accuracy while fine-tuning: (*left*) average source-domain accuracy (forgetting) and (*right*) target-dataset accuracy (learning), (*top*) baseline (*bottom*) LaLoRA with $\lambda = 10^3$. We show mean and standard deviation across three random seeds.

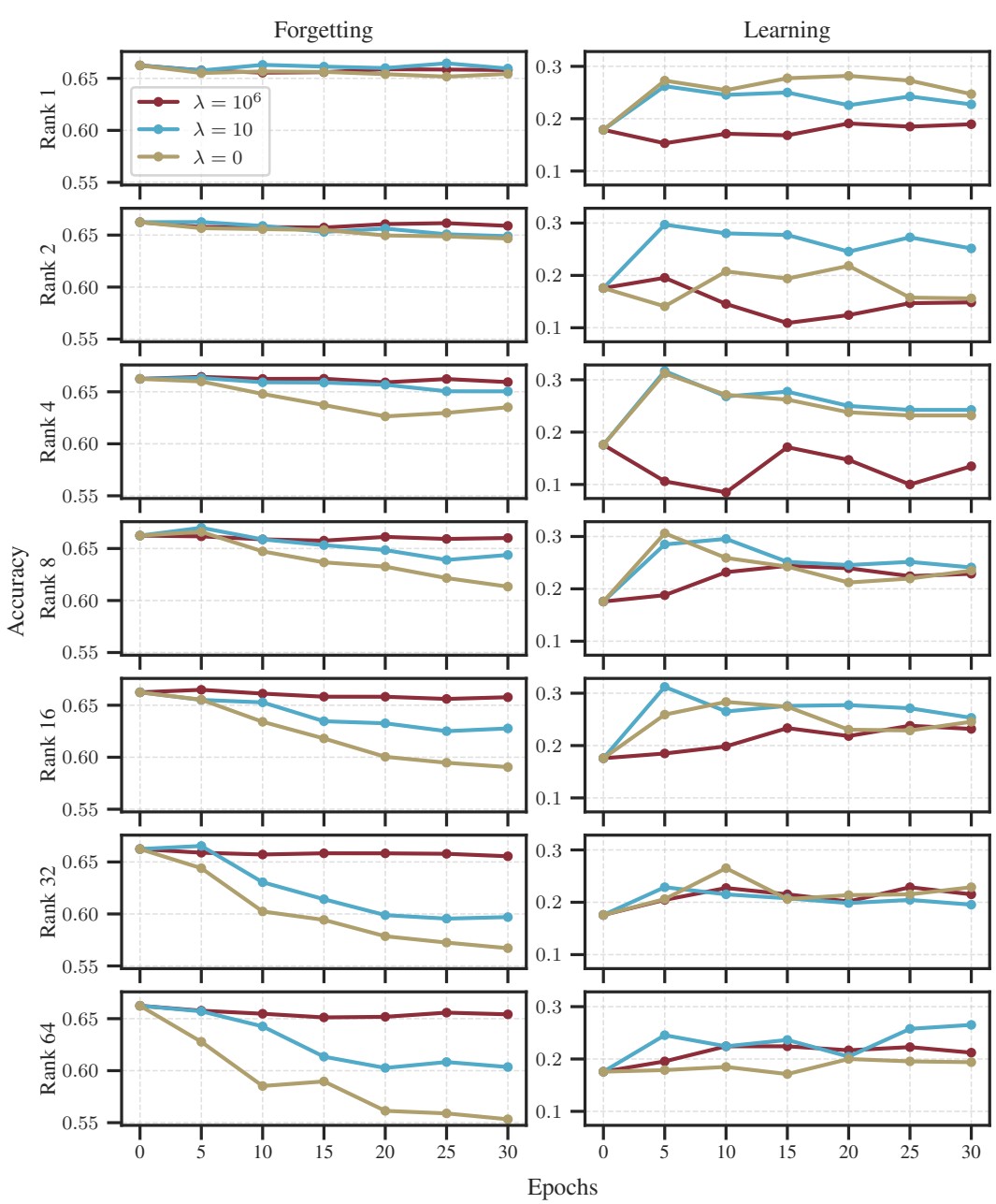

Figure 18: **Learning forgetting tradeoff is visible across different ranks of LoRA adapters**. Accuracy while fine-tuning: (*left*) average source-domain accuracy (forgetting) and (*right*) target-dataset accuracy (learning). We show results for one seed for $\lambda = 10$, $\lambda = 10^6$ and baseline. With bigger rank we notice more forgetting, as expected.

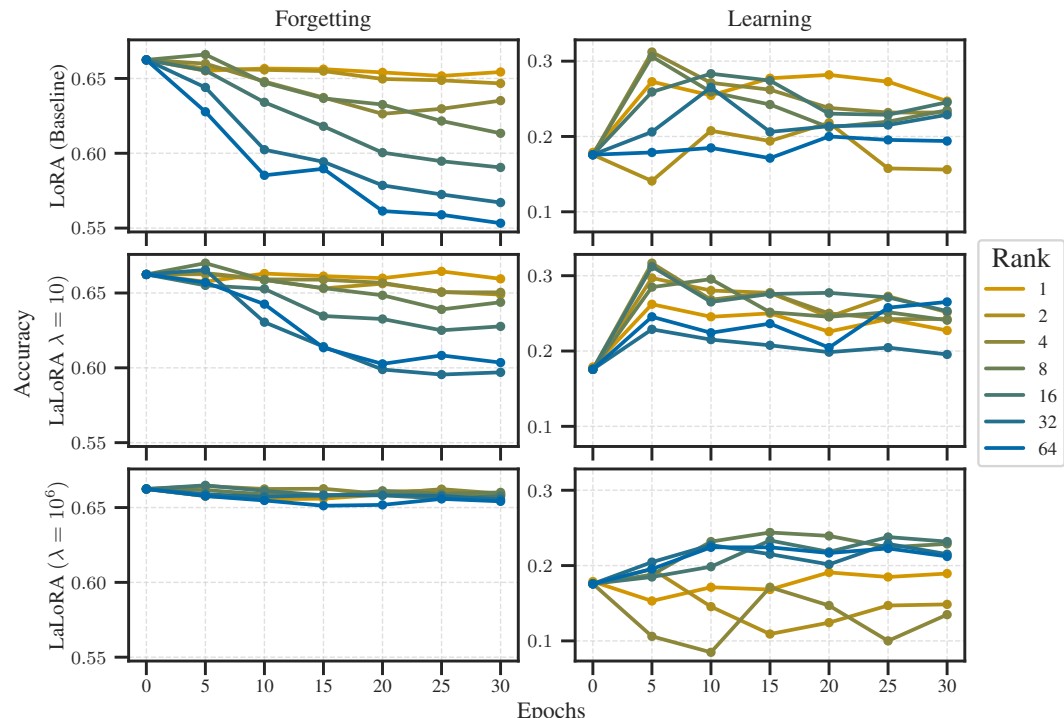

Figure 19: **Learning forgetting tradeoff is visible across different ranks of LoRA adapters**. Accuracy while fine-tuning: (*left*) average source-domain accuracy (forgetting) and (*right*) target-dataset accuracy (learning), (*top*) baseline (*middle*) LaLoRA with $\lambda = 10$ and (*bottom*) $\lambda = 10^6$. We show a run for one random seed.

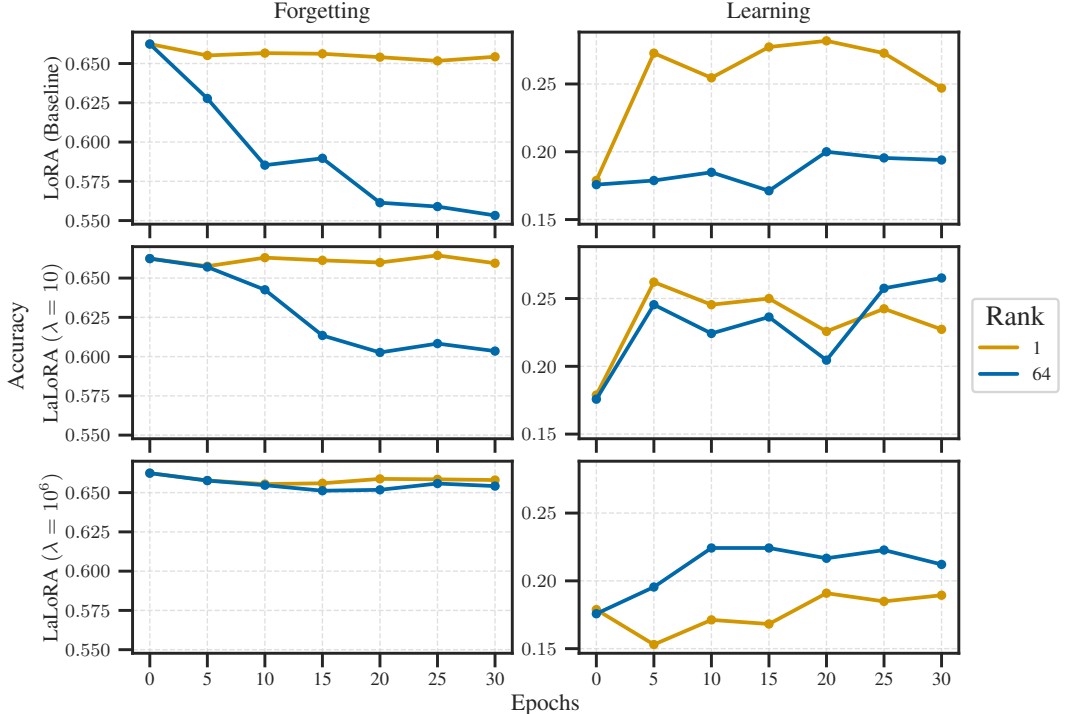

Figure 20: **Rank 1 vs rank 64: learning and forgetting trade-off**. Accuracy while fine-tuning: (*left*) average source-domain accuracy (forgetting) and (*right*) target-dataset accuracy (learning), (*top*) baseline (*middle*) LaLoRA with $\lambda = 10$ and (*bottom*) $\lambda = 10^6$. We show a run for one random seed.

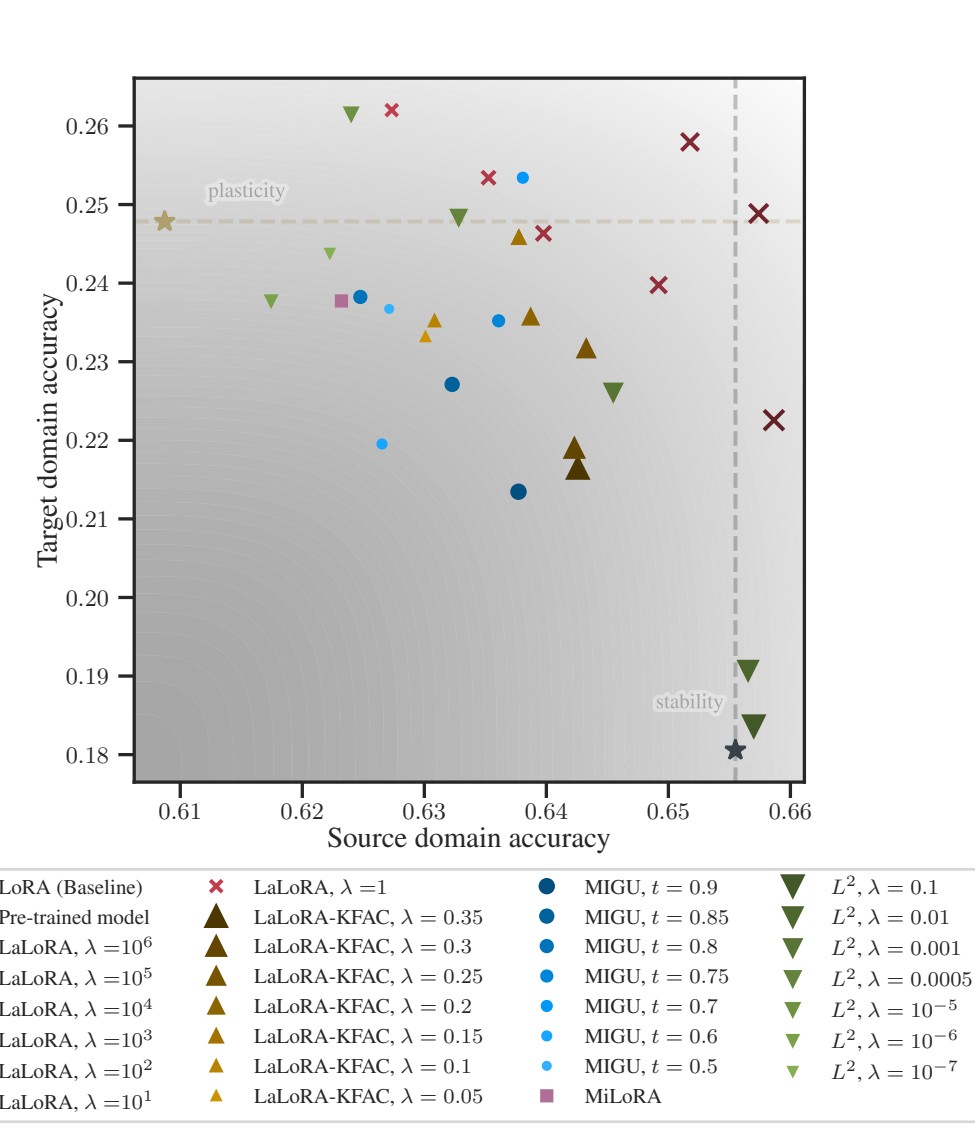

Figure 21: **B-KFAC is a good option but worse than the diagonal regularizer.** We present the results for different methods and LALoRA with the *one* batch of each dataset for the source domain. Final source ($x$-axis) vs. target ($y$-axis) accuracy for LaLoRA and competing methods, averaged over three seeds. The marker size & brightness indicates the value of each method's hyperparameter controlling the learning–forgetting trade-off.

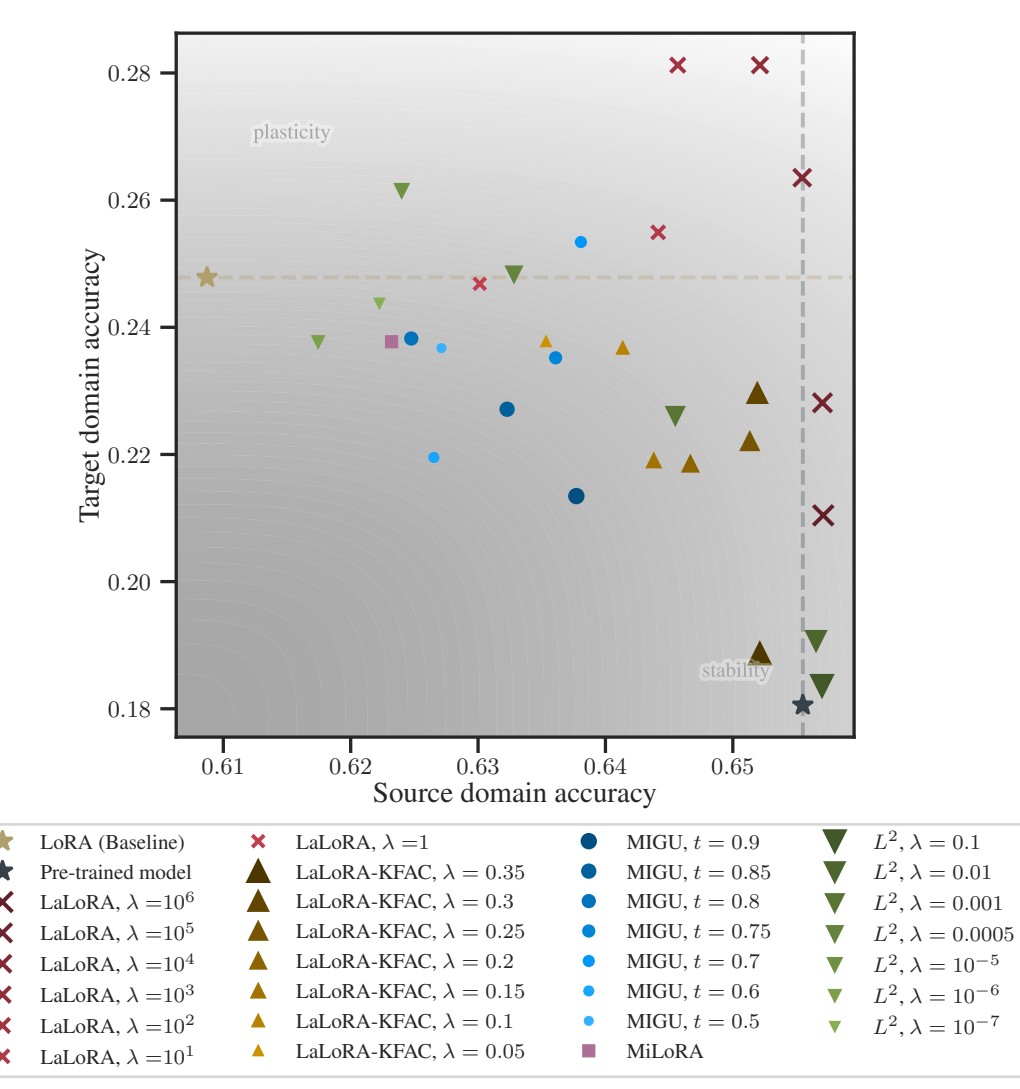

Figure 22: **B-KFAC is a good option but worse than the diagonal regularizer.** We present the results for different methods and LALoRA *two* batches of each dataset for the source domain. Final source ($x$-axis) vs. target ($y$-axis) accuracy for LaLoRA and competing methods, averaged over three seeds. The marker size & brightness indicates the value of each method's hyperparameter controlling the learning–forgetting trade-off.

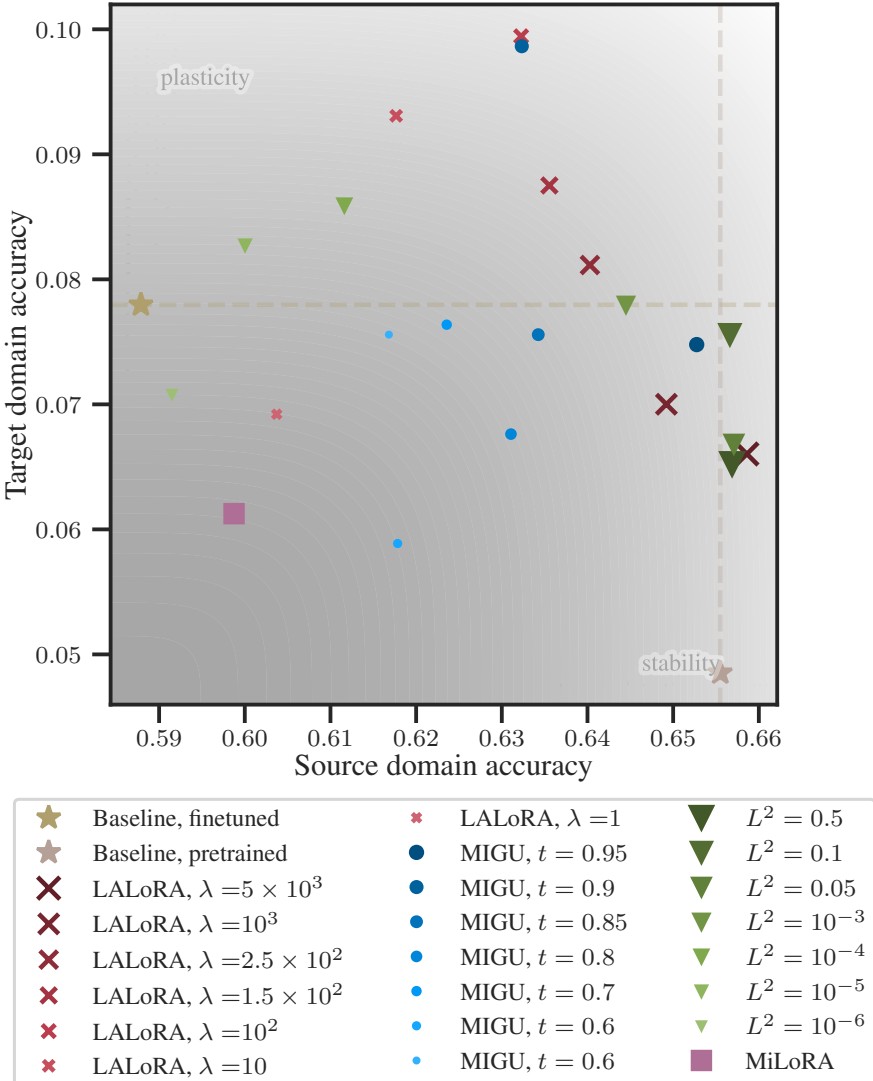

Figure 23: **Another target dataset (MATH): LaLoRA regularization improves the learning–forgetting trade-off.** Final source ($x$-axis) vs. target ($y$-axis) accuracy for LaLoRA and competing methods, averaged over three seeds. The marker size & brightness indicates the value of each method's hyperparameter controlling the learning–forgetting trade-off (e.g. $\lambda$ for LaLoRA, $t$ for MIGU).

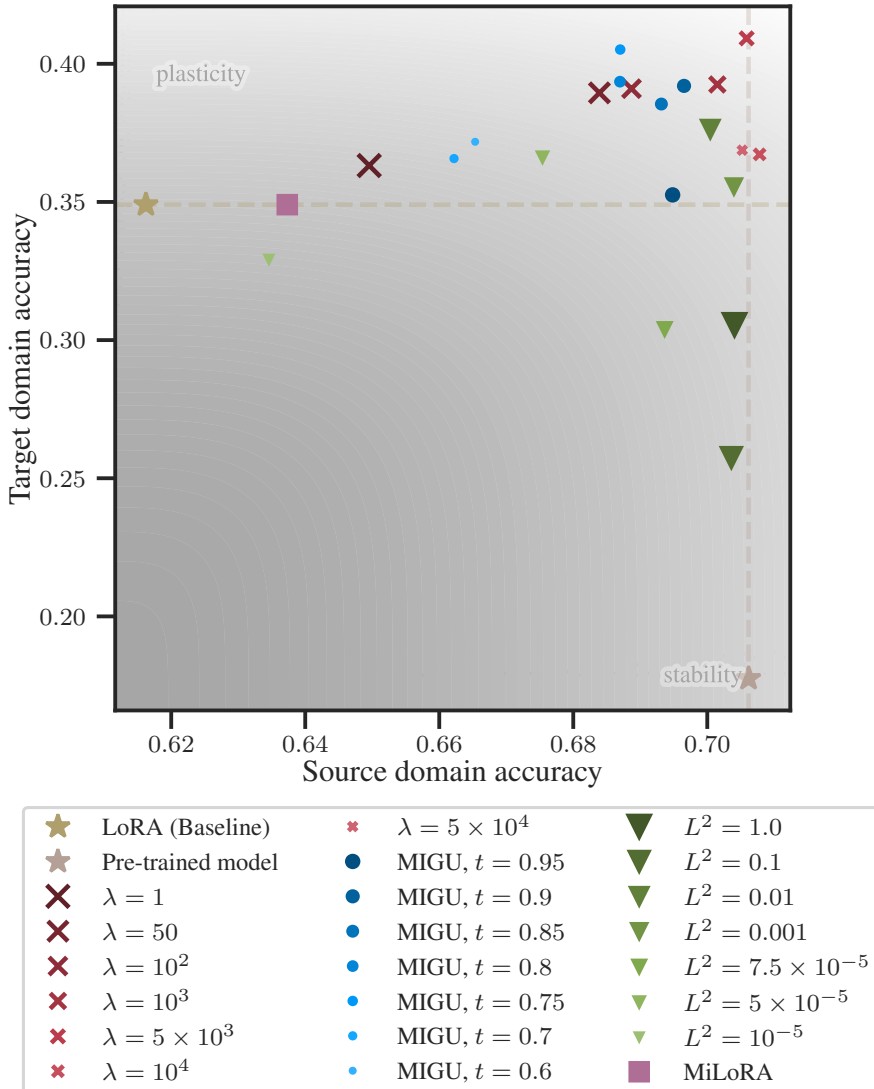

Figure 24: **Another model (Llama-3.1-8B): LaLoRA regularization improves the learning–forgetting trade-off.** Final source ($x$-axis) vs. target ($y$-axis) accuracy for LaLoRA and competing methods, averaged over three seeds. The marker size & brightness indicates the value of each method's hyperparameter controlling the learning–forgetting trade-off (e.g. $\lambda$ for LaLoRA, $t$ for MIGU).

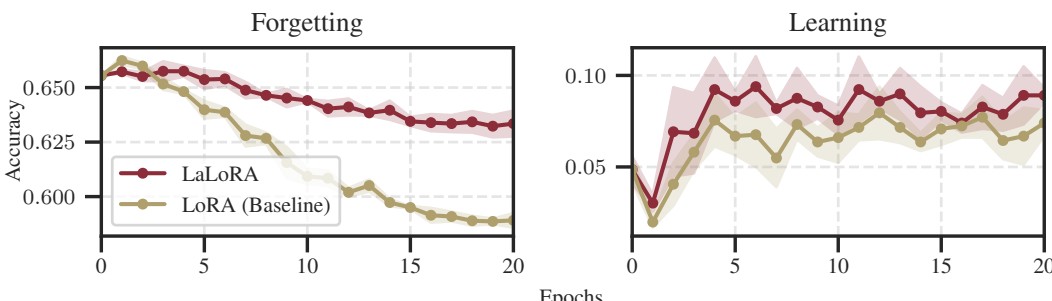

Figure 25: **Another target dataset MATH: Diagonal LALoRA regularization reduces forgetting while maintaining good learning ability.** Accuracy while fine-tuning: (*top*) average source-domain accuracy (forgetting) and (*bottom*) target-dataset accuracy (learning). We show mean and standard deviation across three random seeds. LALoRA is shown for $\lambda = 100$

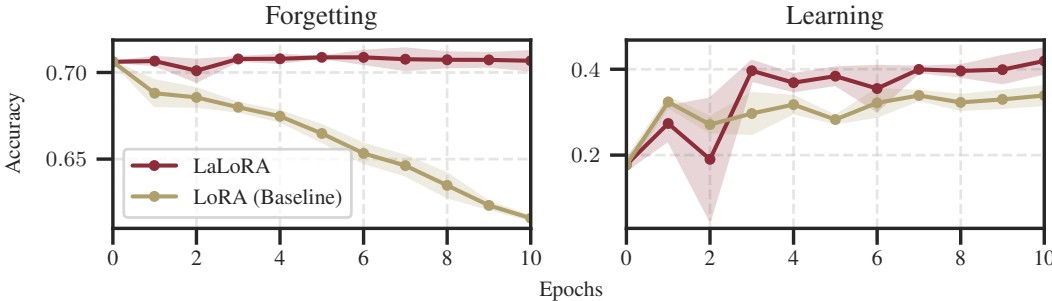

Figure 26: **Another model Llama-3.1-8B: Diagonal LALoRA regularization reduces forgetting while maintaining good learning ability.** Accuracy while fine-tuning: (*left*) average source-domain accuracy (forgetting) and (*right*) target-dataset accuracy (learning). We show mean and standard deviation across three random seeds. LALoRA is shown for $\lambda = 5000$.

Table 11: **Computational and memory cost of LaLoRA and other methods.** The symbols denote $B$ batch size of the current source data proxy sub-dataset for Laplace approximation, $B_{\text{train}}$ the train batch size of the target domain, $D_{\text{in}}$ the dimension of the input to A adapter and $D_{\text{out}}$ the dimension of the output of B adapter, $r$ is the rank, $N_{b_{\text{train}}}$ is the number of train batches and $N_b$ is the number of source dataset proxy sub-dataset number of batches. The costs are presented for **one** adapter layer. For vanilla LoRA, the training costs $\mathcal{O}(N_{b_{\text{train}}} B_{\text{train}}(D_{\text{in}} + D_{\text{out}})r)$ and $\mathcal{O}((D_{\text{in}} + D_{\text{out}})r)$ for storage.

| | Computation | Memory |
|---|---|---|
| LALoRA DIAG | Fit precision: $\mathcal{O}(N_b B(D_{\text{in}} + D_{\text{out}})r)$
Set the mode: $\mathcal{O}((D_{\text{in}} + D_{\text{out}})r)$
Regularizer: $\mathcal{O}((D_{\text{in}} + D_{\text{out}})r)$ | Fit precision: $\mathcal{O}((D_{\text{in}} + D_{\text{out}})r)$
Set the mode: $\mathcal{O}((D_{\text{in}} + D_{\text{out}})r)$
Regularizer: $\mathcal{O}((D_{\text{in}} + D_{\text{out}})r)$ |
| MIGU | Compute the mask: $\mathcal{O}((D_{\text{in}} + D_{\text{out}})r)$ | Compute the mask: $\mathcal{O}((D_{\text{in}} + D_{\text{out}})r)$ |
| MiLoRA | Compute SVD: $\mathcal{O}(D_{\text{in}} D_{\text{out}} \min(D_{\text{in}}, D_{\text{out}}))$
Set the A and B: $\mathcal{O}(D_{\text{in}} D_{\text{out}} r)$ | Compute SVD: $\mathcal{O}(\min(D_{\text{in}}, D_{\text{out}})(D_{\text{in}} + D_{\text{out}}))$
Set the A and B: $\mathcal{O}(D_{\text{in}} D_{\text{out}})$ |
| $L^2$ | Set the mode: $\mathcal{O}((D_{\text{in}} + D_{\text{out}})r)$
Forward: $\mathcal{O}((D_{\text{in}} + D_{\text{out}})r)$
Regularizer: $\mathcal{O}((D_{\text{in}} + D_{\text{out}})r)$ | Set the mode: $\mathcal{O}((D_{\text{in}} + D_{\text{out}})r)$
Forward: $\mathcal{O}((D_{\text{in}} + D_{\text{out}})r)$
Regularizer: $\mathcal{O}((D_{\text{in}} + D_{\text{out}})r)$ |

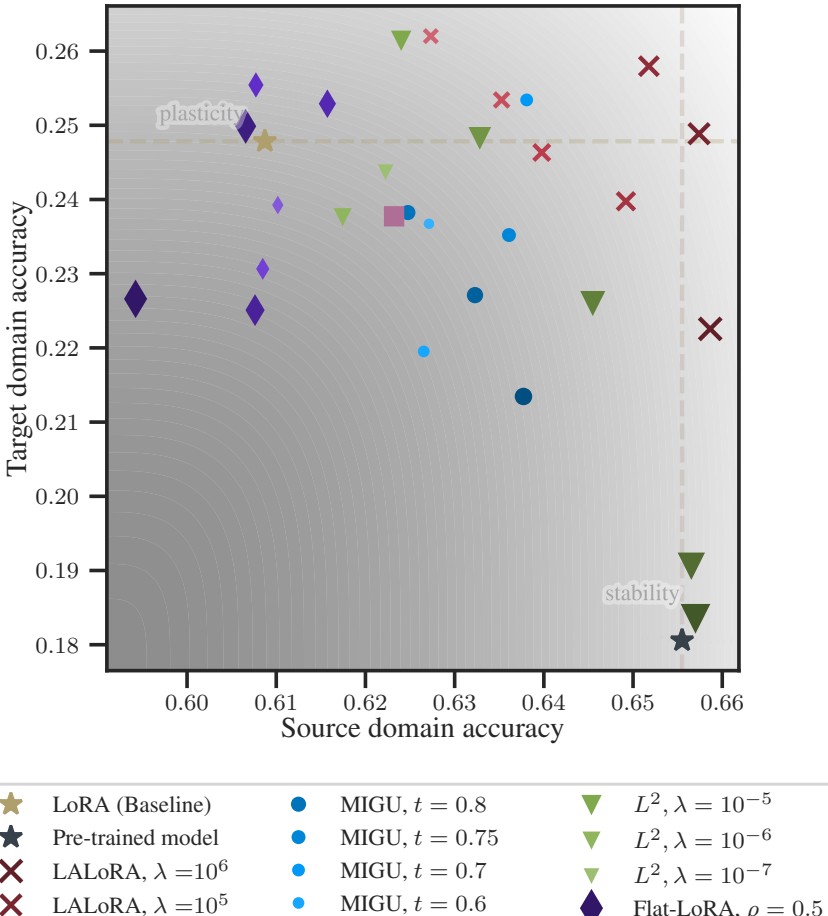

Figure 27: **Comparison to Flat-LoRA.** Final source ($x$-axis) vs. target ($y$-axis) accuracy for LaLoRA and competing methods, averaged over three seeds. The marker size & brightness indicates the value of each method's hyperparameter controlling the learning–forgetting trade-off (e.g. $\lambda$ for LaLoRA, $t$ for MIGU and $\rho$ for Flat-LoRA (Li et al., 2025)). Flat-LoRA can improve learning capabilities but does not mitigate forgetting of the source domain.

Table 12: **Computational and memory cost of different curvature approximations in LaLoRA.** The symbols denote $B$ batch size of the current source data proxy sub-dataset for Laplace approximation, $B_{\text{train}}$ the train batch size of the target domain, $D_{\text{in}}$ the dimension of the input to A adapter and $D_{\text{out}}$ the dimension of the output of B adapter, $r$ is the rank, $N_{b_{\text{train}}}$ is the number of train batches and $N_b$ is the number of source dataset proxy sub-dataset number of batches. The costs are presented for **one** adapter layer. For vanilla LoRA, the training costs $\mathcal{O}(N_{b_{\text{train}}} B_{\text{train}}(D_{\text{in}} + D_{\text{out}})r)$ and $\mathcal{O}((D_{\text{in}} + D_{\text{out}})r)$ for storage.

| LaLoRA | Computation | Memory |
|---|---|---|
| DIAG | Fit precision: $\mathcal{O}(N_b B(D_{\text{in}} + D_{\text{out}})r)$
Set the mode: $\mathcal{O}((D_{\text{in}} + D_{\text{out}})r)$
Regularizer: $\mathcal{O}((D_{\text{in}} + D_{\text{out}})r)$ | Fit precision: $\mathcal{O}((D_{\text{in}} + D_{\text{out}})r)$
Set the mode: $\mathcal{O}((D_{\text{in}} + D_{\text{out}})r)$
Regularizer: $\mathcal{O}((D_{\text{in}} + D_{\text{out}})r)$ |
| B-K-FAC | Fit precision:
 – compute A: $\mathcal{O}(N_b B(D_{\text{in}}^2 + r^2))$
 – compute G: $\mathcal{O}(N_b B(D_{\text{out}}^2 + r^2))$
Set the mode: $\mathcal{O}((D_{\text{in}} + D_{\text{out}})r)$
Regularizer: $\mathcal{O}(D_{\text{in}}^2 r + D_{\text{out}}^2 r + r^2(D_{\text{in}} + D_{\text{out}}))$ | Fit precision:
 – compute A: $\mathcal{O}((D_{\text{in}}^2 + r^2))$
 – compute G: $\mathcal{O}((D_{\text{out}}^2 + r^2))$
Set the mode: $\mathcal{O}((D_{\text{in}} + D_{\text{out}})r)$
Regularizer: $\mathcal{O}(D_{\text{in}}^2 + D_{\text{out}}^2)$ |
| B-TRI-K-FAC | Fit precision:
 – compute A: $\mathcal{O}(N_b B(D_{\text{in}}^2 + r^2 + r(D_{\text{in}} + D_{\text{out}})))$
 – compute G: $\mathcal{O}(N_b B(D_{\text{out}}^2 + r^2 + r(D_{\text{in}} + D_{\text{out}})))$
Set the mode: $\mathcal{O}((D_{\text{in}} + D_{\text{out}})r)$
Regularizer: $\mathcal{O}(D_{\text{in}}^2 r + D_{\text{out}}^2 r + r^2(D_{\text{in}} + D_{\text{out}}))$ | Fit precision:
 – compute A: $\mathcal{O}(D_{\text{in}}^2 + r^2 + r(D_{\text{in}} + D_{\text{out}}))$
 – compute G: $\mathcal{O}(D_{\text{out}}^2 + r^2 + r(D_{\text{in}} + D_{\text{out}}))$
Set the mode: $\mathcal{O}((D_{\text{in}} + D_{\text{out}})r)$
Regularizer: $\mathcal{O}(D_{\text{in}}^2 + D_{\text{out}}^2 + r^2)$ |

Table 13: **Comparing LA-LoRA regularization to other LoRA fine-tuning methods**. Final source/target domain accuracy ($\pm$ one standard deviation across three seeds). The hyperparameters $\lambda$, $t$, and $\rho$ were set to either prioritize source accuracy (*stability*) or target accuracy (*plasticity*). The right-most columns show percentage-point differences compared to the Baseline (LoRA), i.e. fine-tuning without a regularizer. Higher is better for both forgetting and learning.

| Method | Source domain | Target domain | Forgetting (pp) | Learning (pp) |
|---|---|---|---|---|
| LoRA (Baseline, $\lambda = 0$) | $60.9\% \pm 0.9\%$ | $24.8\% \pm 0.7\%$ | +0.0 | +0.0 |
| LALoRA, $\lambda = 10^6$ | $65.9\% \pm 0.1\%$ | $22.3\% \pm 0.9\%$ | +5.0 | -2.5 |
| LALoRA, $\lambda = 10^5$ | $65.7\% \pm 0.3\%$ | $24.9\% \pm 0.9\%$ | +4.9 | +0.1 |
| LALoRA, $\lambda = 10^4$ | $65.2\% \pm 0.4\%$ | $25.8\% \pm 0.4\%$ | +4.3 | +1.0 |
| LALoRA, $\lambda = 10^3$ | $64.9\% \pm 0.1\%$ | $24.0\% \pm 1.7\%$ | +4.0 | -0.8 |
| LALoRA, $\lambda = 10^2$ | $64.0\% \pm 0.3\%$ | $24.6\% \pm 1.7\%$ | +3.1 | -0.2 |
| LALoRA, $\lambda = 10^1$ | $63.5\% \pm 0.4\%$ | $25.3\% \pm 2.3\%$ | +2.7 | +0.6 |
| LALoRA, $\lambda = 1$ | $62.7\% \pm 0.1\%$ | $26.2\% \pm 1.2\%$ | +1.9 | +1.4 |
| MIGU, $t = 0.9$ | $63.8\% \pm 0.7\%$ | $21.3\% \pm 4.6\%$ | +2.9 | -3.4 |
| MIGU, $t = 0.85$ | $63.2\% \pm 0.2\%$ | $22.7\% \pm 0.3\%$ | +2.4 | -2.1 |
| MIGU, $t = 0.8$ | $62.5\% \pm 0.2\%$ | $23.8\% \pm 1.7\%$ | +1.6 | -1.0 |
| MIGU, $t = 0.75$ | $63.6\% \pm 0.1\%$ | $23.5\% \pm 2.4\%$ | +2.7 | -1.3 |
| MIGU, $t = 0.7$ | $63.8\% \pm 0.3\%$ | $25.3\% \pm 1.1\%$ | +2.9 | +0.6 |
| MIGU, $t = 0.6$ | $62.7\% \pm 0.2\%$ | $22.0\% \pm 1.3\%$ | +1.8 | -2.8 |
| MIGU, $t = 0.5$ | $62.7\% \pm 0.4\%$ | $23.7\% \pm 1.3\%$ | +1.8 | -1.1 |
| $L^2, \lambda = 0.1$ | $65.7\% \pm 0.0\%$ | $18.4\% \pm 3.6\%$ | +4.8 | -6.4 |
| $L^2, \lambda = 0.01$ | $65.7\% \pm 0.2\%$ | $19.1\% \pm 2.3\%$ | +4.8 | -5.7 |
| $L^2, \lambda = 10^{-3}$ | $64.5\% \pm 0.1\%$ | $22.6\% \pm 2.2\%$ | +3.7 | -2.2 |
| $L^2, \lambda = 5 \times 10^{-4}$ | $63.3\% \pm 0.4\%$ | $24.8\% \pm 1.4\%$ | +2.4 | +0.1 |
| $L^2, \lambda = 10^{-5}$ | $62.4\% \pm 0.2\%$ | $26.2\% \pm 2.8\%$ | +1.5 | +1.4 |
| $L^2, \lambda = 10^{-6}$ | $61.7\% \pm 0.9\%$ | $23.8\% \pm 2.8\%$ | +0.9 | -1.0 |
| $L^2, \lambda = 10^{-7}$ | $62.2\% \pm 0.5\%$ | $24.4\% \pm 3.6\%$ | +1.4 | -0.4 |
| MiLoRA | $62.3\% \pm 0.2\%$ | $23.8\% \pm 1.2\%$ | +1.4 | -1.0 |
| Flat-LoRA, $\rho = 0.5$ | $59.4\% \pm 0.8\%$ | $22.7\% \pm 1.8\%$ | -1.4 | -2.1 |
| Flat-LoRA, $\rho = 0.1$ | $60.7\% \pm 0.7\%$ | $25.0\% \pm 3.0\%$ | -0.2 | +0.2 |
| Flat-LoRA, $\rho = 0.05$ | $60.8\% \pm 0.7\%$ | $22.5\% \pm 3.2\%$ | -0.1 | -2.3 |
| Flat-LoRA, $\rho = 0.01$ | $61.6\% \pm 0.7\%$ | $25.3\% \pm 1.0\%$ | +0.7 | +0.5 |
| Flat-LoRA, $\rho = 0.005$ | $60.8\% \pm 0.3\%$ | $25.5\% \pm 0.9\%$ | -0.1 | +0.8 |
| Flat-LoRA, $\rho = 0.001$ | $60.8\% \pm 0.6\%$ | $23.1\% \pm 1.9\%$ | -0.0 | -1.7 |
| Flat-LoRA, $\rho = 0.0005$ | $61.0\% \pm 0.6\%$ | $23.9\% \pm 0.8\%$ | +0.1 | -0.9 |

Table 14: **Comparing LA-LoRA regularization to other LoRA fine-tuning methods on MATH**. Final source/target domain accuracy ($\pm$ one standard deviation across three seeds). The hyperparameters $\lambda$ and $t$ were set to either prioritize source accuracy (*stability*) or target accuracy (*plasticity*). The right-most columns show percentage-point differences compared to the Baseline (LoRA), i.e. fine-tuning without a regularizer.

| Method | Source domain | Target domain | Forgetting (pp) | Learning (pp) |
|---|---|---|---|---|
| LoRA (Baseline, $\lambda = 0$) | $58.8\% \pm 0.3\%$ | $7.8\% \pm 0.4\%$ | +0.0 | +0.0 |
| LALoRA, $\lambda = 5 \times 10^3$ | $65.9\% \pm 0.4\%$ | $6.6\% \pm 0.8\%$ | +7.1 | -1.2 |
| LALoRA, $\lambda = 10^3$ | $64.9\% \pm 0.2\%$ | $7.0\% \pm 0.5\%$ | +6.1 | -0.8 |
| LALoRA, $\lambda = 2.5 \times 10^2$ | $64.0\% \pm 0.5\%$ | $8.1\% \pm 1.0\%$ | +5.2 | +0.3 |
| LALoRA, $\lambda = 1.5 \times 10^2$ | $63.6\% \pm 0.4\%$ | $8.8\% \pm 1.2\%$ | +4.8 | +1.0 |
| LALoRA, $\lambda = 10^2$ | $63.2\% \pm 0.6\%$ | $9.9\% \pm 0.3\%$ | +4.4 | +2.1 |
| LALoRA, $\lambda = 10$ | $61.8\% \pm 0.3\%$ | $9.3\% \pm 1.7\%$ | +3.0 | +1.5 |
| LALoRA, $\lambda = 1$ | $60.4\% \pm 0.6\%$ | $6.9\% \pm 1.0\%$ | +1.6 | -0.9 |
| MIGU, $t = 0.95$ | $65.3\% \pm 0.4\%$ | $7.5\% \pm 1.1\%$ | +6.5 | -0.3 |
| MIGU, $t = 0.9$ | $63.2\% \pm 0.5\%$ | $9.9\% \pm 1.4\%$ | +4.4 | +2.1 |
| MIGU, $t = 0.85$ | $63.4\% \pm 0.7\%$ | $7.6\% \pm 0.7\%$ | +4.6 | -0.2 |
| MIGU, $t = 0.8$ | $63.1\% \pm 0.2\%$ | $6.8\% \pm 0.4\%$ | +4.3 | -1.0 |
| MIGU, $t = 0.7$ | $62.4\% \pm 0.4\%$ | $7.6\% \pm 0.9\%$ | +3.6 | -0.2 |
| MIGU, $t = 0.6$ | $61.8\% \pm 0.2\%$ | $5.9\% \pm 0.5\%$ | +3.0 | -1.9 |
| MIGU, $t = 0.6$ | $61.7\% \pm 0.1\%$ | $7.6\% \pm 1.8\%$ | +2.9 | -0.2 |
| $L^2 = 0.5$ | $65.7\% \pm 0.1\%$ | $6.5\% \pm 0.7\%$ | +6.9 | -1.3 |
| $L^2 = 0.1$ | $65.7\% \pm 0.0\%$ | $7.6\% \pm 1.7\%$ | +6.9 | -0.2 |
| $L^2 = 0.05$ | $65.7\% \pm 0.0\%$ | $6.7\% \pm 0.5\%$ | +6.9 | -1.1 |
| $L^2 = 10^{-3}$ | $64.4\% \pm 0.1\%$ | $7.8\% \pm 1.5\%$ | +5.7 | +0.0 |
| $L^2 = 10^{-4}$ | $61.2\% \pm 0.1\%$ | $8.6\% \pm 0.3\%$ | +2.4 | +0.8 |
| $L^2 = 10^{-5}$ | $60.0\% \pm 0.2\%$ | $8.3\% \pm 0.5\%$ | +1.2 | +0.5 |
| $L^2 = 10^{-6}$ | $59.1\% \pm 0.3\%$ | $7.1\% \pm 1.3\%$ | +0.4 | -0.7 |
| MiLoRA | $59.9\% \pm 0.2\%$ | $6.1\% \pm 0.7\%$ | +1.1 | -1.7 |

Table 15: **Comparing LA-LoRA regularization to other LoRA fine-tuning methods for LLAMA-3.1-8B**. Final source/target domain accuracy ($\pm$ one standard deviation across three seeds). The hyperparameters $\lambda$ and $t$ were set to either prioritize source accuracy (*stability*) or target accuracy (*plasticity*). The right-most columns show percentage-point differences compared to the Baseline (LoRA), i.e. fine-tuning without a regularizer.

| Method | Source domain | Target domain | Forgetting (pp) | Learning (pp) |
|---|---|---|---|---|
| LoRA (Baseline, $\lambda = 0$) | $61.6\% \pm 0.1\%$ | $34.9\% \pm 1.8\%$ | +0.0 | +0.0 |
| LALoRA, $\lambda = 5 \times 10^4$ | $70.5\% \pm 0.3\%$ | $36.9\% \pm 0.5\%$ | +8.9 | +2.0 |
| LALoRA, $\lambda = 10^4$ | $70.8\% \pm 0.3\%$ | $36.7\% \pm 6.7\%$ | +9.2 | +1.8 |
| LALoRA, $\lambda = 5 \times 10^3$ | $70.6\% \pm 0.5\%$ | $40.9\% \pm 1.6\%$ | +9.0 | +6.0 |
| LALoRA, $\lambda = 10^3$ | $70.1\% \pm 0.3\%$ | $39.3\% \pm 1.9\%$ | +8.5 | +4.4 |
| LALoRA, $\lambda = 10^2$ | $68.9\% \pm 0.1\%$ | $39.1\% \pm 2.3\%$ | +7.2 | +4.2 |
| LALoRA, $\lambda = 50$ | $68.4\% \pm 0.5\%$ | $38.9\% \pm 1.2\%$ | +6.8 | +4.0 |
| LALoRA, $\lambda = 1$ | $65.0\% \pm 0.4\%$ | $36.3\% \pm 4.7\%$ | +3.3 | +1.4 |
| MIGU, $t = 0.95$ | $69.5\% \pm 0.3\%$ | $35.3\% \pm 2.4\%$ | +7.9 | +0.4 |
| MIGU, $t = 0.9$ | $69.7\% \pm 0.3\%$ | $39.2\% \pm 2.0\%$ | +8.0 | +4.3 |
| MIGU, $t = 0.85$ | $69.3\% \pm 0.2\%$ | $38.5\% \pm 0.6\%$ | +7.7 | +3.6 |
| MIGU, $t = 0.8$ | $68.7\% \pm 0.6\%$ | $39.4\% \pm 1.8\%$ | +7.1 | +4.5 |
| MIGU, $t = 0.75$ | $68.7\% \pm 0.2\%$ | $40.5\% \pm 1.3\%$ | +7.1 | +5.6 |
| MIGU, $t = 0.7$ | $66.2\% \pm 0.8\%$ | $36.6\% \pm 2.2\%$ | +4.6 | +1.7 |
| MIGU, $t = 0.6$ | $66.5\% \pm 0.8\%$ | $37.2\% \pm 1.1\%$ | +4.9 | +2.3 |
| $L^2 = 1.0$ | $70.4\% \pm 0.1\%$ | $30.6\% \pm 2.4\%$ | +8.8 | -4.4 |
| $L^2 = 0.1$ | $70.4\% \pm 0.3\%$ | $25.7\% \pm 12.7\%$ | +8.7 | -9.2 |
| $L^2 = 0.01$ | $70.0\% \pm 0.0\%$ | $37.6\% \pm 0.0\%$ | +8.4 | +2.7 |
| $L^2 = 0.001$ | $70.4\% \pm 0.3\%$ | $35.6\% \pm 1.9\%$ | +8.8 | +0.7 |
| $L^2 = 7.5 \times 10^{-5}$ | $69.4\% \pm 0.3\%$ | $30.4\% \pm 9.7\%$ | +7.7 | -4.5 |
| $L^2 = 5 \times 10^{-5}$ | $67.5\% \pm 0.2\%$ | $36.6\% \pm 5.1\%$ | +5.9 | +1.7 |
| $L^2 = 10^{-5}$ | $63.5\% \pm 0.4\%$ | $32.9\% \pm 2.9\%$ | +1.8 | -2.0 |
| MiLoRA | $63.7\% \pm 0.3\%$ | $34.9\% \pm 1.4\%$ | +2.1 | -0.0 |

