# OpenReview forum: "Mitigating Forgetting in Low Rank Adaptation"
_ICLR.cc/2026/Conference — Submitted to ICLR 2026_

### Official Review · Reviewer_M9Lx · 2025-10-26

**Soundness:** 4
**Presentation:** 3
**Contribution:** 3
**Rating:** 8
**Confidence:** 4

**Summary:**

The paper targets catastrophic forgetting when fine-tuning LLMs with LoRA. It proposes LaLoRA, a curvature-aware regularizer built via a Laplace approximation computed over the LoRA weights only, then used to penalize updates along high-curvature (source-critical) directions during downstream training. Empirically, LaLoRA improves the stability–plasticity trade-off over LoRA and other baselines.

**Strengths:**

- Methodology is principled yet practical: local quadratic (Laplace) approximation over LoRA parameters gives per-weight uncertainty and a simple quadratic regularizer; computing it only for adapters keeps cost manageable and fits existing LoRA codepath.
- Thorough experiments & analyses: the effectiveness and robustness of the proposed method is demonstrated in various different experiment settings. Moreover, the parameter updating behavior and sensitivity to data coverage are both analyzed.

**Weaknesses:**

- Resource reporting gap: I couldn’t find a quantitative comparison of memory/storage and runtime overhead against LoRA/MIGU/MiLoRA under the stability-improving settings shown; adding this would strengthen the practical case.
- Data access assumption: effectiveness depends on having (proxy) source data to estimate curvature; guidance on minimal Ns and robustness is only partially discussed.

**Questions:**

- If one has no access to the pre-training data, is there any possible way of estimating $\Sigma$?
- Can the Laplace regularizer be adapted to other PEFT methods beyond original LoRA, such as DoRA, Pissa, MiLoRA?
- The proposed algorithm is a two-stage one. Will it be benifit if we dynamically adapt $\Sigma$ during fine-tuning?

---

> ### Author Response · Authors · 2025-11-19
>
> We thank the reviewer for their time and a very positive review. We appreciate their assessment of our work as a **principled, efficient and robust** approach to mitigate forgetting with **manageable costs**. Below, we address the two weaknesses and three questions point by point.
> ### W1. Resource reporting gap
> > I couldn’t find a quantitative comparison of memory/storage and runtime overhead against LoRA/MIGU/MiLoRA; adding this would strengthen the practical case.
>
> We agree that this part was not sufficiently visible and clear in the original draft. We provide an analysis of the computational cost in **Appendix F**. In Table 9, we report the measured **wall-clock runtime and GPU memory** of computing the Laplace approximation (Stage I). Table 10 provides the same for each step in the fine-tuning process (Stage II). Overall, we can observe that LaLoRA requires little overhead compared to regular LoRA fine-tuning.
>
> In the updated version, we have also added a **theoretical analysis** of the memory and compute requirements of each individual step in LaLoRA and the baselines (MIGU, MiLoRA, $L^2$) in Table 11. Table 12 compares the compute and memory requirements for different curvature approximations in LaLoRA.
>
> Please let us know what other analysis of computational efficiency you would like to see, or whether the changes addressed your concerns. We are also happy to move some of that analysis to the main text for better visibility.
>
> ---
>
> ### W2. Data Access Assumption
> > Effectiveness depends on having (proxy) source data to estimate curvature; guidance on minimal Ns and robustness is only partially discussed.
>
> It is true that *exact* pre-training data is not available for most LLMs. *However*, our claim is that LaLoRa can work with only *approximate* source domain data. We firmly believe that this is not just feasible but unproblematic in practice, due to the following points:
>
> 1) **LaLoRA works with little and approximate surrogate data**
>
>     In the paper, we explicitly study how sensitive LaLoRA is to the exact surrogate data:
>     * Figure 3b shows that computing the Laplace approximation from *only one* of the three surrogate datasets (e.g., WinoGrande) already improves forgetting *on all three surrogate source datasets* (second row). This indicates that the surrogate data used for the Laplace approximation does not even need to match the evaluation task to be useful.
>
>     * In Figure 4, we vary the number of batches used for the Laplace approximation. We find that even just 1-2 mini-batches per dataset capture most of the benefits. This indicates that even a very rough surrogate (based on very little source proxy data) is sufficient. In the updated paper version, we added Figure 11, where we showcase the learning and forgetting trade-off when using *all* batches from each surrogate sub-dataset and we find that using one batch only, leads to comparable gains for forgetting, may impact the learning capabilities, yet is more efficient.
>
>     * Figure 10 further confirms Figure 3b: broader surrogates help, but even rough approximations are already valuable.
>
> 2) **Surrogate data is available in practice**
>
>     In many realistic scenarios, practitioners do have access to approximate source data:
>
>     * There are many high-quality open pre-training corpora, such as the FineWeb variants or Red Pajama, that cover similar domains and likely have a significant overlap with proprietary pre-training data.
>     * Model releases routinely report performance on public benchmarks designed to capture the very capabilities users want to preserve under fine-tuning. Using them as surrogates for the source domain is therefore aligned with how the models are evaluated during pre-training.
>
> 3) **This is a standard approach in the literature**
>
>     The need for (some notion of) source data is not unique to LaLoRA.
>
>     * Many continual learning methods rely on source surrogates, for example, EWC (Kirkpatrick et al., 2017), OSLA (Ritter et al., 2018), or replay-based methods (Wang et al., 2024, review). Following Biderman et al. (2024), we use WinoGrande/ARC/HellaSwag to "assess degradation of base model capabilities" (Biderman et al., 2024). For LaLoRA, we use the same data to compute a curvature-based regularizer to ensure that we retain these base model capabilities.
>
> **Suggestion**: We agree that the topic of source data requirements is important. We are happy to elaborate on LaLoRA's need for surrogate source data in the updated paper version by incorporating the discussion above, and can highlight it more clearly in the Limitations section.

---

> > ### Author Response · Authors · 2025-11-19
> >
> > ### Q1. Estimating $\mathbf{\bar{\Sigma}}^{-1}$ without Pre-Training Data
> > > If one has no access to the pre-training data, is there any possible way of estimating
> >
> > We don’t need access to the original pre-training data, only data that probes the same capabilities. To mitigate forgetting during fine-tuning, we assume practitioners have at least an approximate measure of the capabilities they want to preserve.
> >
> > As a structural alternative, as noted in the paper, model providers could release curvature information (e.g., a diagonal Laplace approximation) alongside the weights, allowing end-users to apply LaLoRA and related methods without any pre-training data or surrogates.
> >
> > ---
> >
> > ### Q2. Adaptability to other PEFT Methods
> > > Can the Laplace regularizer be adapted to other PEFT methods beyond original LoRA, such as DoRA, Pissa, MiLoRA?
> >
> > Yes! Conceptually, LaLoRA only requires a set of trainable adapter weights and a way to compute a Laplace approximation with respect to them. During fine-tuning, we then use a quadratic regularizer. This recipe also directly works with DoRA, PiSSA, MiLoRA, etc. We will explicitly mention this extensibility in the Conclusion.
> >
> > ---
> >
> > ### Q3. Dynamic Adaptation of $\mathbf{\bar{\Sigma}}^{-1}$ during Fine-Tuning
> > > Will it be benefit if we dynamically adapt Sigma during fine-tuning?
> >
> > This is a very interesting, but non-trivial, research question. We thought about this as well, and we view adaptive or online Laplace regularization as promising future work, but beyond the scope of this paper. In the revised paper version, we will briefly discuss this extension in the Future Work section. We want to briefly elaborate here:
> > * In order to correctly update the quadratic regularizer, we would need to not only update $\mathbf{\bar{\Sigma}}^{-1}$ during fine-tuning but also $\mathbf{\mu}$.
> > * From a computational point of view, online updates are possible, but would add additional computation. In order to keep it efficient, we would likely need to update $\mathbf{\bar{\Sigma}}^{-1}$ only every $n$ steps, with $n$ being an additional hyperparameter.
> >
> > ---
> >
> > *Mentioned papers:*
> >
> > [Kirkpatrick et al., 2017] James Kirkpatrick, Razvan Pascanu, Neil Rabinowitz, Joel Veness, Guillaume Desjardins, Andrei A. Rusu, Kieran Milan, John Quan, Tiago Ramalho, Agnieszka Grabska-Barwinska, Demis Hassabis, Claudia Clopath, Dharshan Kumaran, Raia Hadsell. "Overcoming Catastrophic Forgetting in Neural Networks." 2017
> >
> > [Ritter et al., 2018] Hippolyt Ritter, Aleksandar Botev, David Barber. "Online Structured Laplace Approximations For Overcoming Catastrophic Forgetting." 2018
> >
> > [Wang et al., 2024] Liyuan Wang, Xingxing Zhang, Hang Su, and Jun Zhu. "A Comprehensive Survey of Continual Learning: Theory, Method and Application." 2024
> >
> > [Biderman et al., 2024] Dan Biderman, Jacob Portes, Jose Javier Gonzalez Ortiz, Mansheej Paul, Philip Greengard, Connor Jennings, Daniel King, Sam Havens, Vitaliy Chiley, Jonathan Frankle, Cody Blakeney, John P. Cunningham. "LoRA Learns Less and Forgets Less." 2024

---

### Official Review · Reviewer_wrdx · 2025-10-28

**Soundness:** 3
**Presentation:** 3
**Contribution:** 3
**Rating:** 6
**Confidence:** 4

**Summary:**

The paper introduces LaLoRA, a new weight-space regularization technique designed to mitigate catastrophic forgetting in parameter-efficient fine-tuning scenarios, particularly with Low-Rank Adaptation (LoRA) layers in large language models. By leveraging a Laplace approximation, the method estimates parameter uncertainty and imposes a curvature-aware penalty that restricts changes to "critical" parameters within the LoRA adapters during downstream task fine-tuning. The method is evaluated on math reasoning transfer with Llama models, demonstrates improved learning-forgetting trade-offs over several recent baselines, and is analyzed for sensitivity to curvature approximation, source data, and hyperparameter settings.

**Strengths:**

1. **Clear Motivation**: The application of the Laplace approximation specifically to LoRA adapters is well-justified, namely finding those less important weights in pretraining. Moreover, it is demonstrated to be efficient, and avoids the prohibitive computational cost of full-parameter curvature modeling. The two-stage algorithm is clearly described and mathematically well-formulated, e.g., in Equations (1)-(5), with careful distinction between diagonal and structured Kronecker-Factored (K-FAC) curvature variants.

2. **Improved Pareto Trade-off**: The method effectively controls the stability-plasticity (learning-forgetting) trade-off via the regularization strength $\lambda$. Both main claims as improved source domain retention and competitive target domain learning are supported by compelling empirical evidence across Figures 2a, 2b, and Table 1. Moreover, the authors clearly demonstrate the advantages of the proposed methods with proper visualization.

3.  **Insightful Mechanism Analysis**: The paper investigates the actual update patterns induced by the regularizer (Figure 3a), showing that high-curvature ("important") weights are indeed less modified than more flexible ones. This is an excellent validation of the intuition behind the method.

**Weaknesses:**

1. **Unclear Theoretical Guarantees and Some Ambiguous Symbolism**: The Laplace-regularized loss in Equation (5) and associated regularizer expression could be made clearer, with more rigorous notation for how $\overline{\Sigma}$ is estimated, especially in multi-dataset settings. Although the practical motivation for restricting the Laplace approximation to LoRA weights is strong, a more explicit analysis of the cases where this is justified (i.e., under what assumptions low-rank space alone suffices to capture key catastrophic forgetting axes in full-parameter space) would strengthen the theoretical contribution.

2. **Limited Empirical Setting**: All experiments are conducted on a single base model (Llama-3.2-3B) and on mathematical reasoning as the target with commonsense datasets as proxies for source domains. While this setup is well-motivated (per Biderman et al., 2024; Shuttleworth et al., 2025), the conclusions are less generalizable to other domains and model families. Figure 11 does show per-dataset breakdown but broader task diversity would make the results more convincing.

3. **Unreported Implementation and Efficiency Details**: Several important implementation details remain unreported. For example, it is unclear whether adding the proposed regularization introduces additional GPU memory consumption or computational overhead. Furthermore, the paper does not discuss how efficiently the regularization terms can be computed in practice.

**Questions:**

1. Could the authors more rigorously quantify or theoretically bound how the choice of proxy dataset for the Laplace approximation affects retention? Is there an evaluable metric for proxy dataset representativeness? What are the signs of proxy failure?

2. Since the proposed regularization method is pretrained task-dependent, it often requires selecting a surrogate dataset when the original pretraining data are unavailable. This raises concerns about potential **dataset dependency**. Specifically, would the results vary significantly depending on the choice of surrogate dataset? Could different datasets lead to different degrees of memorization or forgetting? Furthermore, if certain samples are not represented in the surrogate dataset, is it possible for the model to still forget relevant information?

3. K-FAC vs. Diagonal: Why do richer curvature approximations (e.g., block tridiagonal K-FAC) not outperform diagonal ones? Is this due to rank limitations, poor sample efficiency, or structural redundancy in LoRA updates? Could the authors provide a concrete case (or synthetic experiment) demonstrating when more curvature pays off—or never does?

4. While excluding less important parameters may improve efficiency, it simultaneously restricts the flexibility of the adaptation process. The reviewer is uncertain whether, in very low-rank settings (e.g., r=1), this selective updating strategy could lead to degraded performance. Can authors test both very low and high rank cases and draw useful insights? The reviewer doubts the performance if r=1, since we have limited parameters to update, and narrawing down further may (significantly) hurt the performance.

5. Follow-up on Weakness 3 – Computational and Memory Overhead: As noted in Weakness 3, the paper does not clearly report the additional computational time and memory cost introduced by the proposed method. Moreover, for larger models, it remains uncertain whether computing the regularization terms would lead to substantial increases in training time or resource consumption.

6. The authors mention using a diagonal Fisher Information Matrix (FIM) to compute $\Sigma^{-1}$. Could this simplification lead to a performance drop, particularly when dealing with high-dimensional gradients where off-diagonal correlations may be important? Furthermore, how efficient is this approximation in practice compared to using a full or low-rank FIM?

---

> ### Author Response · Authors · 2025-11-19
>
> We thank the reviewer for the positive, detailed, and constructive review. We appreciate their assessment of our work as a **well-justified, efficient** approach to mitigate forgetting, supported by the experiments. Below, we address the three mentioned weaknesses and the remaining open questions.
> ### W1. Theory and Ambiguity around $\mathbf{\bar{\Sigma}}^{-1}$
> > The Laplace-regularized loss in Equation (5) and associated regularizer expression could be made clearer [...], especially in multi-dataset settings.
>
> We agree that the notation around Equation (5) and the definition of $\mathbf{\bar{\Sigma}}^{-1}$ can be made clearer. We updated this section of the paper, please look at lines 156-159 and 168-181, incorporating indices for the current batch $i$ used to compute the precision and a given sub-dataset $j$. We will work on further improving the notation.
>
> The high-level idea is that we essentially compute a curvature estimate (in our case, using the diagonal of a Fisher Information Matrix) of the proxy source-data loss landscape for each source sub-dataset. i.e., how does the source data loss change if we perturb the LoRA weights from their initialization? Combining different source sub-datasets simply means combining the precisions by summing them.
>
> > i.e., under what assumptions low-rank space alone suffices to capture key catastrophic forgetting axes in full-parameter space
>
> During fine-tuning, we only change the low-rank LoRA weights. Therefore, all catastrophic forgetting is due to LoRA weights differing from their initialization (since at initialization, they do not alter the model's output, see Lines 146-149). Our regularizer applies to all LoRA weights and thus all of the weights that (potentially) changed during fine-tuning. We will make this clearer in an updated draft.
>
> ---
>
> ### W2. Limited Empirical Setting
> > All experiments are conducted on a single base model [...] broader task diversity would make the results more convincing.
>
> To tackle your concern about a single model and fine-tuning dataset, we have added *two additional experiments*:
> 1) We ran the same experiment using a **larger base model**, namely *Llama-3.1-8B*. Results can be found in Figure 24 (trade-off across methods), 26 (learning and forgetting across epochs) and Table 8 (validation performance).
> 2) We also tested a **different fine-tuning dataset**, namely the *MATH dataset* (see Figure 23 (trade-off across methods), 25 (learning and forgetting across epochs) and Table 7 (validation performance).
>
> We can observe the same qualitative pattern: For most (but admittedly not all) desired trade-offs of target-domain and source-domain performance, LaLoRA provides the best learning-forgetting trade-off (e.g., better source accuracy at comparable target performance or vice versa).
>
> ---
>
> ### W3. (and Q5) Implementation and Efficiency Details
> > [...] it is unclear whether adding the proposed regularization introduces additional GPU memory consumption or computational overhead
>
> > [Q5] Moreover, for larger models, it remains uncertain whether computing the regularization terms would lead to substantial increases in training time or resource consumption.
>
> We agree that this part was not sufficiently visible and clear in the original draft. We provide an analysis of the computational cost in **Appendix F**. In Table 9, we report the measured **wall-clock runtime and GPU memory** of computing the Laplace approximation (Stage I). Table 10 provides the same for each step in the fine-tuning process (Stage II). Overall, we can observe that LaLoRA requires little overhead compared to regular LoRA fine-tuning.
>
> In the updated version, we have also added a **theoretical analysis** of the memory and compute requirements of each individual step in LaLoRA and the baselines (MIGU, MiLoRA, $L^2$) in Table 11. Table 12 compares the compute and memory requirements for different curvature approximations in LaLoRA.
>
> Please let us know what other analysis of computational efficiency you would like to see, or whether the changes addressed your concerns. We are also happy to move some of that analysis to the main text for better visibility.

---

> ### Author Response · Authors · 2025-11-19
>
> ### Q1. + Q2. Dataset Dependency
> > [Q1] Could the authors more rigorously quantify or theoretically bound how the choice of proxy dataset for the Laplace approximation affects retention? Is there an evaluable metric for proxy dataset representativeness? What are the signs of proxy failure?
>
> > [Q2] Specifically, would the results vary significantly depending on the choice of surrogate dataset? Could different datasets lead to different degrees of memorization or forgetting? Furthermore, if certain samples are not represented in the surrogate dataset, is it possible for the model to still forget relevant information?
>
> While we do not provide a formal bound on the choice of proxy dataset, we empirically probe this. In Figure 3b) (and also more extensively, Figure 10), we vary which source dataset (WinoGrande, ARC, HellaSwag) is used to compute the Laplace approximation. For both WinoGrande and ARC, even when $\mathbf{\bar{\Sigma}}^{-1}$ is estimated using *only a single* surrogate dataset, LaLoRA improves retention across *all three* datasets. This indicates some robustness to proxy choice.
>
> However, the exact result likely strongly depends on the choice of surrogate dataset. E.g., if a capability isn't covered by any proxy source dataset, it is much more likely that our regularizer will not improve its performance.
>
> In Figure 4, we vary the number of mini-batches used to compute the Laplace approximation. We find that 1-2 batches capture most of the benefits. This further indicates that LaLoRA is not overly sensitive to the precise sampling. In the updated version of the paper, we added Figure 11, where we showcase the learning and forgetting trade-off when using *all* batches from each surrogate sub-dataset. We find that using one batch only, leads to comparable gains for forgetting, may impact the learning capabilities, yet is more efficient.
>
> In conclusion, our results indicate that even a rough coverage of source data capabilities is enough to allow LaLoRA to retain source knowledge.
>
> ---
>
> ### Q3. + Q6. Diagonal Approximation (vs. K-FAC)
> > [Q3] Why do richer curvature approximations (e.g., block tridiagonal K-FAC) not outperform diagonal ones?
>
> > [Q6] Could this simplification [the diagonal approximation] lead to a performance drop, particularly when dealing with high-dimensional gradients where off-diagonal correlations may be important? Furthermore, how efficient is this approximation in practice compared to using a full or low-rank FIM?
>
> We did experiment with more expressive curvature approximations (B-K-FAC) but didn't find benefits versus the cheap diagonal approximation (see Figures 21 and 22). We don't have a fully satisfying answer to why this is the case, and in what settings this trend might change. In the paper, we wanted to avoid providing overly speculative answers, but here are a few of our intuitions and possible answers:
> * Due to the low-rank structure of the LoRA update, different curvature approximations on the LoRA weights represent different covariance structures on the parameter space itself. In particular, a block-diagonal K-FAC approximation on the LoRA weights, despite containing a full Kronecker factor for the low-rank space, in fact collapses into a much simpler covariance structure on the parameter space that treats all low-rank terms the same (see Appendix C.1 in the paper).
> * Comparing Figures 21 and 22, we observe that the K-FAC variant performs significantly better with more data (used to compute the Laplace approximation). It is possible that the richer curvature approximation requires more data for a high-fidelity estimation.

---

> ### Author Response · Authors · 2025-11-19
>
> ### Q4. Very Low Rank Setting ($r=1$)
> > The reviewer is uncertain whether, in very low-rank settings (e.g., r=1), this selective updating strategy could lead to degraded performance. Can authors test both very low and high rank cases and draw useful insights? The reviewer doubts the performance if r=1, since we have limited parameters to update, and narrawing down further may (significantly) hurt the performance.
>
> Please note LaLoRA's added regularizer is only a soft constraint (i.e., a quadratic penalty term). So it does not lead to fewer parameters to update or a *hard* narrowing down in the case of $r=1$. It simply penalizes updates that (likely) increase the source data performance.
>
> Instead, the optimal regularization strength $\lambda$ most likely depends on the chosen rank.
>
> * For larger ranks (e.g., Rank 64 or 32 in Figure 14, bottom two rows), we can heavily penalize (red line, $\lambda = 10^6$) and still get good learning behavior (red line shows little forgetting but learning roughly on par with the other lines).
>
> * In contrast, for smaller ranks (e.g., Rank 2 or 4 in Figure 18, top three rows), we should penalize less (blue, $\lambda = 10$). In those cases, a heavy penalization (red line) leads to little learning, but a lower regularization strength still shows improvement over the baseline (LoRA, gold line).
>
> We have added experiments with $r = 1$ to the Appendix, please see the updated Figure 15 (regularization strength $\lambda=10^3$) and Figure 18 (regularization strength $\lambda=10$ and $\lambda=10^6$) , as well as, Figure 16, 17, 19 and 20 for different visualizations. We notice that fine-tuning with a higher rank leads to more forgetting of the source domain. For $r= 1$, we observe a degraded performance on the target domain compared to the baseline for $\lambda=10^3$. As mentioned above, a choice of a different $\lambda$ may be more fitting, we provide an updated Figure 18 with $r=1$ (with $\lambda=10$ and $\lambda=10^6$) to further study this setting. We observe that a lower regularization strength ($\lambda = 10$) yields better source-domain performance and less degradation of the learning capabilities compared to the baseline.
>
> Note that this is only a heuristic and that the optimal $\lambda$ likely depends on many factors, most importantly the practitioner's desired trade-off between prioritizing learning and knowledge retention.
>
> ---
>
> We hope our responses have addressed your concerns. If so, we would like to kindly ask you to increase your score accordingly. If you still have any remaining concerns, please, let us know.

---

> ### Comment · Reviewer_wrdx · 2025-11-27
> **Response to Authors' Rebuttal**
>
> The reviewer would like to thank the authors for their detailed response.
>
> A few concerns, however, still remain:
>
> **(1) Data dependency.**
> This issue was also raised by another reviewer. The current rebuttal provides only conceptual arguments, but the reviewer believes this question requires a more rigorous empirical investigation. In particular, the authors are encouraged to conduct ablation studies that include:
> - evaluations on additional datasets,
> - analyses of how the method behaves when the chosen dataset is suboptimal or inappropriate, and
> - demonstrations of whether and how such choices affect overall performance.
>
> **(2) Rank-dependent hyperparameter \(\lambda\).**
> From the appended figures, it is clear that performance is sensitive to the choice of \(\lambda\) (in particular, rank=1), and its optimal value appears to vary with the rank. This raises an important practical question: **How should \(\lambda\) be selected for different ranks in a principled or reliable way?** Any guidance, rule-of-thumb, or empirical strategy would be helpful to ensure the method is usable in practice.

---

> > ### Author Response · Authors · 2025-12-01
> >
> > We would like to further address the reviewer’s concerns.
> >
> > **(1) Data dependency.**
> >
> > Apart from the conceptual arguments, we **did include empirical investigation in the original paper** i.e.
> >
> > * Figure 3b shows that computing the Laplace approximation from **only one of the three surrogate datasets** (e.g., WinoGrande) already improves forgetting on all three surrogate source datasets (second row). However, using only Hellaswag for LA, leads to worse performance on the source domain which addresses the **suboptimal proxy** question. Figure 10 further confirms Figure 3b for different $\lambda$ values.
> >
> > * In Figure 4, we vary the **number of batches** used for the Laplace approximation. In the updated version in Figure 11, we showcase the learning and forgetting trade-off when using **all batches** from each surrogate sub-dataset.
> >
> > We agree that further analysis of the source domain proxies is an interesting research direction. We have addressed **three** such source domain proxies, where one (HellaSwag) doesn’t seem like the most optimal choice.
> >
> > **Suggestion:** We will extend the ablation by adding more proxies, and include an experiment when "wrong" data is used (e.g. math or coding). Due to the time constraint and nature of conducting such an ablation study, we will add it in the camera-ready version.
> >
> > **(2) Rank-dependent hyperparameter ($\lambda$).**
> >
> > The reviewer posed a valid question on how to couple regularization strength and the rank. Our results in Figures 15-20 shed a light on the relation between the two hyperparameters. We can observe that *lower ranks require smaller $\lambda$ values* ($\lambda= 10$) vs *higher ranks deal well with bigger strengths* e.g. $\lambda=10^6$. We do recognise the limitation of having an additional hyperparameter that requires tuning which we mentioned in the Limitations section.

---

### Official Review · Reviewer_CuHz · 2025-10-30

**Soundness:** 3
**Presentation:** 2
**Contribution:** 2
**Rating:** 4
**Confidence:** 4

**Summary:**

The paper proposes LaLoRA, a Laplace-regularized variant of LoRA designed to mitigate catastrophic forgetting during fine-tuning. It applies a Laplace approximation on LoRA parameters to estimate curvature and penalizes updates along high-curvature directions. Experiments on LLaMA-3B fine-tuning for GSM-8K show improved retention of pre-training knowledge with a minor loss in target performance.

**Strengths:**

- The idea is conceptually sound, combining Laplace-based uncertainty estimation with LoRA.
- The paper draws a clear connection to EWC-style continual learning while adapting it to PEFT.

**Weaknesses:**

- The method assumes the availability of source or surrogate data to estimate curvature, which is unrealistic for most LLM fine-tuning scenarios. The proposed "minimal proxy batches" solution only partially addresses this.
- No analysis of computational efficiency relative to vanilla LoRA is given; specifically, incorporating the cost from Stage I.
- It is unclear how much source-domain data is required for LaLoRA to perform well, or if the regularization is robust when limited data are available.
- Equations 11-12 are poorly typeset and confusing - I believe there is a notation error that makes it unclear how the regularization term is applied.
- Experimental results are modest and high-variance (Table 2; no actual gain in performance with high variance). Only a single model/task was evaluated.
- The claimed practicality needs some quantitative analysis, as Stage I adds extra computation and requires data that may not exist.
- Evaluation lacks diversity; no tests on other domains or larger models, so generality remains unclear.

**Questions:**

Please refer to the weaknesses above.

---

> ### Author Response · Authors · 2025-11-19
>
> Thank you for your detailed and constructive review. We appreciate their assessment of our work as a **conceptually sound** idea, related to a well-grounded method EWC. We address your main concerns clustered in the four main points, below.
> ### W1. Source Data Requirement
> > The method assumes the availability of source or surrogate data to estimate curvature, which is unrealistic for most LLM fine-tuning scenarios. [...] It is unclear how much source-domain data is required for LaLoRA to perform well [...] Stage I adds extra computation and requires data that may not exist.
>
> It is true that *exact* pre-training data is not available for most LLMs. *However*, our claim is that LaLoRa can work with only *approximate* source domain data. We firmly believe that this is not just feasible but unproblematic in practice, due to the following points:
>
> 1) **LaLoRA works with little and approximate surrogate data**
>
>     In the paper, we explicitly study how sensitive LaLoRA is to the exact surrogate data:
>     * Figure 3b shows that computing the Laplace approximation from *only one* of the three surrogate datasets (e.g., WinoGrande) already improves forgetting *on all three surrogate source datasets* (second row). This indicates that the surrogate data used for the Laplace approximation does not even need to match the evaluation task to be useful.
>
>     * In Figure 4, we vary the number of batches used for the Laplace approximation. We find that even just 1-2 mini-batches per dataset capture most of the benefits. This indicates that even a very rough surrogate (based on very little source proxy data) is sufficient. In the updated paper version, we added Figure 11, where we showcase the learning and forgetting trade-off when using *all* batches from each surrogate sub-dataset and we find that using one batch only, leads to comparable gains for forgetting, may impact the learning capabilities, yet is more efficient.
>
>     * Figure 10 further confirms Figure 3b: broader surrogates help, but even rough approximations are already valuable.
>
> 2) **Surrogate data is available in practice**
>
>     In many realistic scenarios, practitioners do have access to approximate source data:
>
>     * There are many high-quality open pre-training corpora, such as the FineWeb variants or Red Pajama, that cover similar domains and likely have a significant overlap with proprietary pre-training data.
>     * Model releases routinely report performance on public benchmarks designed to capture the very capabilities users want to preserve under fine-tuning. Using them as surrogates for the source domain is therefore aligned with how the models are evaluated during pre-training.
>
> 3) **This is a standard approach in the literature**
>
>     The need for (some notion of) source data is not unique to LaLoRA.
>
>     * Many continual learning methods rely on source surrogates, for example, EWC (Kirkpatrick et al., 2017), OSLA (Ritter et al., 2018), or replay-based methods (Wang et al., 2024, review). Following Biderman et al. (2024), we use WinoGrande/ARC/HellaSwag to "assess degradation of base model capabilities" (Biderman et al., 2024). For LaLoRA, we use the same data to compute a curvature-based regularizer to ensure that we retain these base model capabilities.
>
> **Suggestion**: We agree that the topic of source data requirements is important. We are happy to elaborate on LaLoRA's need for surrogate source data in the updated paper version by incorporating the discussion above, and can highlight it more clearly in the Limitations section.

---

> ### Author Response · Authors · 2025-11-19
>
> ### W2. Computational Costs
> > No analysis of computational efficiency relative to vanilla LoRA [...]  Stage I adds extra computation
>
> We agree that this part was not sufficiently visible and clear in the original draft. We provide an analysis of the computational cost in **Appendix F**. In Table 9, we report the measured **wall-clock runtime and GPU memory** of computing the Laplace approximation (Stage I). Table 10 provides the same for each step in the fine-tuning process (Stage II). Overall, we can observe that LaLoRA requires little overhead compared to regular LoRA fine-tuning.
>
> In the updated version, we have also added a **theoretical analysis** of the memory and compute requirements of each individual step in LaLoRA and the baselines (MIGU, MiLoRA, $L^2$) in Table 11. Table 12 compares the compute and memory requirements for different curvature approximations in LaLoRA.
>
> Please let us know what other analysis of computational efficiency you would like to see, or whether the changes addressed your concerns. We are also happy to move some of that analysis to the main text for better visibility.
>
> ---
>
> ### W3. Robustness of Results
> > Experimental results are modest [...] Only a single model/task [...] no tests on other domains or larger models
>
> We agree that the improvements of our method are modest, but visible. In most cases, it provides a **better learning-forgetting trade-off compared to the baselines**, while being a **theoretically well-substantiated** method.
> To tackle your concern about a single model and fine-tuning dataset, we have added *two additional experiments*:
> 1) We ran the same experiment using a **larger base model**, namely *Llama-3.1-8B*. Results can be found in Figure 24 (trade-off across methods), 26 (learning and forgetting across epochs) and Table 8 (validation performance).
> 2) We also tested a **different fine-tuning dataset**, namely the *MATH dataset* (see Figure 23 (trade-off across methods), 25 (learning and forgetting across epochs) and Table 7 (validation performance).
>
> We can observe the same qualitative pattern: For most (but admittedly not all) desired trade-offs of target-domain and source-domain performance, LaLoRA provides the best learning-forgetting trade-off (e.g., better source accuracy at comparable target performance or vice versa).
>
> Since LaLoRA is theoretically well-motivated and in many situations provides improvements, we see LaLoRA as a useful extension to the toolbox of practitioners. We are happy to adapt our performance claims accordingly. Please point us to statements you think are excessive and unsupported by the provided evidence.
>
> ---
>
> ### W4. Clarity
> > Equations 11-12 are poorly typeset and confusing
>
>
> Thank you for flagging this. There were indeed typos that we have now corrected in the updated version.
> Conceptually, in Stage II, we train using a regularized loss $\mathcal{L}_{\text{reg}}$ that consists of two terms: 1) the standard training loss and 2) the quadratic regularizer:
>
> 1) The "regular" loss $\mathcal{L}$ that you would use for fine-tuning.
>
>     It depends on the full model $\mathbf{W}$ (so both the fixed pre-trained weights $\mathbf{W}_0$ and the LoRA weights $\Delta \mathbf{W}$). It is evaluated on the fine-tuning dataset $\mathbb{D}_T$. Note that this is the regular target-domain negative log-likelihood, which is why we rewrite it as $-\log p(\mathbb{D}_T \mid \mathbf{W})$ and $-\log p(\mathbf{y}_t \mid \mathbf{x}_t, \mathbf{W})$ after the second and third equality.
>
> 2) The quadratic penalty regularizer $r$, scaled by a scalar regularization strength $\lambda$.
>
>     In short, we penalized deviations of the LoRA weights $\Delta \mathbf{W}$ from their initialization $\mathbf{\mu}$ with a quadratic penalty. Deviations are weighted based on the precision $\mathbf{\bar{\Sigma}}^{-1}$ (the Laplace approximation from Stage I, Equation (4)).
>
> Please let us know if the updated version clarifies this issue or what part we could elaborate on. We are happy to hear any suggestions.
>
> ---
>
> We hope our responses have addressed your concerns. If so, we would like to kindly ask you to increase your score accordingly. If you still have any remaining concerns, please, let us know.
>
> ---
>
> *Mentioned papers:*
>
> [Kirkpatrick et al., 2017] James Kirkpatrick, Razvan Pascanu, Neil Rabinowitz, Joel Veness, Guillaume Desjardins, Andrei A. Rusu, Kieran Milan, John Quan, Tiago Ramalho, Agnieszka Grabska-Barwinska, Demis Hassabis, Claudia Clopath, Dharshan Kumaran, Raia Hadsell. "Overcoming Catastrophic Forgetting in Neural Networks." 2017
>
> [Ritter et al., 2018] Hippolyt Ritter, Aleksandar Botev, David Barber. "Online Structured Laplace Approximations For Overcoming Catastrophic Forgetting." 2018
>
> [Wang et al., 2024] Liyuan Wang, Xingxing Zhang, Hang Su, and Jun Zhu. "A Comprehensive Survey of Continual Learning: Theory, Method and Application." 2024
>
> [Biderman et al., 2024] Dan Biderman et al. "LoRA Learns Less and Forgets Less." 2024

---

> ### Comment · Reviewer_CuHz · 2025-11-26
>
> I thank the authors for their rebuttal. The following points remain:
>
> > Robustness of Results
>
> "other domains" refers to other domains, not math tasks, but preferably other modalities, where finding the source data/approximate source data could be problematic/challenging. This also applies to W1.
>
> The experimental setting remains "very" limited. The LLM models the authors are testing come already equipped with a knowledge of the tasks they are testing... Please refer to the technical reports of the model.
>
> > Experimental results are modest and high-variance (Table 1; no actual gain in performance with high variance).
>
> Not convinced by the response to this concern. Variance across runs appears high compared to the baselines, suggesting that the results may not be statistically significant.

---

> > ### Author Response · Authors · 2025-12-01
> >
> > We want to further address the reviewer's concerns.
> >
> > > **Robustness and domains.**
> >
> > Our current experiments cover two model sizes (Llama-3.2-3B and Llama-3.1-8B) and two distinct math target datasets (GSM-8k and MATH), and in both cases LaLoRA shows consistent gains on the source task with competitive or improved target performance.
> >
> > **Suggestion:** We agree that it would be valuable to test LaLoRA in other domains, ideally beyond math. Due to the limited rebuttal time and compute budget, we cannot realistically add these experiments during the review period. However, we will extend our study to the coding domain for the camera-ready version. For now, we will explicitly note in the Limitations section that our experiments are restricted to the math domain.
> >
> >
> > > **Variance and robustness of the gains.**
> >
> > To clarify the variance observation, we now provide results (mean and std) for **all** hyperparameters: Table 13 (original setup on GSM-8k and Llama-3.2-3B), Table 14 for another dataset MATH (Llama-3.2-3B), and Table 15 for another model Llama-3.1-8B (GSM-8k).
> >
> > * **Table 13 (GSM-8k, Llama-3.2-3B).** LaLoRA improves source accuracy by about $+2$–$+5$ pp with small stds ($\approx 0.1$–$0.4$ pp), while target changes lie between $-2.5$ and $+1.4$ pp and are often similar to the stds. Thus, varying $\lambda$ mainly lets us trade stronger source gains against roughly comparable target performance.
> >
> > * **Table 14 (MATH, Llama-3.2-3B).** On MATH, LaLoRA reduces forgetting by up to $\approx +7$ pp; for moderate $\lambda \approx 10^2$–$2.5\times10^2$ it also yields learning gains of about $+1$–$+2$ pp with stds around $0.3$–$1.0$ pp. For more extreme $\lambda$ values, the trade-off shifts towards larger forgetting gains with flatter or slightly reduced learning.
> >
> > * **Table 15 (GSM-8k, Llama-3.1-8B).** For Llama-3.1-8B, LaLoRA gives forgetting gains of roughly $+3$–$+9$ pp and, for mid-range $\lambda \approx 10^3$–$5\times10^3$, learning gains of about $+4$–$+6$ pp, with stds typically $\leq 2$ pp. These gains generally exceed run-to-run variation, though the learning–forgetting trade-off still depends on the chosen hyperparameters.
> >
> >
> >
> > **Suggestion:** We will revise the paper text to explicitly distinguish cases where LaLoRA is comparable to LoRA from cases where it provides clear improvements. Under this more cautious interpretation, LaLoRA shows **consistent source gains** and for **certain** $\lambda$ ranges also yields **target improvements**.

---

### Official Review · Reviewer_PGEP · 2025-11-01

**Soundness:** 3
**Presentation:** 3
**Contribution:** 3
**Rating:** 2
**Confidence:** 4

**Summary:**

This paper proposes LaLoRA, a weight-space regularization technique that applies a Laplace approximation to LoRA. It estimates the model's confidence in each parameter and constrains updates in high-curvature directions, thereby preserving prior knowledge while enabling efficient target-domain learning. Experiments on mathematical reasoning tasks show an improved learning-forgetting trade-off that can be directly controlled via the method's regularization strength.

**Strengths:**

1. This paper is clearly written and well-motivated. The topic of combatting catastrophic forgetting is important.
2. The paper proposes an efficient approach to calculating the curvature information, specifically via Fisher information.
3. The experiments demonstrate that LaLoRA is effective in combatting forgetting.

**Weaknesses:**

1. The proposed method requires (a subset of) source data, which is typically unavailable for task-specific fine-tuning.
2. The significance of the proposed approach is questionable. In Figure 2(a), I find that the learning performance saturates around 2 epochs with very little forgetting. Thus, vanilla LoRA with early stopping is sufficient.
3. More baseline methods are needed, especially mentioned Bar, Flat-LoRA, etc.

[1] Implicit Regularization of Sharpness-Aware Minimization for Scale-Invariant Problems
[2] Flat-LoRA: Low-Rank Adaptation over a Flat Loss Landscape

**Questions:**

See weakness

---

> ### Author Response · Authors · 2025-11-19
>
> We thank the reviewer for their time and constructive comments. We appreciate their assessment of our work as a **well-motivated, efficient** approach to mitigate forgetting, supported by the experiments. Below, we address the three main concerns point by point.
> ### W1. Source Data Requirement
> > The proposed method requires (a subset of) source data, which is typically unavailable for task-specific fine-tuning.
>
> It is true that *exact* pre-training data is not available for most LLMs. *However*, our claim is that LaLoRa can work with only *approximate* source domain data. We firmly believe that this is not just feasible but unproblematic in practice, due to the following points:
>
> 1) **LaLoRA works with little and approximate surrogate data**
>
>     In the paper, we explicitly study how sensitive LaLoRA is to the exact surrogate data:
>     * Figure 3b shows that computing the Laplace approximation from *only one* of the three surrogate datasets (e.g., WinoGrande) already improves forgetting *on all three surrogate source datasets* (second row). This indicates that the surrogate data used for the Laplace approximation does not even need to match the evaluation task to be useful.
>
>     * In Figure 4, we vary the number of batches used for the Laplace approximation. We find that even just 1-2 mini-batches per dataset capture most of the benefits. This indicates that even a very rough surrogate (based on very little source proxy data) is sufficient. In the updated paper version, we added Figure 11, where we showcase the learning and forgetting trade-off when using *all* batches from each surrogate sub-dataset and we find that using one batch only, leads to comparable gains for forgetting, may impact the learning capabilities, yet is more efficient.
>
>     * Figure 10 further confirms Figure 3b: broader surrogates help, but even rough approximations are already valuable.
>
> 2) **Surrogate data is available in practice**
>
>     In many realistic scenarios, practitioners do have access to approximate source data:
>
>     * There are many high-quality open pre-training corpora, such as the FineWeb variants or Red Pajama, that cover similar domains and likely have a significant overlap with proprietary pre-training data.
>     * Model releases routinely report performance on public benchmarks designed to capture the very capabilities users want to preserve under fine-tuning. Using them as surrogates for the source domain is therefore aligned with how the models are evaluated during pre-training.
>
> 3) **This is a standard approach in the literature**
>
>     The need for (some notion of) source data is not unique to LaLoRA.
>
>     * Many continual learning methods rely on source surrogates, for example, EWC (Kirkpatrick et al., 2017), OSLA (Ritter et al., 2018), or replay-based methods (Wang et al., 2024, review). Following Biderman et al. (2024), we use WinoGrande/ARC/HellaSwag to "assess degradation of base model capabilities" (Biderman et al., 2024). For LaLoRA, we use the same data to compute a curvature-based regularizer to ensure that we retain these base model capabilities.
>
> **Suggestion**: We agree that the topic of source data requirements is important. We are happy to elaborate on LaLoRA's need for surrogate source data in the updated paper version by incorporating the discussion above, and can highlight it more clearly in the Limitations section.

---

> ### Author Response · Authors · 2025-11-19
>
> ### W2. LoRA with Early Stopping is Sufficient
> > In Figure 2(a), I find that the learning performance saturates around 2 epochs with very little forgetting. Thus, vanilla LoRA with early stopping is sufficient.
>
> Thank you for raising an important observation. We agree that for our setting the baseline performs best on the target domain at the second epoch. At the same time, it exhibits very comparable forgetting to our proposed approach. In this scenario, early stopping could be an interesting approach to consider. However, below we elaborate why it may not be a good approach in general:
>
> * Early stopping seems like a natural way to mitigate forgetting since the model drifts less from the pre-trained solution compared to the longer training. Yet such an approach is **not a principled way to mitigate the problem**. In a *different target dataset* (MATH, please see the added Figure 25, or for more details on the trade-off for different values 23 and validation performance Table 7), or *model* (Llama-3.1-8B, please see the added Figure 23, or for more details on the trade-off for different values 24 and validation performance Table 8) the accuracy curves look different.
>     * For MATH, we notice that for the baseline, the learning is improving across the epochs, peaking around epoch 12 where forgetting is already at around 5.5 pp.
>     * For the bigger model, we notice the same trend, with the highest accuracy at epoch 7, the respective forgetting is around 5.4pp.
>     * Additionally, LaLoRA leads to not only smaller forgetting, but **improved target accuracy compared to the baseline**. For MATH an improvement of 0.6pp for epoch 12 and 1.5pp for the final epoch. For the bigger model, an improvement of 6pp at epoch 7 or 8pp for the final epoch.
> * Additionally, early stopping would require deciding **when** to stop e.g. based on the validation performance, a target accuracy, or a metric that considers both the target and source domain performance, which may not be a trivial choice.
>
> **Suggestion**: We will mention that early stopping could be another strategy to mitigate forgetting in some cases and what are the challenges and limitations of such approach. Additionally, we updated the paper version and showcase cases (Figure 23-26, Table 7, 8) in which the explicit regularizer is needed.
>
> ---
>
> ### W3. Additional Baselines
> > More baseline methods are needed, especially mentioned Bar, Flat-LoRA, etc.
>
>
> Thank you for pointing out these related lines of work. We will add and discuss them in our Related Works section.
> * Our main goal is to mitigate *forgetting* in LoRA fine-tuning. We therefore prioritized biases that explicitly target this type of forgetting in LoRA (e.g., MiGU, MiLoRA, or $L^2$ regularization). Both BAR and Flat-LoRA, as far as we understand them, are designed to find flatter minima and *improve fine-tuning performance*. They do not evaluate or aim to improve the learning-forgetting trade-off.
> * Conceptually, these methods are complementary to LaLoRA rather than direct competitors, and in principle, one could combine these ideas.
>
> **Suggestion**: We will mention both BAR and Flat-LoRA in our Related Works section, also clarifying how they differ in objective from LaLoRA. We can also mention combining them with LaLoRA as interesting future work. If possible, we will add at least Flat-LoRA as a baseline. However, adapting their method to our different setup might require non-trivial changes and tuning.
>
> ---
>
> *Mentioned papers:*
>
> [Kirkpatrick et al., 2017] James Kirkpatrick, Razvan Pascanu, Neil Rabinowitz, Joel Veness, Guillaume Desjardins, Andrei A. Rusu, Kieran Milan, John Quan, Tiago Ramalho, Agnieszka Grabska-Barwinska, Demis Hassabis, Claudia Clopath, Dharshan Kumaran, Raia Hadsell. "Overcoming Catastrophic Forgetting in Neural Networks." 2017
>
> [Ritter et al., 2018] Hippolyt Ritter, Aleksandar Botev, David Barber. "Online Structured Laplace Approximations For Overcoming Catastrophic Forgetting." 2018
>
> [Wang et al., 2024] Liyuan Wang, Xingxing Zhang, Hang Su, and Jun Zhu. "A Comprehensive Survey of Continual Learning: Theory, Method and Application." 2024
>
> [Biderman et al., 2024] Dan Biderman, Jacob Portes, Jose Javier Gonzalez Ortiz, Mansheej Paul, Philip Greengard, Connor Jennings, Daniel King, Sam Havens, Vitaliy Chiley, Jonathan Frankle, Cody Blakeney, John P. Cunningham. "LoRA Learns Less and Forgets Less." 2024

---

> > ### Author Response · Authors · 2025-11-25
> >
> > ### Update for: W3. Additional Baselines
> > > More baseline methods are needed, especially mentioned Bar, Flat-LoRA, etc.
> >
> > We have added an additional comparison with Flat-LoRA; please see Figure 27. The figure reports the mean performance over three runs for different values of the hyperparameter $\rho$, which controls the amount of noise injected into the adapter layers. We observe that, in some settings, Flat-LoRA improves performance on the target domain relative to the baseline. However, it **does not mitigate forgetting on the source domain** (the best configuration yields only ~0.7pp improvement on the source domain and ~0.5pp on the target domain) and performs worse than all other methods.
> >
> > ---
> >
> > We hope our responses have addressed your concerns. If so, we would like to kindly ask you to increase your score accordingly. If you still have any remaining concerns, please, let us know.

---

### Author Response · Authors · 2025-12-03

We appreciate the Area Chair taking on the extra responsibility and workload. To help with their task, we provide a brief summary of the reviewers’ final positions and the outcome of the discussion phase.

Overall, the review process was very helpful: the comments were high-quality and helped strengthen the paper. We have submitted a **revised version of the manuscript with additional experiments and comments**. Below, we summarize the reviewers’ feedback and how we addressed each of their specific concerns.
* A common focus of the reviewers was our **reliance on source data**. In our response, we clarify that (i) LaLoRA works with **small amounts of *approximate surrogate* data**, as shown by our ablations (varying datasets and number of batches in the Appendix), (ii) in practice, suitable proxies (public corpora, benchmarks) are **typically available**. Perhaps most importantly, (iii) this is **a common assumption in continual learning methods** (e.g., EWC-style approaches), and by no means unique to our approach.
* In addition to the results already reported in the Appendix of the original submission, we have now added experiments to test LaLoRA in **different settings**: a **larger base model** (Llama-3.1-8B) and a **different fine-tuning dataset** (MATH). These experiments show similar qualitative learning-forgetting trade-offs and support our main claims beyond the original setup. Additionally, they highlight why early stopping (suggested by reviewer PGEP) would not be an optimal approach.
* We also addressed several smaller but important aspects: we pointed towards our **runtime and memory study** and added a **new theoretical cost comparison** to LoRA/MIGU/MiLoRA/$L^2$ and across different curvature types, included an **additional Flat-LoRA baseline**, clarified the notation and equations for the regularized loss, and provided more guidance on the effect of curvature approximations and the choice of the regularization strength (including its **relation with the LoRA rank**).

While we would have preferred to discuss these aspects with the reviewers and come to an agreement, we believe that these points, in their entirety, fully address the reviewers’ main concerns.

| Reviewer | Main concerns| Rebuttal summary |
|----------|--------------------------|------------------|
| **PGEP** | Access to source data; Comparison to early stopping; Two additional baselines | LaLoRA works with small, approximate surrogate data (shown by ablations), and such surrogates are realistic. Early stopping is not robust across datasets/models (shown in new experiments), and the added baseline (Flat-LoRA) shows LaLoRA provides better forgetting control. |
| **CuHz** | Access to source data; Compute cost study; Robustness and scope of the method | We clarify that only rough proxies are needed, add more extensive runtime/memory analyses, extend experiments to a larger model, as well as another dataset. |
| **wrdx** | Notation inconsistencies; Limited empirical setting; Dependence on proxies; Compute cost study; Rank vs regularization strength | We fix the loss/precision notation, add broader experiments (another model and dataset), provide compute/memory tables, mention proxy ablations, and provide rank-dependent sweeps for practical guidance. |
| **M9Lx** | Compute cost study; Access to source data | We quantify overhead vs LoRA/baselines, mention source domain proxy ablations, note the method extends to other PEFT methods, and discuss online Laplace. |

After we responded to the concerns of Reviewer CuHz (score 4) and Reviewer wrdx (score 6), they engaged in a further discussion focusing mainly on the robustness of LaLoRA, the choice of proxy data, and the interaction between the LoRA rank and the regularization strength. We have addressed these follow-up questions in additional detail.

Reviewer PGEP (score 2) and Reviewer M9Lx (score 8) did not reply to the rebuttal, but we believe our response substantially addresses their main concerns regarding source data requirements, early stopping, computational overhead, and additional baselines.

Overall, we have addressed the main issues raised, and we hope this summary is helpful for the meta-review process.

---

> ### Author Response · Authors · 2025-12-03
>
> ## Detailed summary
>
> ### **Reviewer PGEP**
> * **Strengths.** The reviewer highlighted that the paper is clearly written and well-motivated, that it proposes an efficient curvature-based approach, and that the experiments show LaLoRA effectively mitigates catastrophic forgetting.
> * **Initial critique.** The reviewer mentioned dependency on source data, suggested LoRA + early stopping may be enough, and asked for more baselines.
> * **Our response.** We show LaLoRA works with very little *approximate surrogate* data (shown by ablations, Fig 3b, 4, 10 and 11), argue access to such data is realistic and standard in continual learning, discuss early stopping for source domain forgetting and demonstrate cases where early stopping fails to mitigate forgetting (another model and dataset Fig 23-26 and Table 7 and 8), and add a baseline (Flat-LoRA, Fig 27) where LaLoRA leads to a better learning-forgetting trade-off.
>
> ### **Reviewer CuHz**
> * **Strengths.** They considered the idea conceptually sound, noting that it combines Laplace-based uncertainty estimation with LoRA and draws a clear connection to EWC-style continual learning in PEFT.
> * **Initial critique.** The reviewer questioned the access to source data, robustness with limited data, missing compute/memory comparison, modest/high-variance gains, and the narrow experimental setting.
> * **Our response.** We clarify that only rough proxy data and few batches are needed (shown by ablations, Fig 3b, 4, 10 and 11), add detailed runtime/memory tables and theoretical cost analysis (Table 9-12), expand experiments to a larger model and a new dataset (Fig 23-26 and Table 7 and 8), clean up the equations, and frame LaLoRA as a modest but consistent improvement (added Tables 13-15 for all hyperparameters, mean accuracy and std results).
>
> ### **Reviewer wrdx**
> * **Strengths.** The reviewer praised the paper as well-justified. They noted the efficient, clear, and mathematically well-formulated algorithm, the compelling empirical evidence with an improved learning–forgetting Pareto trade-off, and the insightful visualization and analysis of update patterns ("excellent validation") and curvature choices.
> * **Initial critique.** They asked for clearer theory/notation for multi-dataset setting and further method justification, broader experiments, compute/memory reporting, analysis of proxy dependence and discussion of curvature variants, and guidance on how regularization strength should depend on the LoRA rank.
> * **Our response.** We refine the notation and further clarify the method, add experiments with a larger model and a new target dataset (Fig 23-26 and Table 7 and 8), compute/memory tables for baselines and across different curvature types (Table 9-12), further explain different proxy settings (Fig 3b, 4, 10 and 11), address questions about curvature approximations, and provide rank sweeps (including very low rank) that illustrate how to choose the regularization strength in practice (Fig 15-20).
>
> ### **Reviewer M9Lx**
> * **Strengths.** They viewed the methodology as principled yet practical, noted that it keeps cost manageable and fits existing LoRA codepath, found that the experiments and analyses demonstrate the effectiveness and robustness across different settings, and that the data sensitivity and parameter updating are both analyzed.
> * **Initial critique.** The reviewer requested explicit overhead numbers vs LoRA/MIGU/MiLoRA and more guidance for the proxy data, and asked about using LaLoRA without pretraining data, extending it to other PEFT methods, and adapting precision online.
> * **Our response.** We report runtime and memory for Laplace and fine-tuning stages and compare theoretical costs to baselines and across different curvature types (Table 9-12), expand the discussion of ablations on proxy data (Fig 3b, 4, 10, and 11), explain that the Laplace regularizer directly extends to other PEFT adapters, and discuss online/adaptive precision as interesting but out of scope.

---

### Meta-Review · Area_Chair_rYAU · 2025-12-30

**Summary:**

This paper proposes LaLoRA, a Laplace-regularized variant of LoRA intended to mitigate catastrophic forgetting during parameter-efficient fine-tuning. Reviewers generally agreed that the paper is clearly written, technically sound, and well motivated, with a principled connection to classical continual learning methods (e.g., EWC-style regularization) and careful empirical analysis.

However, the overall consensus among reviewers was that the empirical gains are modest, and the practical significance remains unclear relative to simpler baselines. In particular, concerns were raised about (i) the reliance on source-domain or proxy data, (ii) whether early stopping or simpler regularization could already address most forgetting effects, and (iii) whether the experimental scope is sufficiently broad. While the rebuttal added experiments and clarifications, these concerns were only partially alleviated, leading me to recommend rejection.

**Reviewer Concerns:**

**Concerns substantially addressed by the rebuttal:**

- The authors provided additional clarification and ablations showing that LaLoRA can operate with small amounts of approximate surrogate data. That performance is not highly sensitive to the exact proxy dataset.
- Additional experiments on a larger model and an alternative dataset help demonstrate that the method’s behavior is qualitatively consistent beyond the original setup.
- Runtime and memory overhead were more clearly quantified, addressing earlier ambiguity around computational cost.
- Notation issues and methodological clarity raised by reviewers were largely resolved.

**Concerns that remain outstanding:**

- Practical impact and necessity: Multiple reviewers remained unconvinced that LaLoRA provides sufficiently strong or consistent improvements over simpler alternatives (e.g., early stopping, or existing LoRA heuristics), especially given the additional complexity.
- Magnitude and robustness of gains: The improvements in the learning–forgetting trade-off are often modest and sometimes high-variance, raising questions about robustness and real-world significance.
-Experimental scope: Despite added experiments, the evaluation remains limited to a small number of tasks and settings, making it challenging to assess general applicability.
- Source-data assumption: While mitigated in discussion, the need for proxy source data is still seen as a practical limitation by some reviewers, particularly in deployment-oriented scenarios.

Overall, while the rebuttal was thorough and professional, it did not fully resolve the core concerns around significance and impact.

**Reviewer Scores:**

Based on my reading of the discussion, I would expect Reviewer PGEP to either preserve their score or improve it to a weak accept. Reviewer CuHz participated in the discussion and remained unconvinced. I expect the last two reviewers to keep their mainly positive evaluation of the work.

While I have taken into account all the reviewers' comments, and the paper is technically solid and generally well-executed, I do not believe it meets the acceptance bar for ICLR as the contribution appears incremental relative to existing continual learning and LoRA-based approaches, and the empirical evidence does not convincingly demonstrate a level of impact or robustness commensurate with acceptance, especially in light of simpler competing baselines.

---

### Decision · Program_Chairs · 2026-01-26

Reject